



# Incorporating the stable carbon isotope ¹³C in the ocean biogeochemical component of the Max Planck Institute Earth System Model

**Bo Liu, Katharina D. Six, and Tatiana Ilyina**

Ocean in the Earth System, Max Planck Institute for Meteorology, Hamburg, Germany

**Correspondence:** Bo Liu (bo.liu@mpimet.mpg.de)

**Abstract.** CE1 The stable carbon isotopic composition ($\delta^{13}$C) is an important variable to study the ocean carbon cycle across different timescales. We include a new representation of the stable carbon isotope ¹³C into the HAMburg Ocean Carbon Cycle model (HAMOCC), the ocean biogeochemical component of the Max Planck Institute Earth System Model (MPI-ESM). ¹³C is explicitly resolved for all oceanic carbon pools considered. We account for fractionation during air–sea gas exchange and for biological fractionation $\epsilon_p$ associated with photosynthetic carbon fixation during phytoplankton growth. We examine two $\epsilon_p$ parameterisations of different complexity: $\epsilon_p^{\mathrm{Popp}}$ varies with surface dissolved $CO_2$ concentration (Popp et al., 1989), while $\epsilon_p^{\mathrm{Laws}}$ additionally depends on local phytoplankton growth rates (Laws et al., 1995). When compared to observations of $\delta^{13}$C of dissolved inorganic carbon (DIC), both parameterisations yield similar performance. However, with regard to $\delta^{13}$C in particulate organic carbon (POC) $\epsilon_p^{\mathrm{Popp}}$ shows a considerably improved performance compared to $\epsilon_p^{\mathrm{Laws}}$. This is because $\epsilon_p^{\mathrm{Laws}}$ produces too strong a preference for ¹²C, resulting in $\delta^{13}C_{\mathrm{POC}}$ that is too low in our model. The model also well reproduces the global oceanic anthropogenic $CO_2$ sink and the oceanic ¹³C Suess effect, i.e. the intrusion and distribution of the isotopically light anthropogenic $CO_2$ in the ocean.

The satisfactory model performance of the present-day oceanic $\delta^{13}$C distribution using $\epsilon_p^{\mathrm{Popp}}$ and of the anthropogenic $CO_2$ uptake allows us to further investigate the potential sources of uncertainty of the Eide et al. (2017a) approach for estimating the oceanic ¹³C Suess effect. Eide et al. (2017a) derived the first global oceanic ¹³C Suess effect estimate based on observations. They have noted a potential

underestimation, but their approach does not provide any insight about the cause. By applying the Eide et al. (2017a) approach to the model data we are able to investigate in detail potential sources of underestimation of the ¹³C Suess effect. Based on our model we find underestimations of the ¹³C Suess effect at 200 m by 0.24 ‰ in the Indian Ocean, 0.21 ‰ in the North Pacific, 0.26 ‰ in the South Pacific, 0.1 ‰ in the North Atlantic and 0.14 ‰ in the South Atlantic. We attribute the major sources of underestimation to two assumptions in the Eide et al. (2017a) approach: the spatially uniform preformed component of $\delta^{13}C_{\mathrm{DIC}}$ in year 1940 and the neglect of processes that are not directly linked to the oceanic uptake and transport of chlorofluorocarbon-12 (CFC-12) such as the decrease in $\delta^{13}C_{\mathrm{POC}}$ over the industrial period.

The new ¹³C module in the ocean biogeochemical component of MPI-ESM shows satisfying performance. It is a useful tool to study the ocean carbon sink under the anthropogenic influences, and it will be applied to investigating variations of ocean carbon cycle in the past.

## 1 Introduction

The stable carbon isotopic composition ($\delta^{13}$C) measured in carbonate shells of fossil foraminifera is one of the most widely used properties in paleoceanographic research (Schmittner et al., 2017). It is defined as a normalised ratio between the stable carbon isotopes ¹³C and ¹²C:

$$\delta^{13}\mathrm{C}(‰) = \left( \frac{^{13}\mathrm{C}/^{12}\mathrm{C}}{R_{\mathrm{std}}} - 1 \right) \cdot 1000, \tag{1}$$

where $R_{std}$ is an arbitrary standard ratio. In observational studies, the ratio $^{13}$C/$^{12}$C in Pee Dee Belemnite (PDB; Craig, 1957) is conventionally used for $R_{std}$.

$\delta^{13}$C provides information on past changes in water mass distribution and properties (e.g. Curry and Oppo, 2005; Peterson et al., 2014). Direct comparison between paleo-$\delta^{13}$C measurements and simulated $\delta^{13}$C facilitates evaluating the ability of Earth system models (ESMs) to simulate paleo-ocean states. For this reason, we present a new implementation of $^{13}$C in the HAMburg Ocean Carbon Cycle model (HAMOCC6), the ocean biogeochemical component of the Max Planck Institute Earth System Model (MPI-ESM). A comprehensive representation of $\delta^{13}$C is a timely extension of MPI-ESM in support of planned simulations of a complete last glacial cycle within the German climate modelling initiative PalMod (Latif et al., 2016). Before applying the new $^{13}$C module to paleo-simulations, we evaluate it by comparison to observational data in the present-day ocean.

Earlier versions of HAMOCC already featured a $^{13}$C module, for instance HAMOCC2s (Heinze and Maier-Reimer, 1999) and HAMOCC3 (Maier-Reimer, 1993). HAMOCC3 included prognostic $^{13}$C variables for dissolved inorganic carbon (DIC), particulate organic matter and calcium carbonate. HAMOCC3 also accounted for temperature-dependent isotopic fractionation during air–sea gas exchange (higher $\delta^{13}$C of surface DIC in colder water) and biological fractionation during carbon fixation. Due to the simplified representation of marine biological production in HAMOCC3, biological fractionation was based on fixation of inorganic carbon into non-living particulate organic matter and was parameterised by a spatially and temporally uniform factor. This approach for biological fractionation of $^{13}$C, however, could not reproduce the observed large meridional gradient of $\delta^{13}$C in particulate organic matter (Goericke and Fry, 1994). Since then, HAMOCC3 was refined in particular with regard to its representation of plankton dynamics. The current version HAMOCC6 resolves bulk phytoplankton, zooplankton, detritus, dissolved organic carbon (Six and Maier-Reimer, 1996) and nitrogen-fixing cyanobacteria (Paulsen et al., 2017). We thus develop an updated $^{13}$C module that considers the refined ecosystem representation and test different non-uniform parameterisations for biological fractionation during phytoplankton growth.

To choose a suitable biological fractionation parameterisation for our model, we test the parameterisations of Popp et al. (1989) and Laws et al. (1995). These parameterisations are selected for two reasons. First, they are of different complexities. The parameterisation of Popp et al. (1989) empirically relates $^{13}$C biological fractionation to the concentration of dissolved $CO_2$ in seawater, whereas that of Laws et al. (1995) considers dissolved $CO_2$ concentration and phytoplankton growth rate. Second, input variables in these two parameterisations are explicitly computed in the model. We omit more complex parameterisations that include effects of cell membrane permeability of molecular $CO_2$ diffusion, cell size and shape (e.g. Rau et al., 1996; Keller and Morel, 1999), as HAMOCC6 does not resolve these features of plankton cells.

Oceanic $\delta^{13}$C measurements were mostly carried out in the late 20th century. In the upper ocean $\delta^{13}$C in dissolved inorganic carbon ($\delta^{13}$C$_{DIC}$) has been observed to noticeably decrease in response to the intrusion of anthropogenic $CO_2$ from fossil fuel combustion which carries a lower $^{13}$C/$^{12}$C signal (Gruber et al., 1999; Quay et al., 2003). Such $\delta^{13}$C$_{DIC}$ decrease is referred to as the oceanic $^{13}$C Suess effect (Keeling, 1979). Recently, Eide et al. (2017a) derived an observation-based estimate of the global ocean $^{13}$C Suess effect since pre-industrial times. Such an observation-based estimate is valuable as it is the basis of an almost independent estimate of the global ocean anthropogenic carbon uptake. And it could be used for evaluating models at pre-industrial states (Buchanan et al., 2019; Tjiputra et al., 2020) and for setting up paleo-simulations (O'Neill et al., 2019).

Yet, Eide et al. (2017a) have noted that their approach might underestimate the oceanic $^{13}$C Suess effect. They conjectured an underestimation of the $^{13}$C Suess effect between 0.15‰–0.24‰ at 200 m depth in 1994. However, the quantitative spatial distribution of this underestimation is unclear. Moreover, although Eide et al. (2017a) have related the underestimation to several assumptions in the approach they applied, the quantitative impact of these assumptions is still unclear as the measurements are too limited in space and time to perform in-depth investigation.

Our model data include all parameters needed to apply the Eide et al. (2017a) procedure, which relies on regressional relationships between preformed $\delta^{13}$C$_{DIC}$ (related to the transport of surface waters with specific DIC and DI$^{13}$C) and CFC-12 (chlorofluorocarbon-12) partial pressure. Thus, our consistent model framework, with the complete spatio-temporal information of the hydrological and biogeochemical variables, enables us to investigate the spatial distribution of the above-mentioned potential underestimation of the oceanic $^{13}$C Suess effect. Moreover, our model framework also allows for the attribution of the underestimation to the assumptions of the procedure Eide et al. (2017a) applied.

In the following sections, we first provide a brief introduction to the global ocean biogeochemical model HAMOCC6, followed by a description of the new $^{13}$C module including the experimental setup (Sect. 2). Section 3 presents the model evaluation against observations in the late 20th century, and Sect. 4 evaluates the simulated oceanic $^{13}$C Suess effect. Section 5 addresses our findings on testing the Eide et al. (2017a) approach for estimating the oceanic $^{13}$C Suess effect. Summary and conclusions are given in Sect. 6.

## 2 Model description

### 2.1 The global ocean biogeochemical model (HAMOCC6)

HAMOCC6 (Ilyina et al., 2013; Paulsen et al., 2017; Mauritsen et al., 2019) includes biogeochemical processes in the water column and in the sediment. In the water column, the following biogeochemical tracers are simulated: dissolved inorganic carbon (DIC), total alkalinity (TA), phosphate ($PO_4$), nitrate ($NO_3$), nitrous oxide ($N_2O$), dissolved nitrogen gas ($N_2$), silicate ($SiO_4$), dissolved bioavailable iron (Fe), dissolved oxygen ($O_2$), bulk phytoplankton (Phy), cyanobacteria (Cya), zooplankton (Zoo), dissolved organic matter (DOM), particulate organic matter (POM), opal shells, calcium carbonate shells ($CaCO_3$), terrigenous material (Dust) and hydrogen sulfide ($H_2S$). Below the model-defined export depth (100 m), the sinking speed of POM linearly increases with depth. Theoretically, this leads to a power-law-like attenuation of POM fluxes as observations (Martin et al., 1987; Kriest and Oschlies, 2008). Constant sinking speeds are set for opal, $CaCO_3$ and Dust. Except for $CaCO_3$ and opal, whose sinking speeds (30 and 25 m d$^{-1}$, respectively) are considerably faster than the horizontal velocities of ocean flow, the water-column biogeochemical tracers are transported by the hydrodynamical fields in the same manner as salinity.

The sediment module is based on Heinze et al. (1999). It simulates remineralisation and dissolution processes as in the water column concerning dissolved tracers ($PO_4$, $NO_3$, $N_2$, $O_2$, $SiO_4$, Fe, $H_2S$, DIC and TA) in the pore water and the solid sediment constituents (POM, opal, $CaCO_3$). The tracers in the pore water are exchanged with the overlying water column by diffusion. Pelagic sedimentation fluxes of POM, $CaCO_3$ and opal are added to the solid components of the sediment. Below the active sediment there is one layer containing only solid sediment components and representing burial. To balance the loss of nutrients, TA, DIC and $SiO_4$ in the water column, constant input fluxes of DOM, $CO_3^{2-}$ and $SiO_4$ are added uniformly at the ocean surface, whose rates are derived from a linear regression of the long-term (approximately 100 years) temporal evolution of the sediment (active and burial) inventory.

A detailed description of HAMOCC6 is provided in Mauritsen et al. (2019) and the references therein. Different to the HAMOCC6 version in Mauritsen et al. (2019), we allow DOM degradation in low-oxygen conditions until all available $O_2$ is consumed.

### 2.2 The stable carbon isotope $^{13}$C in HAMOCC6

HAMOCC6 simulates total carbon C, which is the sum of the three natural isotopes $^{12}$C, $^{13}$C and $^{14}$C. Because in nature $^{12}$C constitutes about 98.9 % of the total carbon and $^{13}$C only constitutes about 1.1 % (Lide, 2002), in HAMOCC6 we

assume $^{12}$C = C. We include a $^{13}$C counterpart for each $^{12}$C prognostic variable; that is, we introduce seven new tracers for the water column and three for the sediment. $^{13}$C only mimics the $^{12}$C biogeochemical fluxes, modified by the corresponding isotopic fractionation. We assume $^{13}$C inventory to be as large as the inventory of $^{12}$C to reduce numerical errors. Consequently, the reference standard of the stable carbon isotope ratio $R_{std}$ is set to 1 in Eq. (1). In this section, we describe the implementation of $^{13}$C fractionation during air–sea exchange and carbon uptake by bulk phytoplankton and by cyanobacteria. Because the isotopic fractionation during the production of calcium carbonate is small (Turner, 1982) and uncertain (Zeebe and Wolf-Gladrow, 2001), it is not considered in this study, following the model studies of e.g. Lynch-Stieglitz et al. (1995), Schmittner et al. (2013) and Tjiputra et al. (2020).

#### 2.2.1 Fractionation during air–sea gas exchange

The net air–sea $CO_2$ gas exchange flux $F$ reads

$$F = -k_{CO_2}\gamma_{CO_2}\left(pCO_2^{surf} - pCO_2^{atm}\right). \tag{2}$$

Here, TS1 $pCO_2^{surf}$ and $pCO_2^{atm}$ are the partial pressures of $CO_2$ in the surface seawater and in the atmosphere, respectively. The piston velocity $k_{CO_2}$ (m s$^{-1}$) for $CO_2$ and the solubility $\gamma_{CO_2}$ (mol L$^{-1}$ atm$^{-1}$) of $CO_2$ are calculated following Wanninkhof (2014) and Weiss (1974), respectively.

Similar to the air–sea flux of total carbon in Eq. (2), the net air–sea $^{13}CO_2$ exchange flux $^{13}F$ reads

$$^{13}F = -^{13}k_{CO_2}\,^{13}\gamma_{CO_2}\left(pCO_2^{surf}R_g - pCO_2^{atm}R_{atm}\right), \tag{3}$$

in which $R_g$ and $R_{atm}$ are the ratios of $^{13}$C/$^{12}$C in surface $pCO_2$ and in atmospheric $CO_2$, respectively. Following Zhang et al. (1995), we can rewrite Eq. (3) as

$$^{13}F = -k_{CO_2}\alpha_k\gamma_{CO_2}\alpha_{aq\leftarrow g}$$
$$\left(pCO_2^{surf}\frac{R_{DIC}}{\alpha_{DIC\leftarrow g}} - pCO_2^{atm}R_{atm}\right). \tag{4}$$

Here, $\alpha_k = {}^{13}k_{CO_2}/k_{CO_2}$ is the kinetic fractionation factor TS2, $\alpha_{aq\leftarrow g} = {}^{13}\gamma_{CO_2}/\gamma_{CO_2}$ is the equilibrium isotopic fractionation factor for gas dissolution (from gaseous to aqueous $CO_2$), $\alpha_{DIC\leftarrow g} = R_{DIC}/R_g$ is the equilibrium isotopic fractionation factor from gaseous $CO_2$ to DIC and $R_{DIC} = {}^{13}C_{DIC}/{}^{12}C_{DIC}$. Parameters $\alpha_k$, $\alpha_{aq\leftarrow g}$ and $\alpha_{DIC\leftarrow g}$ are temperature-dependent, and they are obtained from laboratory experiments (Zhang et al., 1995), often expressed in terms of a per mil fractionation factor $\epsilon(\text{‰}) = (\alpha - 1) \times 10^3$:

$$\epsilon_k = -0.85, \tag{5}$$
$$\epsilon_{aq\leftarrow g} = 0.0049T_C - 1.31, \tag{6}$$
$$\epsilon_{DIC\leftarrow g} = 0.014T_C f_{CO_3} - 0.105T_C + 10.53. \tag{7}$$

https://doi.org/10.5194/bg-18-1-2021

Here, $T_C$ is the seawater temperature in °C, and $f_{CO_3} = CO_3^{2-}/DIC$ is the fraction of carbonate ions in DIC. Because in Eq. (6) the temperature dependency is weak, we use a constant $\epsilon_{aq \leftarrow g} = -1.24$, obtained at $T_C = 15\,°C$ in the model, following Schmittner et al. (2013). In Eq. (7) we neglect the first term $0.014\,T_C\,f_{CO_3}$, because $f_{CO_3}$ is generally smaller than 0.1 and because the constant factor is 1 CE2 order of magnitude smaller than that of the second term $0.105\,T_C$.

Note that Eq. (5) ($\epsilon_k = -0.85$) and the simplified Eq. (7) ($\epsilon_{DIC \leftarrow g} = -0.105\,T_C + 10.53$) in this study, adopting those of Schmittner et al. (2013), are slightly different from the OMIP protocol (Orr et al., 2017; $\epsilon_k = -0.88$ and $\epsilon_{DIC \leftarrow g} = 0.014\,T_C\,f_{CO_3} - 0.107\,T_C + 10.53$). Results of a short preindustrial simulation with $\epsilon_k$ and $\epsilon_{DIC \leftarrow g}$ from OMIP protocol yield a negligible difference (not shown). In our future simulations $\epsilon_k$ and $\epsilon_{DIC \leftarrow g}$ suggested by the OMIP protocol will be used.

### 2.2.2 Fractionation during phytoplankton growth

The lighter stable carbon isotope $^{12}$C is preferentially utilised over $^{13}$C during photosynthesis (O'Leary, 1988). Following Schmittner et al. (2013), we formulate this isotopic fractionation during net growth of the bulk phytoplankton and cyanobacteria as TS3

$$^{13}G = R_{DIC}\,\alpha_{Phy \leftarrow DIC}\,G, \tag{8}$$

with

$$\alpha_{Phy \leftarrow DIC} = \alpha_{aq \leftarrow DIC}\,\alpha_{Phy \leftarrow aq} = \frac{\alpha_{aq \leftarrow g}}{\alpha_{DIC \leftarrow g}}\,\alpha_{Phy \leftarrow aq}. \tag{9}$$

Here $G$ ($\mu mol\,C\,L^{-1}\,d^{-1}$ TS4) denotes the growth of bulk phytoplankton or cyanobacteria. $\alpha_{Phy \leftarrow DIC}$ is the isotopic fractionation factor for DIC fixation, which is determined by the equilibrium fractionation factor $\alpha_{aq \leftarrow DIC}$ from DIC to aqueous $CO_2(aq)$ and by the biological fractionation factor $\epsilon_p = (\alpha_{Phy \leftarrow aq} - 1) \times 10^3$ related to the fixation of $CO_2(aq)$. Here the subscript "Phy" denotes either the bulk phytoplankton or cyanobacteria.

We test the parameterisations for biological fractionation from Popp et al. (1989) and from Laws et al. (1995), i.e.

$$\epsilon_p^{Popp} = -17\log(CO_2(aq)) + 3.4, \tag{10}$$

$$\epsilon_p^{Laws} = \left( \frac{\mu}{CO_2(aq)/\rho_{sea}} - 0.371 \right) / 0.015. \tag{11}$$

Here, $CO_2(aq)$ ($\mu mol\,L^{-1}$) is aqueous $CO_2$ in surface water, and $\mu$ TS5 ($d^{-1}$) is the specific growth rate of bulk phytoplankton or of cyanobacteria. Note that Laws et al. (1995) measured $\epsilon_{aq \leftarrow Phy}$. Because $\alpha_{Phy \leftarrow aq}$ is close to unity, $\epsilon_p \approx -\epsilon_{aq \leftarrow Phy}$ (Zeebe and Wolf-Gladrow, 2001). In Eq. (11), we set the seawater density $\rho_{sea}$ a constant value of $1.025\,kg\,L^{-1}$. Then Eq. (11) is simplified to

$$\epsilon_p^{Laws} = 68.3\,\frac{\mu}{CO_2(aq)} - 24.7. \tag{12}$$

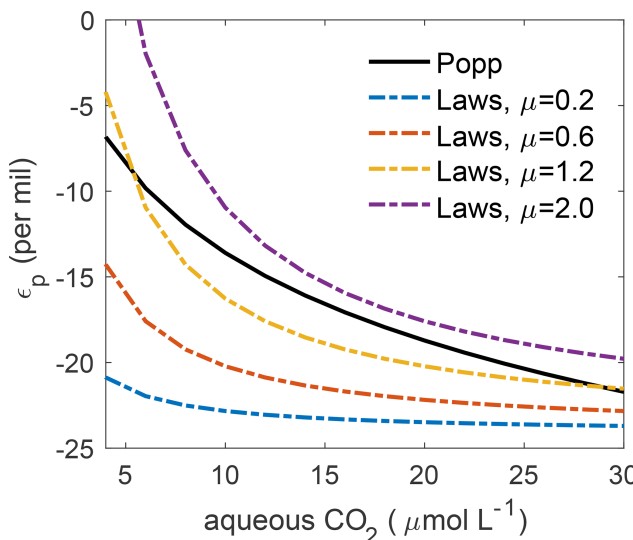

**Figure 1.** The per mil biological fractionation factor $\epsilon_p$ against aqueous $CO_2$ concentration. The solid line illustrates $\epsilon_p^{Popp}$, in which the biological fractionation during phytoplankton growth is only a function of $CO_2(aq)$. The dashed–dotted lines show $\epsilon_p^{Laws}$, which depends on $\mu/CO_2$, the ratio of phytoplankton growth rate to $CO_2(aq)$, for $\mu = 0.2$ (blue), 0.6 (red), 1.2 (yellow) and 2.0 (purple) $d^{-1}$.

Both $CO_2(aq)$ and $\mu$ (depending on local conditions of light, water temperature and nutrient availability) are determined in HAMOCC6. Figure 1 illustrates the values of $\epsilon_p^{Popp}$ and $\epsilon_p^{Laws}$ under typical ranges of $CO_2(aq)$ and $\mu$ in the ocean. When $\mu \leq 1$, $\epsilon_p^{Laws}$ is generally more negative than $\epsilon_p^{Popp}$. For high $\mu$ values, e.g. $\mu = 2$, $\epsilon_p^{Laws}$ is constantly less negative than $\epsilon_p^{Popp}$. Under high $\mu$ and low $CO_2(aq)$, $\epsilon_p^{Laws}$ becomes positive, which is unrealistic. However, our simulated ratios of phytoplankton growth rate to dissolved $CO_2$ concentration do not produce unrealistic positive $\epsilon_p^{Laws}$ at any time step in this study.

### 2.3 Model setup and experimental design

#### 2.3.1 Setup

We conduct ocean-only simulations using the MPIOM-1.6.3p1 (Jungclaus et al., 2013; Notz et al., 2013; Mauritsen et al., 2019) with HAMOCC6. MPIOM is a free-surface ocean general circulation model. It uses a curvilinear grid with the grid poles located over Greenland and Antarctica. We use a low-resolution configuration with a nominal horizontal resolution of 1.5°. This configuration has a minimum grid spacing of 15 km around Greenland and a maximum grid spacing of 185 km in the tropical Pacific. There are 40 unevenly spaced vertical levels. The layer thickness increases from 10 m in the upper ocean to 600 m in the deep ocean. The upper 100 m of the water column is represented by nine

levels. The time step is 1 h. In this setup, we additionally include the oceanic uptake and transport of CFC-12. CFC-12 is chemically inert and can therefore be treated as a conservative and passive tracer participating in all hydrodynamical processes within the ocean identical to e.g. salinity. The implementation of the air–sea gas exchange of CFC-12 follows the OMIP protocol (Orr et al., 2017).

## 2.3.2 Experimental design

For the pre-industrial spin-up simulations we cyclically apply the 1905–1929 sea-surface boundary conditions from ERA20C (Poli et al., 2016, covering 1901–2010). The atmospheric $CO_2$ mixing ratio is set to 280 ppmv. A spin-up run is first conducted without $^{13}$C tracers until the long-term averaged global net air–sea $CO_2$ flux is smaller than 0.05 Pg C yr$^{-1}$ (adequate to the C4MIP criterion for steady-state conditions of $< 0.1$ Pg C yr$^{-1}$; Jones et al., 2016). This model state is the starting point for the two spin-up runs including $^{13}$C tracers, PI_Popp and PI_Laws, which are based on the biological fractionation parameterisation $\epsilon_p^{Popp}$ (Eq. 10) and $\epsilon_p^{Laws}$ (Eq. 12), respectively.

The $^{13}$C tracers are initialised as follows. The mean $\delta^{13}$C of the marine organic matter is about $-20$‰ (Degens et al., 1968). Therefore, we set the initial concentrations of $^{13}$C in the bulk phytoplankton, cyanobacteria, zooplankton, dissolved organic carbon, particulate organic carbon in the water column and particulate organic carbon in the sediment to 0.98 (according to Eq. 1) of their $^{12}$C counterparts. The initial $^{13}$C$_{DIC}$ in the water column is calculated using the relation between $\delta^{13}$C$_{DIC}$ and $PO_4$ (Lynch-Stieglitz et al., 1995),

$$\delta^{13}C_{DIC} = 2.7 - 1.1 PO_4, \tag{13}$$

and Eq. (1). Here $PO_4$ and DIC are from the quasi-equilibrium state of the spin-up run without $^{13}$C tracers. The initial concentrations of $^{13}$C$_{CaCO_3}$ in the water column and in the sediment and the initial concentration of $^{13}$C$_{DIC}$ in pore water are set identical to their $^{12}$C counterparts.

The pre-industrial stable carbon isotope ratio $\delta^{13}CO_2$ of atmospheric $CO_2$ is fixed at $-6.5$‰. The inputs of dissolved organic $^{13}$C (DO$^{13}$C) and $^{13}CO_3^{2-}$ are uniformly added at the ocean surface. The input rate of DO$^{13}$C is calculated as the product of the input rate of DOC and the sea-surface DO$^{13}$C/DOC ratio; the input rate of $^{13}CO_3^{2-}$ is the product of the input rate of $CO_3^{2-}$ and the sea-surface $^{13}CO_3^{2-}/CO_3^{2-}$ ratio. This approach to determine $^{13}$C input rates results in a small drift in the water-column $^{13}$C inventory, but it only has minor impact on the simulation results (see Appendix A).

PI_Popp and PI_Laws are spun up for 2500 simulation years. Equilibrium states are reached with 98 % of the ocean volume having a $\delta^{13}$C$_{DIC}$ drift of less than 0.001 ‰ yr$^{-1}$ (employing the same criteria as for $^{14}$C in OMIP protocol, Orr et al., 2017). An equilibrium of the sediment is, however, not achieved for either $^{13}$C or other biogeochemical tracers.

In the transient simulations for the historical period 1850–2010, Hist_Popp and Hist_Laws, we prescribe increasing atmospheric $CO_2$ mixing ratios (Meinshausen et al., 2017) due to anthropogenic activities and decreasing atmospheric $\delta^{13}CO_2$ following OMIP and C4MIP protocols (Jones et al., 2016) (Fig. 2a). For the period 1850–1900, when forcing data are absent, we continue applying the 1905–1929 ERA20C cyclic forcing. From 1901 to 2010, we use the transient ERA20C forcing. The evolution of the atmospheric CFC-12 concentration (Fig. 2b) follows Bullister (2017). Because the atmospheric CFC-12 is slightly higher in the Northern Hemisphere, we prescribe a linear transition between 10° S and 10° N. Input rates of DO$^{13}$C, DOC, $^{13}CO_3^{2-}$, $CO_3^{2-}$ and $SiO_4$ are kept constant and are the same as those of pre-industrial simulations.

## 3 Model results and observations in the late 20th century

Our model generally simulates the physical and biogeochemical state for the present-day ocean well. The detailed model–observation comparisons for the ocean physical variables (e.g. seawater temperature and salinity, Atlantic Meridional Overturning Circulation stream function, CFC-12) and for the ocean biogeochemical tracers (e.g. primary production, nutrients, DIC) are summarised in Appendix Sects. B and C.

In this section, we compare simulated $^{13}$C between the two simulations Hist_Popp and Hist_Laws and evaluate the two experiments by comparison to observed $\delta^{13}$C$_{POC}$ and $\delta^{13}$C$_{DIC}$. The observations used here are the surface $\delta^{13}$C$_{POC}$ measurements assembled by Goericke and Fry (1994) and the observed $\delta^{13}$C$_{DIC}$, for both the surface and the interior ocean, compiled by Schmittner et al. (2013). For the model–observation comparison, we first grid the observed $\delta^{13}$C$_{POC}$ and $\delta^{13}$C$_{DIC}$ horizontally onto a $1 \times 1°$ TS7 grid and vertically (only for $\delta^{13}$C$_{DIC}$) onto the 40 depth layers of the model. Multiple data points in the same grid cell in the same month and year are averaged. Then we bilinearly interpolate the simulated monthly-mean $\delta^{13}$C$_{POC}$ and $\delta^{13}$C$_{DIC}$ over a $1 \times 1°$ grid. To quantitatively compare the performance between Hist_Popp and Hist_Laws and to other $^{13}$C models, we calculate the spatial correlation coefficient $r$ and the normalised root mean squared error (NRMSE, normalised by the standard deviation that is calculated using all the available measurements of $\delta^{13}$C$_{POC}$ or $\delta^{13}$C$_{DIC}$ during the observational periods) between model results and observation.

A global ocean climatology of pre-industrial $\delta^{13}$C$_{DIC}$ has recently been derived by first estimating the oceanic $^{13}$C Suess effect (Eide et al., 2017a) and then removing it from the observed $\delta^{13}$C$_{DIC}$ (Eide et al., 2017b). This pre-industrial $\delta^{13}$C$_{DIC}$ estimate has been used to evaluate model performance (Tjiputra et al., 2020). We do not include a $\delta^{13}$C$_{DIC}$ evaluation for the pre-industrial ocean because the historical simulations in this study facilitate the direct comparison

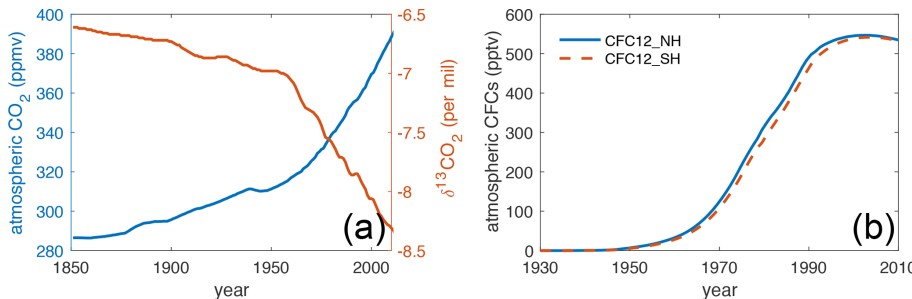

**Figure 2. (a)** The evolution of atmospheric $CO_2$ (blue, Meinshausen et al., 2017) and $\delta^{13}CO_2$ (red, Jones et al., 2016) during 1850–2010. **(b)** The evolution of atmospheric CFC-12 concentrations (Bullister, 2017). The solid blue line indicates the Northern Hemisphere, and the dashed red line indicates Southern Hemisphere. TS6

to observations in the late 20th century, which is different from Tjiputra et al. (2020), who only include pre-industrial simulations with $^{13}$C tracers. Moreover, as has already been discussed by Eide et al. (2017a) and is discussed in Sect. 5 of this study, the $^{13}$C Suess effect is possibly underestimated by the Eide et al. (2017a) approach. This suggests Eide et al. (2017b) likely overestimate the pre-industrial $\delta^{13}C_{DIC}$.

### 3.1 Isotopic signature of particular organic carbon in the surface ocean

For comparison between Hist_Popp and Hist_Laws, the climatological mean state of $\delta^{13}C_{POC}$ is derived by averaging over 1960-1991, the period when most $\delta^{13}C_{POC}$ measurements were collected. In Hist_Popp, the climatological annual-mean surface $\delta^{13}C_{POC}$ has a global mean value of $-22.5\,‰$, and it shows a distinct horizontal pattern (Fig. 3a). Less negative values up to $-19.3\,‰$ are found in the subtropical regions, where alkalinity is typically high and $CO_2(aq)$ is consequently low. This low $CO_2(aq)$ results in a smaller isotope fractionation during carbon fixation by phytoplankton (Eq. 10, Fig. 1) with a biological fractionation factor $\epsilon_p > -13\,‰$ (Fig. 3c). Poleward of the subtropical regions, $\delta^{13}C_{POC}$ gradually decreases. The reason for this is twofold. First, $\epsilon_p$ decreases from $-13\,‰$ to about $-20\,‰$ following the increase in $CO_2(aq)$. Second, the thermal effect of equilibrium fractionation causes about $3\,‰$ more fractionation in the polar regions than in the tropical and subtropical regions (according to Eqs. 7 and 9). The lowest $\delta^{13}C_{POC}$ of about $-30\,‰$ occurs close to Antarctica where highest surface DIC concentrations are typically found because of the upwelling of deep waters and the reduced air–sea gas exchange by ice cover (Takahashi et al., 2014). The annual range of $\delta^{13}C_{POC}$ (Fig. 3e), i.e. the difference between the minimum and the maximum of its climatological monthly-mean annual cycle, is low ($< 0.5\,‰$) in the subtropical regions, and it increases polewards up to $\sim 9\,‰$ in the Southern Ocean, mirroring meridional changes in the annual range of $CO_2(aq)$.

Compared to Hist_Popp, Hist_Laws shows lower annual-mean surface $\delta^{13}C_{POC}$ (Fig. 3b), with a global-mean value

of $-29.9\,‰$ due to more negative $\epsilon_p$ (Fig. 3d). This is because $\epsilon_p^{Laws}$ (Fig. 1) is always more negative than $\epsilon_p^{Popp}$ when the simulated mean growth rates (Fig. C1a and b) are lower than $1\,d^{-1}$. As $\epsilon_p^{Laws}$ increases with growth rate (Eq. 12), we find less negative $\delta^{13}C_{POC}$ (up to $-24.1\,‰$) in the central tropical Pacific, where highest growth rates are simulated (Fig. C1a and b). The lowest $\delta^{13}C_{POC}$ of $-33\,‰$ occurs in the Arctic Ocean and around Antarctica due to the combination of low growth rate, high $CO_2(aq)$ and low seawater temperature. The meridional range of the annual-mean $\delta^{13}C_{POC}$ in Hist_Laws ($\sim 9\,‰$) is smaller than that of Hist_Popp ($\sim 11\,‰$) because for low growth rates $\epsilon_p^{Laws}$ is generally less sensitive to $CO_2(aq)$ changes compared to $\epsilon_p^{Popp}$ (Fig. 1). This also results in a smaller annual range of $\delta^{13}C_{POC}$ in high latitudes (Fig. 3f) than Hist_Popp (Fig. 3e). In the low and mid-latitudes, Hist_Laws shows larger annual range of $\delta^{13}C_{POC}$ because in these regions $CO_2(aq)$ concentrations are relatively stable but growth rates shows noticeable seasonal variability.

Hist_Popp captures major features of the observed $\delta^{13}C_{POC}$ (Fig. 4a, c and e). The meridional gradient, with less negative values in the low latitudes and minimal values around $60°\,S$, is well reproduced. In contrast, Hist_Laws shows generally lower $\delta^{13}C_{POC}$ than the observations (a global mean bias of $-8\,‰$) and a smaller $\delta^{13}C_{POC}$ difference between low and high latitudes (Fig. 4b, d and f). This is also seen in a recent study by Dentith et al. (2020), who tested $\epsilon_p^{Popp}$ and $\epsilon_p^{Laws}$ with the FAst Met Office/UK Universities Simulator (FAMOUS). The underestimation in the global mean and in the meridional gradient of $\delta^{13}C_{POC}$ in Hist_Laws suggests that the parameters of the linear fit in Eq. (12) (slope and intercept) would need to be increased to gain a better performance. Around $60°\,S$ of the Atlantic Ocean (Fig. 4b), Hist_Laws simulates a smaller range of $\delta^{13}C_{POC}$ than the observations. This is also a result of the small $\delta^{13}C_{POC}$ annual range produced by $\epsilon_p^{Laws}$ (Fig. 3f). Between $40°\,S$ and $40°\,N$ in the Atlantic Ocean, Hist_Laws simulates $\delta^{13}C_{POC}$ peaks in the region of high growth rates south

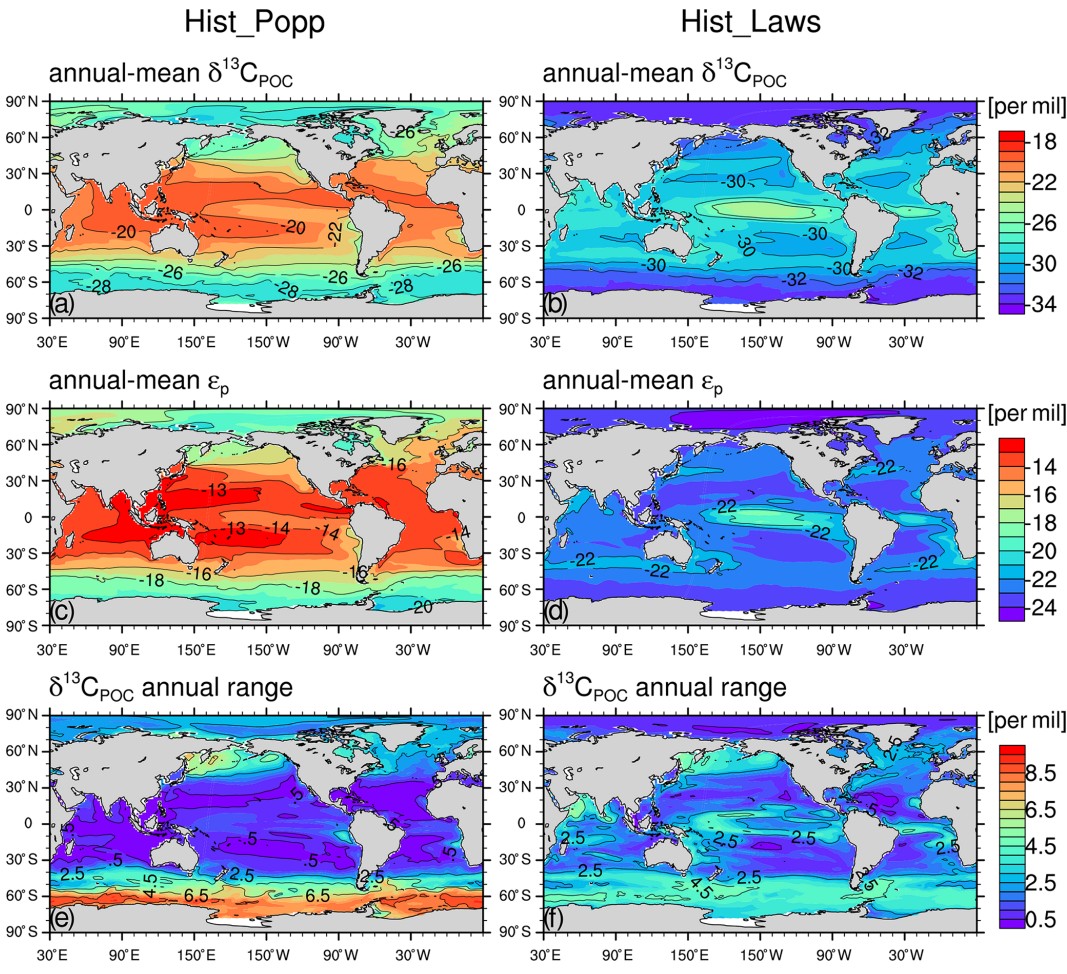

**Figure 3.** The climatological (1960–1991) annual-mean surface values for Hist_Popp **(a, c, e)** and Hist_Laws **(b, d, f)** for $\delta^{13}C_{POC}$ **(a, b)**, $\epsilon_p$ **(c, d)**, and for the annual range of $\delta^{13}C_{POC}$ **(e, f)**. All values are given in per mil (‰).

of the Equator, whereas the observed high $\delta^{13}C_{POC}$ values are located between the Equator and 20° N.

In the Indian Ocean around 45° S, Hist_Popp does not capture the prominent $\delta^{13}C_{POC}$ peak in the field data (Fig. 4e), despite the fact that the simulated $CO_2$(aq), the controlling factor in the parameterisation $\epsilon_p^{Popp}$ (Eq. 10), well reproduces the meridional variation of the contemporaneous $CO_2$(aq) measurements (Fig. 4g). Although the empirical correlation between $\epsilon_p$ and $CO_2$(aq), such as Eq. (10), holds true to the first order over large areas of the global ocean, other factors, such as growth rate, affect the local variability in $\epsilon_p$ (Popp et al., 1998; Hansman and Sessions, 2016; Tuerena et al., 2019). Hist_Laws captures the $\delta^{13}C_{POC}$ peak around 45° S in the observations (Fig. 4f), owing to the dependency of $\epsilon_p^{Laws}$ on phytoplankton growth rate and to the model successfully reproducing the high productivity in this region (illustrated by phytoplankton biomass, Fig. 4h). This is in alignment with the field study by Francois et al. (1993) and the model study by Hofmann et al. (2000), who ascribed this observed

$\delta^{13}C_{POC}$ peak to a local high phytoplankton production during the measurement period.

Overall, Hist_Popp ($r = 0.84$ and NRMSE = 0.57) better reproduces the observed $\delta^{13}C_{POC}$ than Hist_Laws ($r = 0.71$, NRMSE = 2.5). Here a higher NRMSE indicates the model captures a smaller fraction of the variation in observations. The performance of Hist_Popp regarding $\delta^{13}C_{POC}$ compares well to that of the FAMOUS model (Dentith et al., 2020; comparing their Fig. 8 and Fig. 4 in this study CE3) and the University of Victoria (UVic) Earth System Model of intermediate complexity (with $r = 0.74$ and NRMSE = 0.92; Schmittner et al., 2013). Note that Schmittner et al. (2013) compared climatological annual-mean model output to the $\delta^{13}C_{POC}$ measurements from Goericke and Fry (1994), whereas our study uses model results of the corresponding month and year of the measurements. This difference leads to a better comparison of Hist_Popp to the observed $\delta^{13}C_{POC}$ in high latitudes, particularly in the South Atlantic Ocean around 60° S, and therefore it is one reason for the slight better performance of Hist_Popp compared to Schmittner et al.

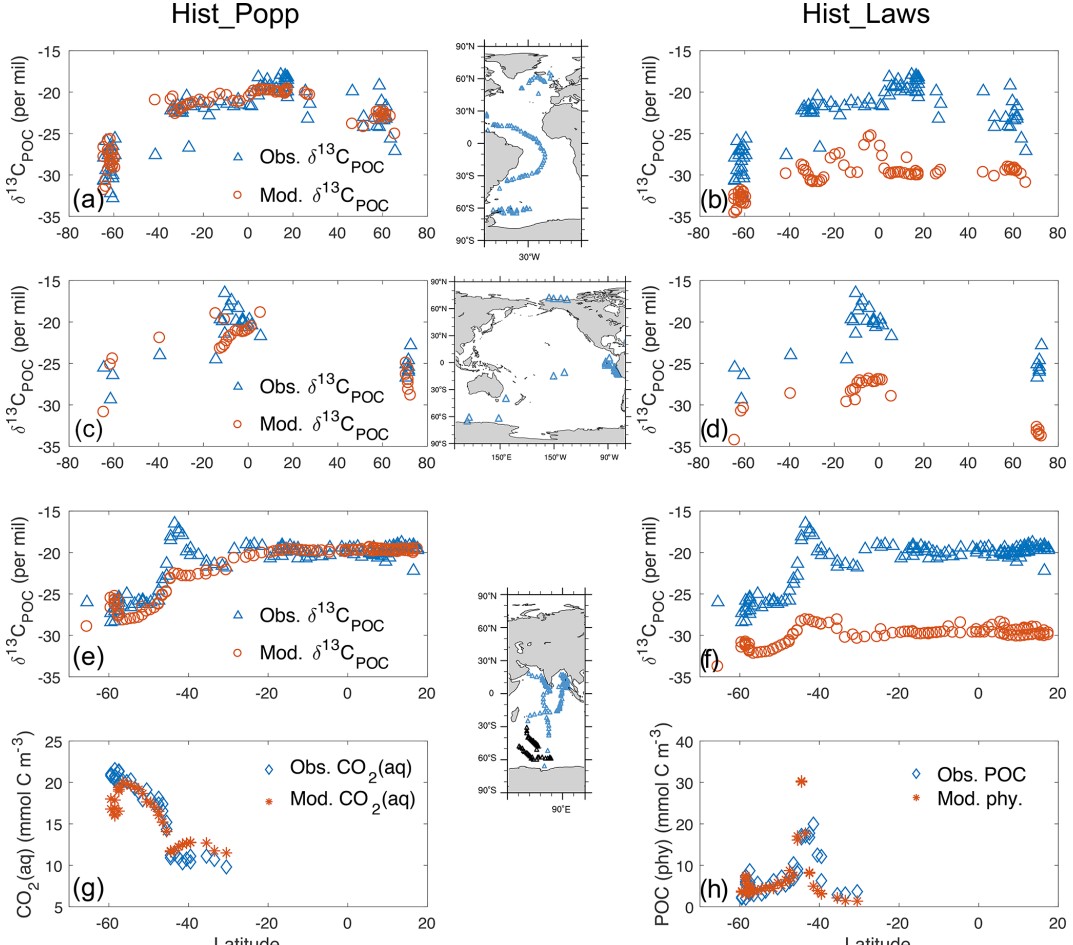

**Figure 4.** Comparison of surface $\delta^{13}C_{POC}$ (‰) observations (blue triangle) from Goericke and Fry (1994) to model data (red circle) in Hist_Popp (**a, c, e**) and Hist_Laws (**b, d, f**) for the Atlantic, Pacific and Indian Ocean, respectively. Inserted maps show cruise tracks of the measuring campaigns. (**g**) Comparison of simulated $CO_2(aq)$ (red star) to observations (blue diamond) in the South Indian Ocean (Francois et al., 1993; measurement locations indicated by black triangles in the inset map for the Indian Ocean). Panel (**h**) is as panel (**g**) but for particulate organic matter, represented by total POC in Francois et al. (1993) and by phytoplankton biomass in the model. The measurement precision is $\pm 0.17$‰ for $\delta^{13}C_{POC}$ and 2 % for $CO_2(aq)$ and particulate organic matter, according to Francois et al. (1993).

(2013), aside from the underlying differences between the two models.

Hist_Popp also well reproduces the temporal changes of the biological fractionation factor $\epsilon_p$ when compared to the observation-based estimates of Young et al. (2013). In Hist_Popp, the change rate of $\epsilon_p$ has a global-mean value of $-0.026$‰ yr$^{-1}$ for the period 1960–2009 (Fig. C7a), similar to an estimate of $-0.022$‰ yr$^{-1}$ in Young et al. (2013). Modest $\epsilon_p$ changes are found in eastern tropical Pacific and south of 60° S, in good agreement with Young et al. (2013). Hist_Laws, on the other hand, shows a very small global-mean $\epsilon_p$ change rate of $-0.005$‰ yr$^{-1}$ (Fig. C7b) as $\epsilon_p^{Laws}$ is less sensitive to the increase in $CO_2(aq)$ than $\epsilon_p^{Popp}$.

## 3.2 Isotopic signature of dissolved inorganic carbon $\delta^{13}C_{DIC}$

### 3.2.1 Comparison between Hist_Popp and Hist_Laws and to observations

Figures 5a–b and 6a–f compare the climatological annual mean of $\delta^{13}C_{DIC}$ (averaged over 1990–2005, when most $\delta^{13}C_{DIC}$ measurements were collected) between Hist_Popp and Hist_Laws. The two simulations exhibit very similar $\delta^{13}C_{DIC}$ patterns for both the surface and interior ocean. The surface seawater DIC is enriched in $^{13}$C due to the preferential uptake of the light isotope $^{12}$C by phytoplankton during primary production. As particulate organic matter sinks and is remineralised at depth, the negative $\delta^{13}C_{POC}$ signal is released. Consequently, in both Hist_Popp and Hist_Laws, $\delta^{13}C_{DIC}$ at the surface is generally higher than in the ocean

interior. At the surface of the equatorial central Pacific, the eastern boundary upwelling systems and the Southern Ocean south of 60° S, lower $\delta^{13}C_{DIC}$ ($< 1.6‰$) is seen due to the upward transport of the $^{13}$C-depleted water (Fig. 5a and b). In the interior ocean, we find higher $\delta^{13}C_{DIC}$ ($> 1‰$) in well-ventilated water masses, in particular the North Atlantic Deep Water (NADW) (Fig. 6a and d). The lowest $\delta^{13}C_{DIC}$ values ($< -0.5‰$) occur at depth in tropical and subtropical regions (Fig. 6a–f), where a large amount of organic matter is remineralised.

The global-mean surface $\delta^{13}C_{DIC}$ of the two experiments only differs marginally (1.64‰ for Hist_Popp and 1.7‰ for Hist_Laws), which is expected as they are run using the same prescribed atmospheric $\delta^{13}CO_2$ (Schmittner et al., 2013). Given very similar mean surface DI$^{13}$C, the larger vertical DI$^{13}$C gradients in Hist_Laws, established by more negative $\delta^{13}C_{POC}$ (Fig. 3a and b), yield lower DI$^{13}$C concentration at depth. This adjustment of DI$^{13}$C content in the ocean interior takes place during the pre-industrial spin-up phase of the simulations via air–sea $^{13}CO_2$ exchange (Appendix A). At the end of the 2500-year spin-up, the water-column DI$^{13}$C inventory in PI_Laws is $1.1 \times 10^{12}$ kmol lower than PI_Popp, yielding a global mean $\delta^{13}C_{DIC}$ difference of 0.25‰ (Fig. 6g–i). Such interior-ocean $\delta^{13}C_{DIC}$ difference caused by using different parameterisation for biological fractionation is also seen in Jahn et al. (2015) and Dentith et al. (2020). The seasonal upward transport of the lower deep-ocean $\delta^{13}C_{DIC}$ in Hist_Laws leads to lower annual-mean surface $\delta^{13}C_{DIC}$ and larger $\delta^{13}C_{DIC}$ annual range in regions of upwelling (Fig. 5c and d).

When compared to the observed $\delta^{13}C_{DIC}$, Hist_Popp ($r = 0.81$, NRMSE $= 0.7$) has a slightly better performance than Hist_Laws ($r = 0.80$, NRMSE $= 1.1$). Hist_Laws generally shows vertical gradients of $\delta^{13}C_{DIC}$ that are too strong and therefore $\delta^{13}C_{DIC}$ values that are too low in the ocean interior, as is seen in the depth profiles of horizontally averaged $\delta^{13}C_{DIC}$ (Fig. 7). This points to too strong a preference for the isotopically light carbon simulated by $\epsilon_p^{Laws}$ as is already discussed in Sect. 3.1. Given the slightly better performance of Hist_Popp than Hist_Laws regarding $\delta^{13}C_{DIC}$, we focus in the following on the comparison between Hist_Popp and observed $\delta^{13}C_{DIC}$.

### 3.2.2 Source of surface $\delta^{13}C_{DIC}$ biases in Hist_Popp

Figure 8 contains model–observation comparison for the surface $\delta^{13}C_{DIC}$. Overall, the magnitude and spatial distribution of the observed $\delta^{13}C_{DIC}$ is well-captured by Hist_Popp. In the surface ocean, the mean $\delta^{13}C_{DIC}$ is slightly overestimated by Hist_Popp (1.7‰ compared to 1.5‰ in observation). Positive biases are widely seen in the Indian and Pacific Ocean, and the negative biases are mostly found in the Atlantic Ocean (Fig. 8c). To better understand the source of differences between model and observations, we follow the method of Broecker and Maier-Reimer (1992) to decompose

$\delta^{13}C_{DIC}$ into a biological component $\delta^{13}C_{DIC}^{bio}$ and a residual component $\delta^{13}C_{DIC}^{resi}$, driven by air–sea exchange and ocean circulation:

$$\delta^{13}C_{DIC}^{bio} = \delta^{13}C_{DIC}|_{M.O.} + \frac{\Delta_{photo}}{DIC_{M.O.}} R_{C:P}(PO_4 - PO_4|_{M.O.}). \quad (14)$$

Here the subscript M.O. refers to mean ocean values, $\Delta_{photo}$ is the carbon isotope fractionation during marine photosynthesis and $R_{C:P}$ is the C : P ratio of marine organic matter. We use $\Delta_{photo} = -19‰$ (Eide et al., 2017b) and $R_{C:P} = 122$ (Takahashi et al., 1985) for both model and observational data. In reality $\Delta_{photo}$ shows spatial variability due to the variations of $CO_2(aq)$ (Fig. 3c) and temperature (Eq. 7) at the sea surface. However, using a constant $\Delta_{photo}$ only has limited quantitative impact on the model–observation comparison of the two components. To calculate $\delta^{13}C_{DIC}^{bio}$ from observations, we employ $\delta^{13}C_{DIC}|_{M.O.} = 0.5‰$, $DIC_{M.O.} = 2255$ mmol m$^{-3}$ (Eide et al., 2017b) and $PO_4$ from the World Ocean Atlas (WOA13; Garcia et al., 2013a). Considering the strong seasonality in $PO_4$ in the surface ocean, we select the phosphate concentration from the climatological monthly WOA data (available only for the upper 500 m of the water column) and the climatological monthly-mean model data for the same month as the $\delta^{13}C_{DIC}$ observations. The observed mean ocean phosphate concentration $PO_4|_{M.O.} = 1.7$ mmol m$^{-3}$ is obtained by first merging the time mean of the $PO_4$ monthly WOA data in the upper 500 m and the $PO_4$ annual-mean WOA data below 500 m and then mapping the combined data to the vertical grid of our model. For simulated $\delta^{13}C_{DIC}^{bio}$, the model data of $\delta^{13}C_{DIC}|_{M.O.} = 0.67‰$, $DIC_{M.O.} = 2197$ mmol m$^{-3}$, $PO_4|_{M.O.} = 1.5$ mmol m$^{-3}$ and $PO_4$ are used. The model–observation $\delta^{13}C_{DIC}^{resi}$ difference is calculated by subtracting the model–observation $\delta^{13}C_{DIC}^{bio}$ difference from the model–observation $\delta^{13}C_{DIC}$ difference.

The model captures the major features of the observed $\delta^{13}C_{DIC}^{bio}$ at the surface; that is, higher values are seen in the subtropical regions and lower values in the high latitudes (Fig. C8a and b). Nevertheless, noticeable quantitative differences exist (Fig. 9a), which resemble the distribution of $(PO_4 - PO_4|_{M.O.}$ TS8$)$ bias (Fig. 9b). Between 30° N and 30° S in the surface ocean, we find a mean negative bias of about $-0.1‰$. This is caused by the underestimation of primary production in the subtropical gyres (due to the underestimation of phytoplankton growth rates; see Appendix C1) and the consequently reduced enrichment of $^{13}$C in surface DIC. A strong positive $\delta^{13}C_{DIC}^{bio}$ bias of 0.6‰ to 1‰ is seen in the North Pacific, where in the model iron is not a limiting nutrient (Fig. C3), in contrast to observations (Moore et al., 2013). In the equatorial central Pacific, a weak positive $\delta^{13}C_{DIC}^{bio}$ bias $< 0.2‰$ is caused by a primary production that is too high. Specifically, the simulated phytoplankton growth rates in this region compare well to observations, whereas the simulated phytoplankton biomass is too high (Appendix C1). The latter is mainly induced by an upwelling that is too strong. The observed mean upward vertical veloc-

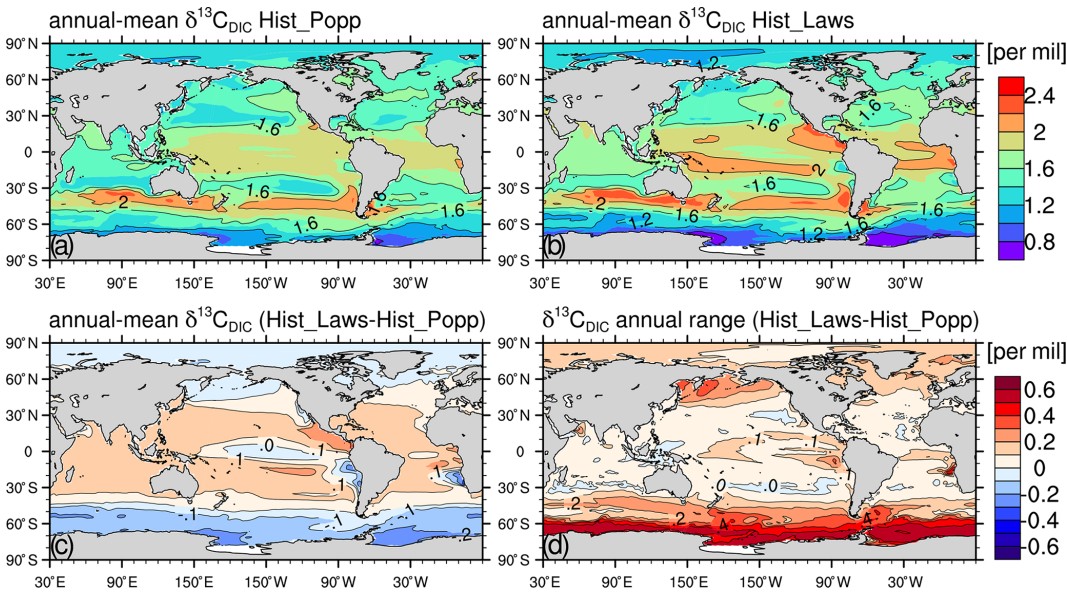

**Figure 5.** Climatological (averaged over 1990–2005) annual-mean surface $\delta^{13}$C$_{DIC}$ for Hist_Popp **(a)** and Hist_Laws **(b)**, respectively. Panels **(c)** and **(d)** show the difference in the climatological annual-mean $\delta^{13}$C$_{DIC}$ between Hist_Laws and Hist_Popp, and the difference in the climatological annual range of $\delta^{13}$C$_{DIC}$ between the two simulations, respectively.

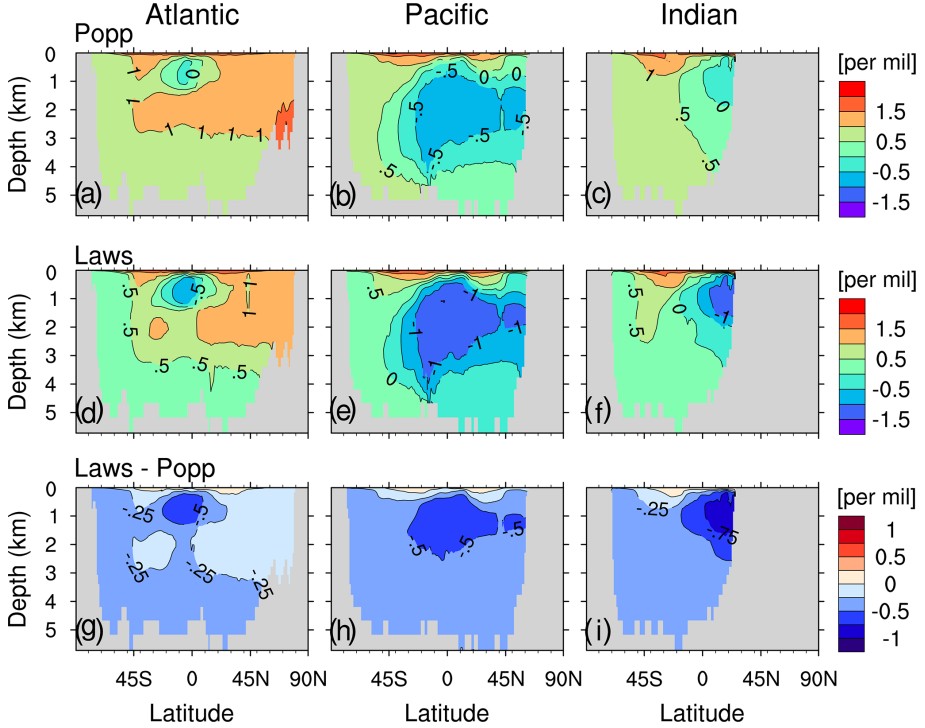

**Figure 6.** Zonal-mean $\delta^{13}$C$_{DIC}$ of the Atlantic Ocean **(a, d, g)**, the Pacific Ocean **(b, e, h)** and the Indian Ocean **(c, f, i)** for Hist_Popp **(a–c)**, Hist_Laws **(d–f)** and for the difference between Hist_Laws and Hist_Popp **(g–i)**.

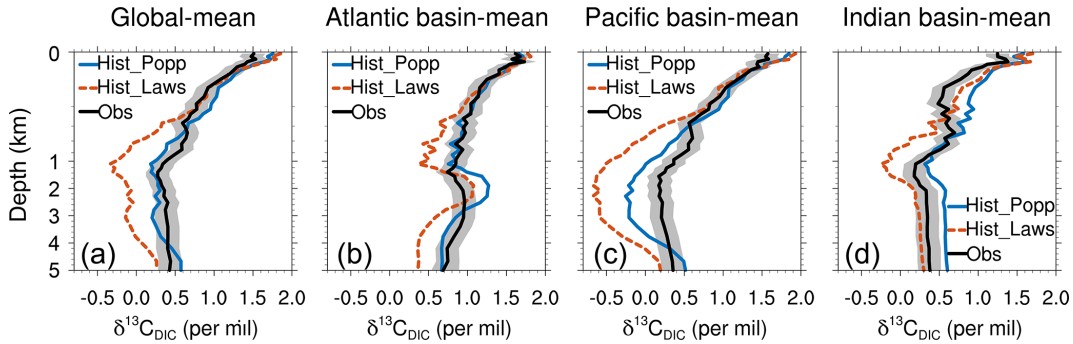

**Figure 7.** Depth profiles of horizontally averaged $\delta^{13}C_{DIC}$ of Hist_Popp (solid blue line), Hist_Laws (dashed red line) and the observational data from Schmittner et al. (2013) (solid black line) for the global ocean **(a)**, the Atlantic Ocean **(b)**, the Pacific Ocean **(c)** and for the Indian Ocean **(d)**. The grey shading indicates observation uncertainty of $\pm0.15\,‰$, which relates to the estimated accuracy due to unresolved intercalibration issues between laboratories (0.1‰–0.2‰; Schmittner et al., 2013).

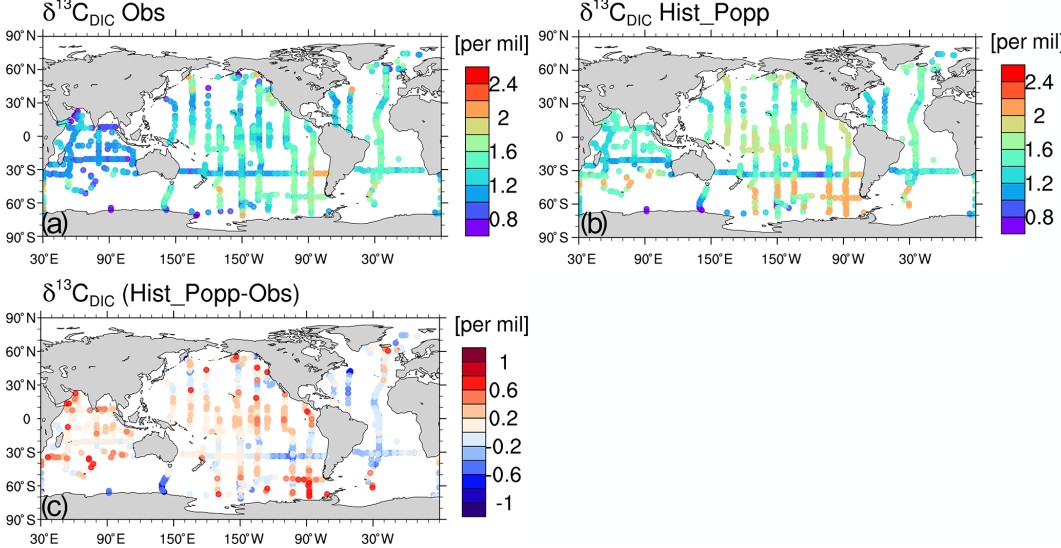

**Figure 8.** Observed surface $\delta^{13}C_{DIC}$ (Schmittner et al., 2013) **(a)** and simulated $\delta^{13}C_{DIC}$ in Hist_Popp sampled at the location, month and year of the observation **(b)**. **(c)** The difference in $\delta^{13}C_{DIC}$ between Hist_Popp and observations.

ity at 0, 140° W and 60 m depth during May 1990–June 1991 is $2.3 \times 10^{-5}\,\mathrm{m\,s^{-1}}$ (Weisberg and Qiao, 2000), whereas the model simulates $3.2 \times 10^{-5}\,\mathrm{m\,s^{-1}}$ for the same location and period.

In the Southern Ocean, a strong positive $\delta^{13}C_{DIC}^{bio}$ bias of 0.6 to 1‰ (Fig. 9a) results from a primary production that is too high under surface iron concentrations that are too high (0.2–0.4 nmol L$^{-1}$ compared to generally $< 0.25$ nmol L$^{-1}$ from data of the GEOTRACES programme (http://www.geotraces.org, last access: 15 April 2021, not shown). Primary production is limited by iron only south of 50° S in the model compared to south of 40° S from observation (Moore et al., 2013). One cause for the high surface iron concentration is that in HAMOCC6 CE4 organic matter is remineralised at depths that are too shallow. This can been seen from

the positive apparent oxygen utilisation (AOU) biases above 500 m south of 45° S (Fig. 10j–l).

Another reason for the high surface iron concentration in the Southern Ocean is that MPIOM simulates an upwelling that is too strong. In particular, below 1000 m, the simulated upward velocity shows noticeably larger magnitude ($> 5 \times 10^{-6}\,\mathrm{m\,s^{-1}}$, Fig. B4) than that of a dynamically consistent and data-constrained ocean state estimate (see Fig. 1 in Liang et al., 2017). The upwelling that is too strong in the model is consistent with the volume transport that is too large across the Drake Passage of 192 Sv compared to 134–173 Sv from observations (Nowlin Jr. and Klinck, 1986; Cunningham et al., 2003; Meredith et al., 2011; Donohue et al., 2016). Our model also features larger downward velocities than the estimate from Liang et al. (2017), which correspond to mixed layer depths that are too deep in the Southern

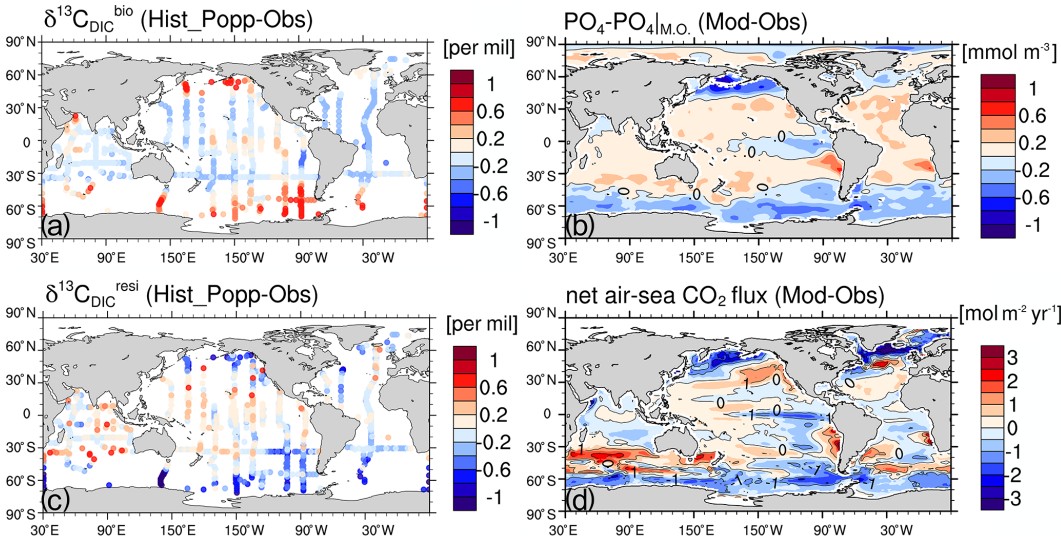

**Figure 9.** Model–observation difference in the biological component $\delta^{13}C_{DIC}^{bio}$ **(a)**, $(PO_4 - PO_4|_{M.O.})$ **(b)**, and the residual component $\delta^{13}C_{DIC}^{resi}$ **(c)** at the ocean surface. **(d)** The net air–sea $CO_2$ flux (positive into the air, averaged over 1990–2005) difference between the model data and observation-based data product from Landschützer et al. (2015).

Ocean (up to 3000 m, Fig. B5) than observations ($< 700$ m; de Boyer Montégut et al., 2004; Holte et al., 2017).

We find strong $\delta^{13}C_{DIC}^{resi}$ negative biases of $-0.5$‰ to $-1$‰ (Fig. 9c) in the North Pacific and the Southern Ocean, which partially compensate for the positive biases of $\delta^{13}C_{DIC}^{bio}$ (Fig. 9a) in these regions. One major cause for the negative $\delta^{13}C_{DIC}^{resi}$ bias in these two regions is our model overestimating the uptake of anthropogenic carbon, as is illustrated by the net air–sea $CO_2$ difference between the model and the observation (Fig. 9d). Consequently, the decreased atmospheric $^{13}C/^{12}C$ ratio over the industrial period further lowers $\delta^{13}C_{DIC}$ in the two ocean regions in the model. In the Southern Ocean, the upward transport of $^{13}C$-depleted water is too large, which also contributes to a negative $\delta^{13}C_{DIC}^{resi}$ bias CE6.

### 3.2.3 Source of $\delta^{13}C_{DIC}$ biases in the interior ocean of Hist_Popp

Figure 10 contains the model–observation comparison for zonal-mean $\delta^{13}C_{DIC}$ in the Atlantic, Pacific and Indian Ocean. In the interior ocean, $\delta^{13}C_{DIC}$ is controlled by remineralisation of $^{13}C$-depleted organic matter and by ocean circulation (Broecker and Peng, 1993; Lynch-Stieglitz et al., 1995; Schmittner et al., 2013). Low $\delta^{13}C_{DIC}$ is often found in waters of high nutrient concentration and vice versa. Thus, we find positive (negative) $\delta^{13}C_{DIC}$ biases coincide with negative (positive) phosphate biases (Fig. 10d–i). In the Atlantic Ocean between 1000 and 3000 m, the North Pacific above 1500 m and the Indian Ocean below 1000 m, positive $\delta^{13}C_{DIC}$ biases and negative phosphate biases are mainly

caused by a remineralisation that is too low, as is shown by the negative AOU biases (Fig. 10j–l).

North of $30°$ S in the Atlantic Ocean, the negative $\delta^{13}C_{DIC}$ biases below 3000 m, together with the positive $\delta^{13}C_{DIC}$ biases between 1000 and 3000 m, suggest $\delta^{13}C_{DIC}$ vertical gradients that are too strong in the model (Fig. 10d). This results from a lower boundary of the NADW cell that is too shallow, constantly located above 2800 m (Fig. B3), compared to an estimated NADW lower boundary of about 4300 m deep at $26°$ N (Msadek et al., 2013; Smeed et al., 2017). A possible reason for the shallow NADW in the model is that the Lower North Atlantic Deep Water (LNADW), forming from the Denmark Strait Overflow Water and the Iceland–Scotland Overflow Water, is not dense enough to flow further southward. This can be seen from the CFC-12 distribution along the zonal Sect. A5 at $24°$ N (Fig. B7). The observed deeper CFC-12 maximum (3000–4500 m west of $60°$ W) indicates the presence of LNADW (Dutay et al., 2002), which is not represented in our model.

We find the strongest negative $\delta^{13}C_{DIC}$ bias in the deep eastern equatorial Pacific (Fig. 10e). The cause is the "nutrient trapping" problem in the model, characterised by nutrient concentrations that are too high in the deep eastern equatorial Pacific (Fig. 10h), which is a persistent problem in many ESMs (Aumont et al., 1999; Dietze and Loeptien, 2013). Based on sensitivity experiments with the Geophysical Fluid Dynamics Laboratory model and the UVic model, Dietze and Loeptien (2013) concluded the primary cause of the nutrient trapping problem is likely model biases in physical ocean state – in particular, the poor representation of the Equatorial Intermediate Current system and equatorial deep jets. The latter two current systems are indeed poorly repre-

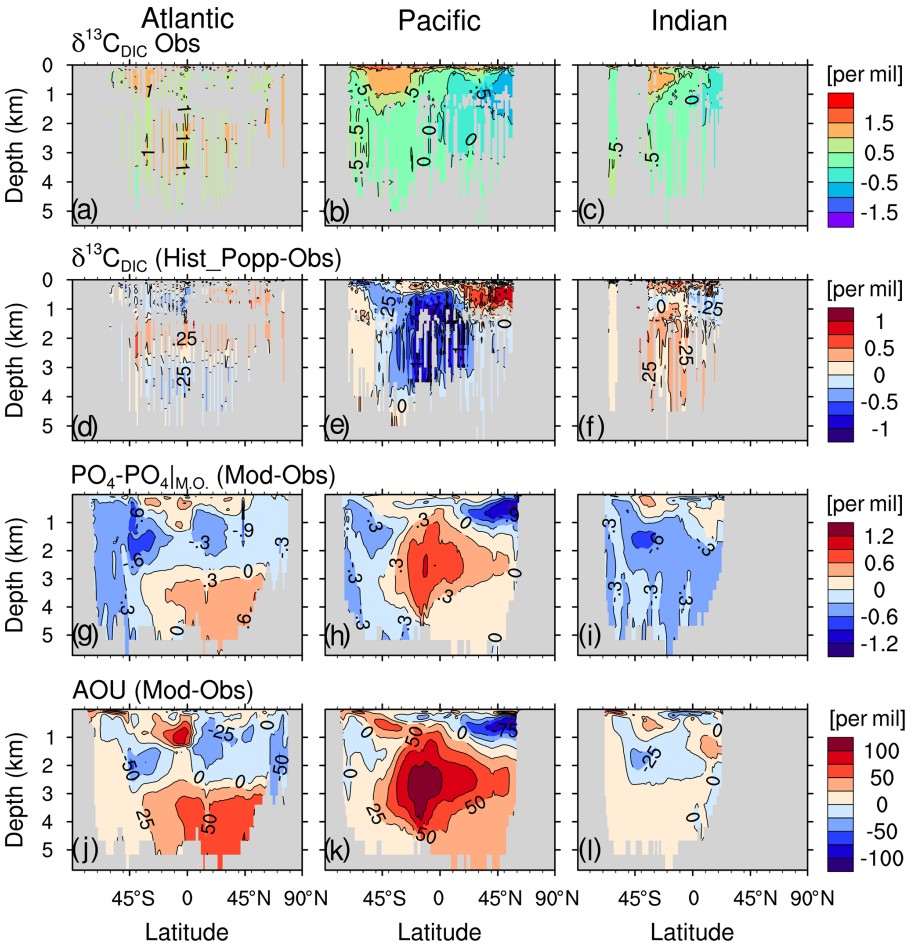

**Figure 10.** Zonal-mean distribution in the Atlantic Ocean (left column), the Pacific Ocean (middle column) and the Indian Ocean (right column) for the $\delta^{13}C_{DIC}$ observations from Schmittner et al. (2013) CE5 **(a–c)**, for the difference between Hist_Popp (sampled at the same location, year and month of the observations) and $\delta^{13}C_{DIC}$ measurement **(d–f)**, for the $(PO_4 - PO_4|_{M.O.})$ difference between model and WOA data (WOA13; Garcia et al., 2013a) **(g–i)**, and for the apparent oxygen utilisation (AOU) difference between model and WOA data (WOA13; Garcia et al., 2013b) **(j–l)**. Here the climatological annual mean values of $PO_4$ and AOU are used for both model and WOA data because seasonal variation is negligible in the interior ocean and WOA only provides monthly data above 500 m. TS9

sented in our model as well. Specifically, the zonal current at 1000 m depth (typical depth for the Equatorial Intermediate Current system) shows too little spatial variability and too low speeds of $\sim 0.2\,\mathrm{cm\,s^{-1}}$ (Fig. B6), compared to the observed alternating jets with a meridional scale of 1.5° and speeds of $\sim 5\,\mathrm{cm\,s^{-1}}$ (see Fig. 2 from Cravatte et al., 2012).

The performances of both Hist_Popp and Hist_Laws regarding $\delta^{13}C_{DIC}$ are comparable with the Norwegian Earth System Model version 2 (NorESM2, Tjiputra et al., 2020; comparing their Fig. 21), the Commonwealth Scientific and Industrial Research Organisation Mark 3L climate system model with the Carbon of the Ocean, Atmosphere and Land (CSIRO Mk3L-COAL), Pelagic Interactions Scheme for Carbon and Ecosystem Studies (PISCES) and LOch-Vecode-Ecbilt-CLio-agIsm Model (LOVECLIM) (see Table 2 and Figs. 3, S2 and S3 of Buchanan et al., 2019, and references therein), the Community Earth System Model (CESM,

Jahn et al., 2015; comparing their Figs. 5 and 6 to our Figs. 7 and 6, respectively), and the UVic Earth System Model (Schmittner et al., 2013). The latter two studies used the same $\delta^{13}C_{DIC}$ data set for model evaluation. Schmittner et al. (2013) reported a better performance ($r = 0.88$ and NRMSE = 0.5) than ours ($r = 0.81$ and NRMSE = 0.7 in Hist_Popp). One main reason is that the nutrient trapping problem in HAMOCC6 does not occur in the simulations of Schmittner et al. (2013).

## 4   Evaluation of the simulated oceanic $^{13}$C Suess effect

The oceanic $\delta^{13}$C measurements taken during the late 20th century already include a signal that originates from burning of isotopically light fossil fuel over the industrial period. The associated decrease in atmospheric $\delta^{13}$C (Fig. 2) affects

oceanic $\delta^{13}$C via air–sea gas exchange, leading to a general decrease in $\delta^{13}$C$_{\text{DIC}}$. The distribution of this $\delta^{13}$C$_{\text{DIC}}$ change, i.e. the oceanic $^{13}$C Suess effect, could serve as benchmark for ocean models to evaluate the uptake and re-distribution of the anthropogenic $CO_2$ emissions in the ocean.

The model is able to reproduce the size of the global oceanic anthropogenic $CO_2$ sink, though some local biases in the net air–sea $CO_2$ flux exist (Fig. 9d). The simulated sink by year 1994 is 99 Pg C, which compares well to the observation-based estimate of $118 \pm 19$ Pg C from Sabine et al. (2004) and to other model estimates (e.g. 94 Pg C in Tagliabue and Bopp, 2008). For a direct comparison to published studies, we calculate the oceanic $\delta^{13}$C Suess effect, $\delta^{13}$C$_{\text{SE}}$, as the difference between the 1990s-averaged $\delta^{13}$C$_{\text{DIC}}$ from Hist_Popp and the pre-industrial climatological (50-year mean) $\delta^{13}$C$_{\text{DIC}}$ from PI_Popp. $\delta^{13}$C$_{\text{SE}}$ calculated using the results of Hist_Laws and PI_Laws only shows a marginal difference (global mean $< 0.04\,‰$) and is therefore not presented.

The surface mean $\delta^{13}$C$_{\text{SE}}$ in this study is $-0.66\,‰$, similar to the model study of Schmittner et al. (2013) ($-0.67\,‰$) and to the estimate by Sonnerup et al. (2007) ($-0.76 \pm 0.12\,‰$), who used an observation-based approach. The strongest oceanic $^{13}$C Suess effect is found in the subtropical gyres in the model (Fig. 11a), where water masses have long residence times at the ocean surface and therefore receive a strong anthropogenic imprint (Quay et al., 2003). In the subtropical gyres, the simulated surface $\delta^{13}$C$_{\text{SE}}$ generally varies between $-0.8\,‰$ and $-1.1\,‰$, which compares well to the surface ocean $\delta^{13}$C decrease of $-0.9 \pm 0.1\,‰$ recorded by coral and sclerosponges (Wörheide, 1998; Böhm et al., 1996, 2000; Swart et al., 2002, 2010) and to the estimates of $-1.0 \pm 0.09\,‰$ extracted from GLODAPv2 (Olsen et al., 2016; Eide et al., 2017a).

Along the vertical sections A16, P19 and I8S9N, $\delta^{13}$C$_{\text{SE}}$ is mainly confined to the upper 1000 m depth in the subtropical gyres of the South Atlantic, the Pacific Ocean and the Indian Ocean (Fig. 12a–c). In the North Atlantic, $\delta^{13}$C$_{\text{SE}}$ penetrates deeper than the other ocean regions, due to the intensive ventilation related to the formation of NADW. The simulated $\delta^{13}$C$_{\text{SE}}$ distributions show similar features to those of CFC-12 (Fig. B8). This is because both the decrease in $\delta^{13}$C$_{\text{DIC}}$ and increase in CFC-12 in the ocean are predominantly caused by the uptake of atmospheric anthropogenic signals and the subsequent transport by ocean circulation. Since changes in $\delta^{13}$C$_{\text{DIC}}$ are also induced by changes in marine biological activity, we separate $\delta^{13}$C$_{\text{DIC}}$ into a component depicting changes due to the transport of the surface $^{13}$C signal, i.e. the "preformed" $\delta^{13}$C$_{\text{DIC}}$, and to a regenerated component $\delta^{13}$C$^{\text{reg}}$, following Sonnerup et al. (1999):

$$\delta^{13}\text{C}^{\text{pref}} = \frac{\delta^{13}\text{C}_{\text{DIC}} \cdot \text{DIC} - \text{AOU} \cdot \left(\frac{\text{C}}{-\text{O}_2}\right)_{\text{org}} \cdot \delta^{13}\text{C}_{\text{org}}}{\text{DIC} - \text{AOU} \cdot \left(\frac{\text{C}}{-\text{O}_2}\right)_{\text{org}}}. \quad (15)$$

The $\left(\frac{\text{C}}{-\text{O}_2}\right)_{\text{org}}$ ratio is $122 : 172$ in HAMOCC6, and we use the simulated $\delta^{13}$C$_{\text{POC}}$ for $\delta^{13}$C$_{\text{org}}$. Clearly, the change in the preformed component $\delta^{13}$C$_{\text{SE}}^{\text{pref}} = \delta^{13}$C$_{\text{1990s}}^{\text{pref}} - \delta^{13}$C$_{\text{PI}}^{\text{pref}}$ dominates $\delta^{13}$C$_{\text{SE}}$ (comparing Fig. 12a–c to Fig. 12d–f). A major difference between $\delta^{13}$C$_{\text{SE}}^{\text{pref}}$ and $\delta^{13}$C$_{\text{SE}}$ is that positive $\delta^{13}$C$_{\text{SE}}^{\text{pref}}$ is widely seen below 1000 m, particularly in the Pacific Ocean (Fig. 12e). These positive $\delta^{13}$C$_{\text{SE}}^{\text{pref}}$ values relate to changes in the regenerated component $\delta^{13}$C$^{\text{reg}}$ (see Appendix D).

## 5 Potential sources of uncertainties in an observation-based global oceanic $^{13}$C Suess effect estimate

Eide et al. (2017a) (hereafter E17) derived the first observation-based estimate of the global ocean $^{13}$C Suess effect since pre-industrial times. E17's approach uses the concept of the similarity between the oceanic uptake of the anthropogenically produced CFC-12 and isotopically light $CO_2$ (see details in Appendix E1). Due to method- and data-specific limitations E17 stated that they potentially underestimate the oceanic $^{13}$C Suess effect. However, based on observations alone it is not possible to gain insight into the spatial distribution of this uncertainty or into its origin.

Our model simulations, particularly PI_Popp and Hist_Popp, provide an opportunity to learn more about the source of this uncertainty because the oceanic $\delta^{13}$C in the late 20th century (Sect. 3), the oceanic anthropogenic $CO_2$ sink (Sect. 4) and the invasion of CFC-12 into the ocean (Fig. B8) are well represented. Moreover, our simulated $\delta^{13}$C$_{\text{SE}}$ qualitatively resembles the oceanic $^{13}$C Suess effect estimate of E17 (see comparison between Fig. 11b and E17's Fig. 7, as well as comparison between Fig. 12a–c and g–i).

Based on the similarity between the oceanic uptake of atmospheric CFC-12 and $\delta^{13}$CO$_2$ signal, E17 link the $^{13}$C Suess effect since 1940 (when CFC-12 becomes detectable in the ocean) to CFC-12 partial pressure (pCFC-12) with a proportionality factor. Under the assumption of a temporally constant regenerated fraction $\delta^{13}$C$^{\text{reg}}$, this proportionality factor is considered equivalent to the slope of a linear regression relationship between the preformed component $\delta^{13}$C$^{\text{pref}}$ and pCFC-12 at any time after 1940. Thus, this slope $a$ can be obtained by performing linear regression for field measurements of $\delta^{13}$C$^{\text{pref}}$ and pCFC-12. Multiplying $a$ and pCFC-12 data yields the $^{13}$C Suess effect since 1940, which is then scaled to the full industrial period by a constant factor $f_{\text{atm}}$ (Eq. E7) related to changes in the atmospheric $\delta^{13}$C signature:

$$\delta^{13}\text{C}_{\text{SE}(t-\text{PI})} = f_{\text{atm}} \cdot a \cdot \text{pCFC-12}_t. \quad (16)$$

Here $a$ is the regression slope for the linear relationship between $\delta^{13}$C$_t^{\text{pref}}$ and pCFC-12$_t$ (Eq. E5). The value of $a$ is determined for each ventilation region defined in E17 (i.e. the

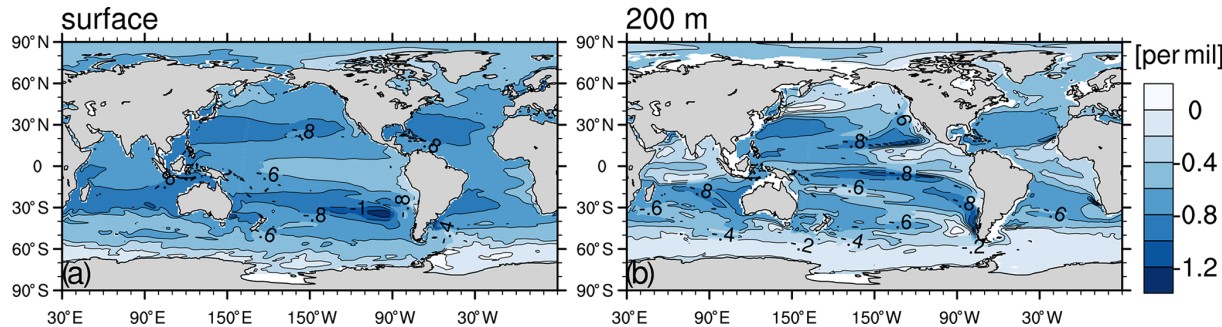

**Figure 11.** The simulated oceanic Suess effect $\delta^{13}C_{SE}$ from the pre-industrial period to the 1990s at sea surface **(a)** and at 200 m **(b)**.

Indian Ocean, North Pacific, South Pacific, North Atlantic and South Atlantic). Details of the E17 approach are given in Appendix E1.

By applying E17's approach to our model data that are sampled at the same geographical locations as observations used in E17, we obtain the regression slopes, hereafter referred to as $a_{\text{pref}}$, for each ventilation region. Taking year $t = 1994$ we obtain the estimated oceanic $^{13}$C Suess effect, SE$_{\text{pref}}$, for the period from the pre-industrial to 1994 following Eq. (16). The detailed calculation of SE$_{\text{pref}}$ is given in Appendix E2.

To quantify if SE$_{\text{pref}}$ under- or overestimates the oceanic $^{13}$C Suess effect, we compare SE$_{\text{pref}}$ to the simulated oceanic $^{13}$C Suess effect SE$_{\text{Mod}} = \delta^{13}C_{\text{DIC,1994}} - \delta^{13}C_{\text{DIC,PI}}$. Figure 13a presents (SE$_{\text{pref}}$ − SE$_{\text{Mod}}$) for 200 m depth. Positive values of (SE$_{\text{pref}}$ − SE$_{\text{Mod}}$) indicate underestimation of the oceanic $^{13}$C Suess effect.

At 200 m SE$_{\text{pref}}$ mostly underestimates SE$_{\text{Mod}}$ (Fig. 13a). The region-mean underestimation is 0.24‰ for the Indian Ocean, 0.21‰ for the North Pacific, 0.26‰ for the South Pacific, 0.1‰ for the North Atlantic and 0.14‰ for the South Atlantic (Table 1). Our model findings are very similar to the underestimation range discussed by E17. They determined an uncertainty range of 0.15‰ to 0.24‰ by comparing their global-mean estimate (−0.4‰ at 200 m depth) to an estimate (−0.55‰ to −0.64‰ at 200 m) which they deduced from previous model studies. Specifically, based on Broecker and Peng (1993) and Bacastow et al. (1996) E17 assumed an ocean-to-atmosphere ratio of the $^{13}$C Suess effect of 0.65 and the 200 m to surface ratio of the $^{13}$C Suess effect of 0.6–0.7. Multiplying the above two ratios with the atmospheric $\delta^{13}CO_2$ decrease of −1.4‰ by year 1994 yields the global-mean $^{13}$C Suess effect estimate of −0.55‰ to −0.64‰ at 200 m. In our model, the global-mean surface-ocean–atmosphere ratio of the $^{13}$C Suess effect is only 0.46, significantly lower than the five-box model of Broecker and Peng (1993). The 200 m to surface ratio of the $^{13}$C Suess effect is 0.75 in our model, and it is slightly higher than that of Bacastow et al. (1996), who employed an ocean general circulation model with coarse vertical resolution (four layers for the upper 200 m).

### 5.1 Source of underestimation attributed to data coverage

E17 have speculated that the major cause of the underestimation of the oceanic $^{13}$C Suess effect is that the available observations are mostly from the intermediate and deep waters. The ocean–atmosphere equilibration timescale for $\delta^{13}$C (10 years, Broecker and Peng, 1974) is significantly longer than that of pCFC-12 (1 month, Gammon et al., 1982). Thus, waters that have a shorter surface residence time, such as the deep waters ventilated in the South Hemisphere, would show less negative regression slope $a_{\text{pref}}$ (for the linear relationship between $\delta^{13}C^{\text{pref}}$ and pCFC-12, Eq. E5) than waters that have a longer surface residence time, e.g. subtropical gyres. In other words, $a_{\text{pref}}$ for the subtropical gyre water should be more negative than $a_{\text{pref}}$ for the entire corresponding ventilation region (the North Pacific, South Pacific, North Atlantic, South Atlantic or the Indian Ocean).

We test this potential explanation for the Indian Ocean and North Pacific. We are able to span regressional relationships for the subtropical gyres only because we have a larger data base. Specifically, we consider only model data at the geographical location of observations, but we use all model levels between 200 m and the pCFC-12 penetration depth (see Appendix E3). For the Indian Ocean, we combine model data from Subtropical Gyre Water (STGW) and Sub-Antarctic Mode Water (SAMW) as both water masses have a strong $^{13}$C Suess effect (Eide et al., 2017a). We find for this combined water mass (STGW) $a_{\text{pref}}$ (−0.65 × 10$^{-3}$, $r^2 = 0.49$) is more negative than that for the whole ventilation region (−0.47 × 10$^{-3}$, Fig. E3a). So indeed, with additional observations in the subtropical gyre we would receive a stronger $^{13}$C Suess effect estimate for the Indian Ocean. However, this difference in $a_{\text{pref}}$ only corresponds to an underestimation of about 0.12‰ at 200 m for the Indian subtropical region (see calculation in Appendix E3), which does not explain the total underestimation of 0.24‰ in the Indian Ocean (Table 1). In the North Pacific $a_{\text{pref}}$ for the Subtropical Gyre Water (−0.44 × 10$^{-3}$, $r^2 = 0.26$) is even less negative than that for the whole ventilation region (−0.71 × 10$^{-3}$) in the model, which is in contrast to the conjecture of E17.

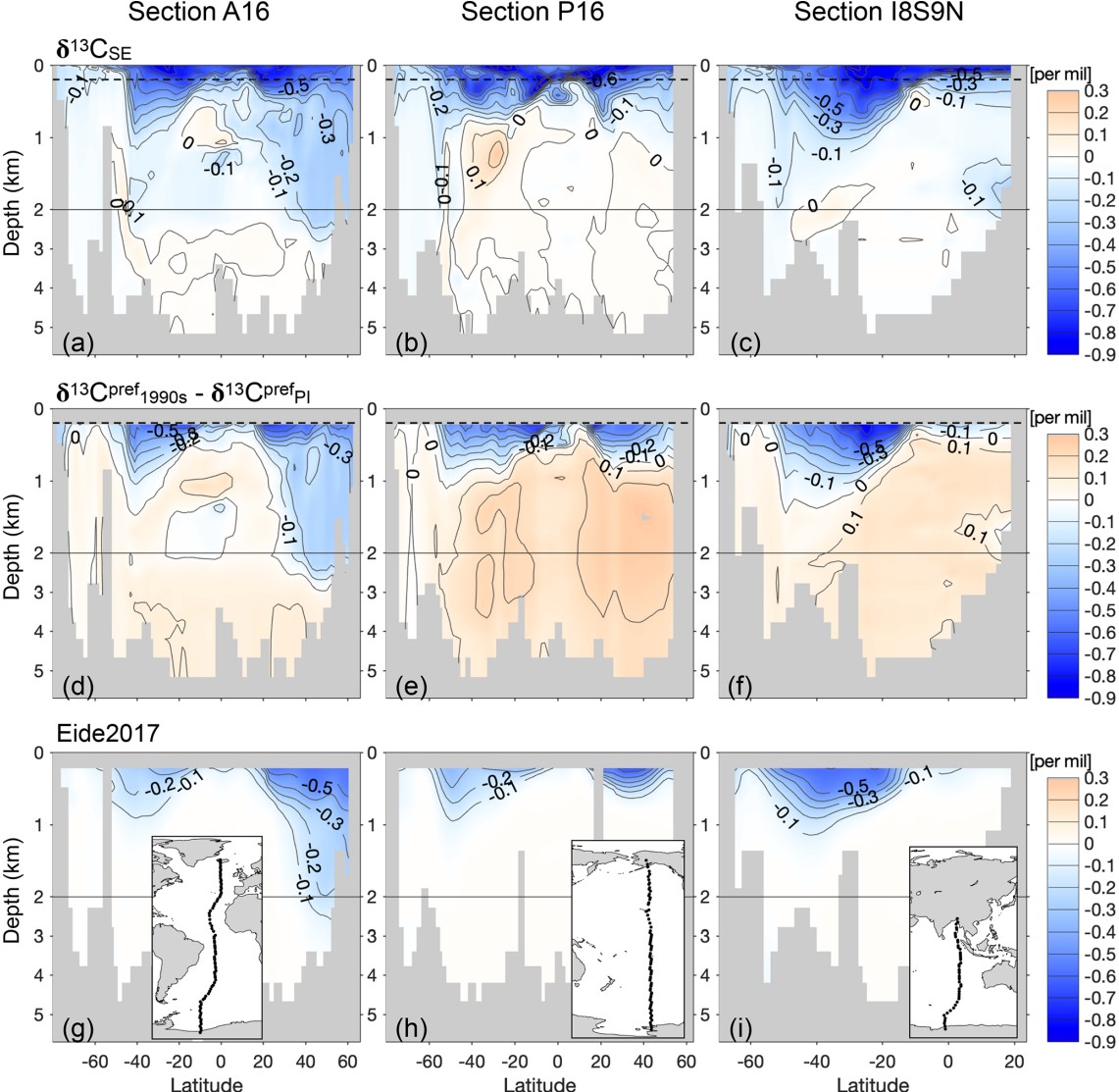

**Figure 12.** The simulated oceanic Suess effect $\delta^{13}C_{SE}$ since pre-industrial times for vertical sections A16 in the Atlantic Ocean **(a)**, P16 in the Pacific Ocean **(b)** and I8S9N in the Indian Ocean **(c)**. Panels **(d)**–**(f)** and **(g)**–**(i)** are as panels **(a)**–**(c)** but for the change in the preformed component $\delta^{13}C_{SE}^{pref} = \delta^{13}C_{1990s}^{pref} - \delta^{13}C_{PI}^{pref}$ and for the observation-based estimate of the oceanic Suess effect from Eide et al. (2017a), respectively. Inserted maps show the location of the vertical sections. The horizontal dashed black lines in panels **(a)**–**(f)** indicate 200 m depth, below which the Eide et al. (2017a) estimate is available. Note the bathymetry is different between the model and Eide et al. (2017a). TS10

## 5.2 Source of underestimation attributed to assumptions of E17's approach

A potential under-representation of data from subtropical gyres does not fully explain the underestimation of the $^{13}$C Suess effect found in our model. Instead, we argue that the source of uncertainty mainly relates to different assumptions that have been made in the E17 approach. Specifically, in the expression of the preformed component $\delta^{13}C_{1994}^{pref}$ (following Eq. E3),

$$\delta^{13}C_{1994}^{pref} = \delta^{13}C_{SE(1994-1940)} + \delta^{13}C_{1940}^{pref} \\ - (\delta^{13}C_{1994}^{reg} - \delta^{13}C_{1940}^{reg}), \qquad (17)$$

E17 assume that the regenerated component is constant in time, i.e. $-(\delta^{13}C_{1994}^{reg} - \delta^{13}C_{1940}^{reg}) = 0$. Consequently, Eq. (17) is reduced to

$$\delta^{13}C_{1994}^{pref} = \delta^{13}C_{SE(1994-1940)} + \delta^{13}C_{1940}^{pref}. \qquad (18)$$

Furthermore, they assume that the regression slope $a_{pref}$ for $\delta^{13}C_{1994}^{pref}$ and pCFC-12$_{1994}$ is equivalent to the regression

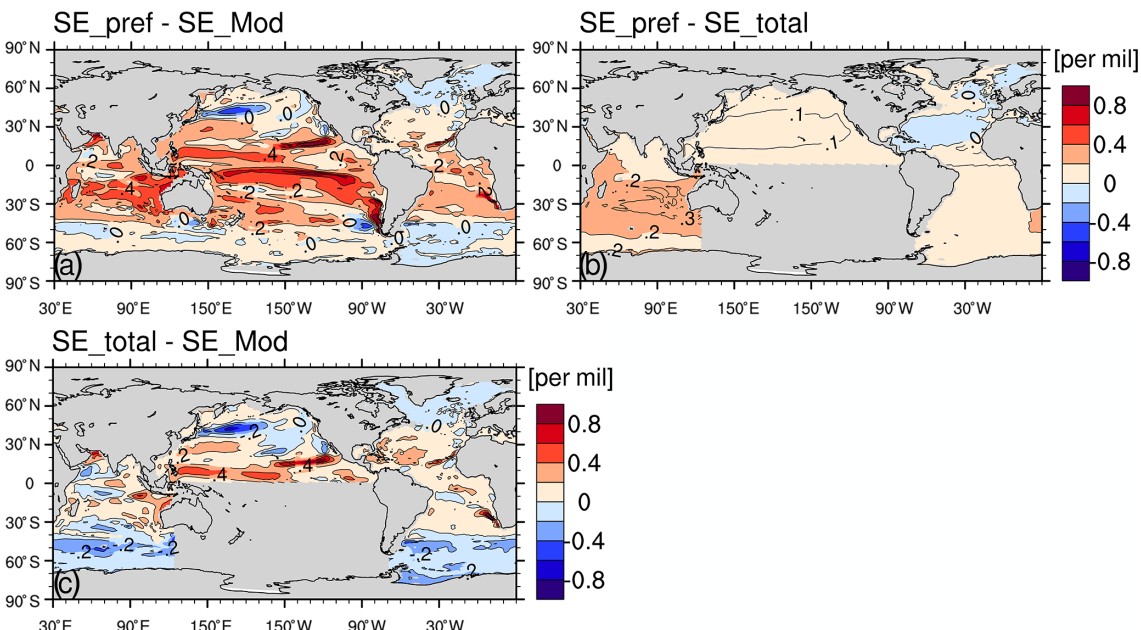

**Figure 13.** Distribution at 200 m depth for $SE_{pref} - SE_{Mod}$ **(a)**, $SE_{pref} - SE_{total}$ **(b)** and $SE_{total} - SE_{Mod}$ **(c)** CE7. The isoline increment is 0.2‰. In panels **(b)** and **(c)**, the South Pacific Ocean is not presented because the relationship between the total oceanic $^{13}$C Suess effect $\delta^{13}C_{SE(1994-1940)}$ and pCFC-12$_{1994}$ is too weak ($r^2 = 0.07$), and therefore $SE_{total}$ can not be estimated (see Appendix E4).

**Table 1.** Region-mean $(SE_{pref} - SE_{Mod})$, $(SE_{pref} - SE_{total})$ and $(SE_{total} - SE_{Mod})$ for five ventilation regions defined by E17, i.e. the Indian Ocean, North Pacific, South Pacific, North Atlantic and South Atlantic. The unit is per mil. $(SE_{pref} - SE_{total})$ is further decomposed into the two contributions $f_{atm} \cdot (a_{pref} - a_{total}) \cdot pCFC\text{-}12$ and $-f_{atm} \cdot b_{total}$ according to Eq. (20). TS11

|  | $(SE_{pref} - SE_{Mod})$ | $(SE_{pref} - SE_{total})$ | $f_{atm} \cdot (a_{pref} - a_{total}) \cdot pCFC\text{-}12$ $-f_{atm} \cdot b_{total}$ | $(SE_{total} - SE_{Mod})$ |
|---|---|---|---|---|
| Indian Ocean | 0.24 | 0.23 | 0.12 <br> 0.11 | 0.01 |
| North Pacific | 0.21 | 0.09 | 0.06 <br> 0.03 | 0.13 |
| South Pacific | 0.26 | \ | \ <br> \ | \ |
| North Atlantic | 0.1 | 0.02 | −0.1 <br> 0.12 | 0.09 |
| South Atlantic | 0.14 | 0.15 | 0.04 <br> 0.11 | −0.01 |

slope for the total $^{13}$C Suess effect $\delta^{13}C_{SE(1994-1940)}$ and pCFC-12$_{1994}$ (see Eqs. E1, E4 and E5). This implies that the preformed component $\delta^{13}C^{pref}_{1940}$ of 1940 has to be spatially uniform.

However, we find a specific vertical structure in the simulated $\delta^{13}C^{pref}_{1940}$ (Fig. 14a–c). Over large regions of the ocean, $\delta^{13}C^{pref}_{1940}$ generally decreases with increasing depth. This vertical distribution of $\delta^{13}C^{pref}$ is already present in pre-industrial times. High surface $\delta^{13}C_{DIC}$ caused by biological fractionation is transported into the ocean interior. Therefore, the preformed component generally decreases with increasing water depth. From pre-industrial times to 1940, the decrease in the atmospheric $^{13}$C/$^{12}$C ratio is relatively small (0.4‰, Fig. 2a), and therefore also the impact on the oceanic $\delta^{13}C_{DIC}$ is small. Thus, $\delta^{13}C^{pref}_{1940}$ has the similar vertical structure as that of the pre-industrial ocean.

Both the total $\delta^{13}C_{SE(1994-1940)}$ (mostly negative, similar to the distribution of $\delta^{13}C_{SE(1990s-PI)}$ TS13 in Fig. 12a–c) and pCFC-12 (Fig. B8a–c) show larger absolute values at the sur-

https://doi.org/10.5194/bg-18-1-2021      **Biogeosciences, 18, 1–41, 2021**

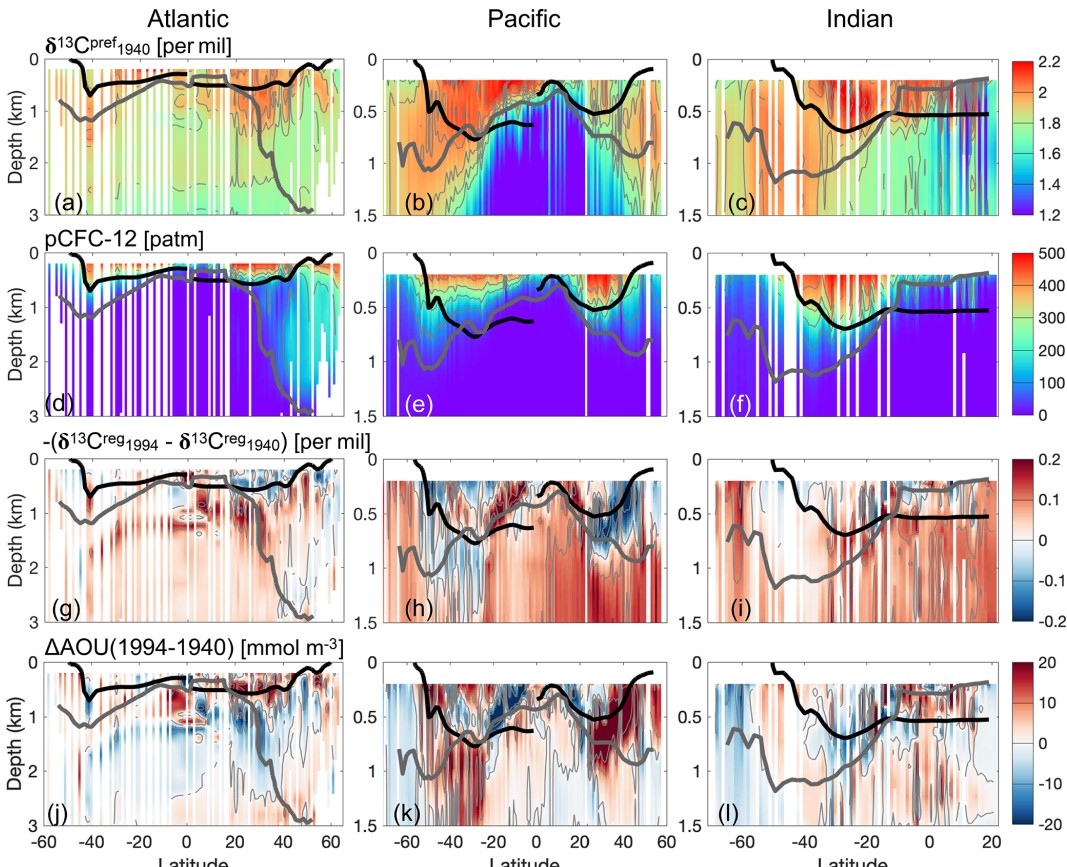

**Figure 14. (a–c)** The zonal mean of the simulated $\delta^{13}C^{\text{pref}}_{1940}$ for the locations where both observed CFC-12 and $\delta^{13}C_{\text{DIC}}$ are available. The thick grey line is the pCFC-12$_{1994}$ = 20 patm isoline, above which model data are used to perform linear regression. The thick black lines outline the Subtropical Gyre Water in the Atlantic and North Pacific Ocean, as well as the Subtropical Gyre Water and Sub-Antarctic Mode Water in the Indian Ocean and South Pacific Ocean (definition of water masses in Table E1). Panels **(d)–(f)**, **(g)–(i)** and **(j)–(l)** are as panels **(a)–(c)** but for pCFC-12$_{1994}$, for $-(\delta^{13}C^{\text{reg}}_{1994} - \delta^{13}C^{\text{reg}}_{1940})$ and for AOU changes between year 1940 and 1994, respectively. Note that for the Atlantic Ocean the upper 3 km is shown, whereas for the Pacific and Indian Ocean the upper 1.5 km is presented. TS12

face than in the interior ocean. As $\delta^{13}C^{\text{pref}}_{1940}$ is more positive in the upper ocean than the deep ocean, $\delta^{13}C^{\text{pref}}_{1994}$ has a smaller vertical gradient than $\delta^{13}C_{\text{SE(1994–1940)}}$ (see Eq. 18). Thus, a linear regression for $\delta^{13}C^{\text{pref}}_{1994}$ and pCFC-12 results in a less negative slope than a slope obtained with a spatially uniform $\delta^{13}C^{\text{pref}}_{1940}$, which implicates a contribution to an underestimation of the oceanic $^{13}$C Suess effect.

We also find that $-(\delta^{13}C^{\text{reg}}_{1994} - \delta^{13}C^{\text{reg}}_{1940})$ is non-zero, and it shows considerable spatial variability (Fig. 14g–i). Most prominently, in the North Atlantic $-(\delta^{13}C^{\text{reg}}_{1994} - \delta^{13}C^{\text{reg}}_{1940})$ is mostly negative above 500 m, and it is mostly positive below 500 m. This vertical structure of $-(\delta^{13}C^{\text{reg}}_{1994} - \delta^{13}C^{\text{reg}}_{1940})$ in the North Atlantic leads to stronger vertical gradient in $\delta^{13}C^{\text{pref}}_{1994}$ and therefore a more negative regression slope than that obtained with $-(\delta^{13}C^{\text{reg}}_{1994} - \delta^{13}C^{\text{reg}}_{1940}) = 0$. This implies the overestimation of the $^{13}$C Suess effect in the North Atlantic.

To evaluate the impact of assuming a spatially uniform $\delta^{13}C^{\text{pref}}_{1940}$ and $-(\delta^{13}C^{\text{reg}}_{1994} - \delta^{13}C^{\text{reg}}_{1940}) = 0$, we calculate an estimated $^{13}$C Suess effect from pre-industrial times to 1994, SE$_{\text{total}}$, based on a linear regression for the simulated total oceanic $^{13}$C Suess effect $\delta^{13}C_{\text{SE(1994–1940)}}$ and pCFC-12:

$$\text{SE}_{\text{total}} = f_{\text{atm}} \cdot (a_{\text{total}} \cdot \text{pCFC-12}_{1994} + b_{\text{total}}). \quad (19)$$

Here $a_{\text{total}}$ and $b_{\text{total}}$ are regression coefficients for $\delta^{13}C_{\text{SE(1994–1940)}}$ and pCFC-12 (more details in Appendix E4). With Eqs. (16) and (19) we get

$$\begin{aligned}\text{SE}_{\text{pref}} - \text{SE}_{\text{total}} &= f_{\text{atm}} \cdot (a_{\text{pref}} - a_{\text{total}}) \\ &\quad \cdot \text{pCFC-12}_{1994} - f_{\text{atm}} \cdot b_{\text{total}}.\end{aligned} \quad (20)$$

Comparison between the regressional slope $a_{\text{pref}}$ (obtained for $\delta^{13}C^{\text{pref}}_{1994}$ and pCFC-12) and $a_{\text{total}}$ facilitates the quantification of the under- or overestimation of the $^{13}$C Suess effect linked to the above two assumptions.

In the Indian Ocean $a_{\text{pref}} = -0.47 \times 10^{-3}$ (Fig. E2a) is less negative than $a_{\text{total}} = -0.74 \times 10^{-3}$ (Fig. E3a). This results

in an underestimation of 0.12 ‰ according to Eq. (20). Similarly, for the North Pacific $a_{pref} = -0.71 \times 10^{-3}$ (Fig. E2b) is less negative than $a_{total} = -0.83 \times 10^{-3}$ (Fig. E3b), which leads to an underestimation of 0.06 ‰. For the South Atlantic $a_{pref} = -0.6 \times 10^{-3}$ (Fig. E2e) and $a_{total} = -0.7 \times 10^{-3}$ (Fig. E3e), which yields an underestimation of 0.04 ‰. Such underestimation is mainly due to the decreasing $\delta^{13}C_{1940}^{pref}$ with increasing depth in these regions.

Different from these three ventilation regions, in the North Atlantic $a_{pref} = -0.81 \times 10^{-3}$ (Fig. E2d) is more negative than $a_{total} = -0.62 \times 10^{-3}$ (Fig. E3d). This is due to the specific vertical structure of $-(\delta^{13}C_{1994}^{reg} - \delta^{13}C_{1940}^{reg})$ as previously discussed.

Another major difference between $SE_{pref}$ and $SE_{total}$ is the non-negligible negative intercept $b_{total}$ (Eq. 20). This reveals the underestimation of $SE_{pref}$ related to E17's assumption that the $^{13}$C Suess effect is directly proportional to pCFC-12. The intercept $b_{total}$ emerges possibly due to the different atmospheric time history of the $^{13}$C Suess effect compared to CFC-12, as is discussed by E17 for the deep ocean with very low or zero CFC-12. The decreasing of $\delta^{13}C_{POC}$ under increasing surface $CO_2$(aq) (Appendix D) also contributes to an non-negligible $b_{total}$ as lower $\delta^{13}C_{POC}$ leads to lower $\delta^{13}C_{DIC}$ in the ocean interior. In the South Atlantic and Indian Ocean, $b_{total} = -0.07$ ‰ corresponds to an underestimation of 0.12 and 0.11 ‰ (Table 1), respectively.

Table 1 summaries of the contributions from ($SE_{pref} - SE_{total}$) for different ventilation regions. The comparison to the total underestimation given by ($SE_{pref} - SE_{Mod}$) shows that this underestimation, which is attributed to the assumption of E17's approach, is the largest contributor for the Indian Ocean and the South Atlantic.

The residual under-/overestimation of $SE_{pref}$ given by ($SE_{total} - SE_{Mod}$) = ($SE_{pref} - SE_{Mod}$) − ($SE_{pref} - SE_{total}$) shows how well a method based on linear regression relationships between $\delta^{13}C_{SE}$ and pCFC-12$_{1994}$ can estimate the global ocean Suess effect. ($SE_{total} - SE_{Mod}$) at 200 m generally show positive values, i.e. underestimation, in low latitudes (between 40° S and 40° N), and it is rather negative poleward of 40° (Fig. 13c). This pattern results from pooling data from different water masses to generate one regression relationship for a large ventilation region. The waters ventilated in lower latitudes typically have a stronger $^{13}$C Suess effect than those ventilated in high latitudes. This is clearly reflected in the linear regression relationships between $\delta^{13}C_{SE(1994-1940)}$ and pCFC-12$_{1994}$ for the North Atlantic (Fig. E3d), which shows that the regression slope $a_{total}$ for the Subtropical Gyre Water is noticeably steeper than that of the deep waters. Accordingly in the interior ocean, the water masses ventilated in the low latitudes generally show an underestimation of the $^{13}$C Suess effect (positive values of $SE_{total} - SE_{Mod}$), and the water masses ventilated in the high latitudes show an overestimation (Fig. E1g–i). In the North Atlantic Ocean, the region-mean underestimation ($SE_{pref} - SE_{Mod}$) = 0.1 ‰ is predominantly contributed by ($SE_{total} - SE_{Mod}$) = 0.09 ‰. In the North Pacific Ocean ($SE_{total} - SE_{Mod}$) = 0.13 ‰ accounts for more than half of the total underestimation 0.21 ‰. In the Indian and South Atlantic Ocean, however, ($SE_{total} - SE_{Mod}$) has hardly any influence to the region-mean underestimation.

In summary, our analysis points out two major causes for the underestimation of $^{13}$C in E17's approach. The first is the assumption of a spatially uniform preformed $\delta^{13}$C component in 1940. The second cause is the neglect of processes not directly linked to the oceanic uptake and transport of CFC-12, e.g. the uptake of anthropogenically light $CO_2$ in the times prior to the emission of CFC-12 and the decrease in $\delta^{13}C_{DIC}$ due to the decrease in $\delta^{13}C_{POC}$ over the industrial period.

## 6 Summary and conclusions

We present results of the new $^{13}$C module in the ocean biogeochemical model HAMOCC6 for the historical period forced by reanalyses data (ERA20C). We test two parameterisations of different complexity for the biological fractionation factor: $\epsilon_p^{Popp}$ depends on dissolved $CO_2$ (Popp et al., 1989); $\epsilon_p^{Laws}$ is a function of dissolved $CO_2$ and phytoplankton growth rate (Laws et al., 1995). Furthermore, we use our consistent model framework to assess the approach by Eide et al. (2017a), which yields the first global oceanic $^{13}$C Suess effect estimate based on a correlation between preformed $\delta^{13}C_{DIC}$ and CFC-12 partial pressure.

The comparison between simulated and observed isotopic ratio of organic matter $\delta^{13}C_{POC}$ reveals that $\epsilon_p^{Popp}$ ($r = 0.84$ and NRMSE = 0.57) has a better performance than $\epsilon_p^{Laws}$ ($r = 0.71$ and NRMSE = 2.5). Using $\epsilon_p^{Laws}$ results in noticeably lower $\delta^{13}C_{POC}$ values and smaller $\delta^{13}C_{POC}$ gradients between low and high latitudes compared to observations. The parameterisation of Laws et al. (1995), obtained based on cultures of marine diatom *Phaeodactylum tricornutum*, results in too strong a preference of isotopically light carbon in our global ocean biogeochemical model.

Regarding $\delta^{13}C_{DIC}$, $\epsilon_p^{Popp}$ also yields slightly better agreement with observations than $\epsilon_p^{laws}$ ($r = 0.81$ and NRMSE = 0.7 versus $r = 0.80$ and NRMSE = 1.1), because $\epsilon_p^{Laws}$ produces lower $\delta^{13}C_{POC}$ and therefore lower interior-ocean $\delta^{13}C_{DIC}$ than those found in observations. $\epsilon_p^{Popp}$ performs well considering the uncertainties in observed $\delta^{13}C_{DIC}$ (0.1 ‰–0.2 ‰; Schmittner et al., 2013). Our model slightly overestimates surface $\delta^{13}C_{DIC}$. By decomposing $\delta^{13}C_{DIC}$ into a biological component and a residual component, we find the overestimation in the high-latitude ocean is dominated by biases in the biological component caused by e.g. surface iron concentration that is too high. In the interior-ocean $\delta^{13}C_{DIC}$ biases are mainly due to biases in the physical

state (for instance, a boundary that is too shallow between the NADW cell and the Antarctic Bottom Water cell in MPIOM).

Our model represents well the temporal evolution of the oceanic $\delta^{13}C_{DIC}$ since pre-industrial times, i.e. the oceanic $^{13}$C Suess effect due to the intrusion of isotopically light carbon into the ocean. With the complete information on the spatial and temporal $^{13}$C evolution in the ocean, together with the simulated evolution of CFC-12, we identify the sources for the potential uncertainties in the framework of Eide et al. (2017a) for deriving an observation-based oceanic $^{13}$C Suess effect. Based on our model, we find underestimations of the $^{13}$C Suess effect at 200 m by 0.24‰ in the Indian Ocean, 0.21‰ in the North Pacific Ocean, 0.26‰ in the South Pacific Ocean, 0.1‰ in the North Atlantic Ocean and 0.14‰ in the South Atlantic Ocean. These numbers are in line with the underestimation range 0.15‰ to 0.24‰ conjectured by Eide et al. (2017a). They speculated this underestimation is due to the under-representation of the water masses with a stronger $^{13}$C Suess effect, such as the Subtropical Gyre Water and Sub-Antarctic Mode Water, in the observational data. Our analysis shows that their hypothesis only explain half of the underestimation in the Indian Ocean. For the North Atlantic Ocean this hypothesis is not supported by the model data . We identify two major causes for the underestimation of the $^{13}$C Suess effect by the applied method. The first relates to the assumption of a spatially uniform preformed component of $\delta^{13}C_{DIC}$ in year 1940. In our model this preformed component is generally more positive in the upper ocean than in the interior ocean, which contributes to the underestimation of the $\delta^{13}$C Suess effect. The second cause relates to the neglect of processes that are not directly linked to the oceanic uptake and transport of CFC-12 – for instance, the $^{13}$C Suess effect prior to the emission of CFC-12 and the decrease in $\delta^{13}C_{POC}$ over the industrial period.

We conclude that the new $^{13}$C module with biological fractionation factor $\epsilon_p^{Popp}$ from Popp et al. (1989) has a satisfactory performance. We are aware that the parameterisation $\epsilon_p^{Popp}$ omits any potential changes, e.g. in ecosystem structure, which might have occurred in the paleo-ocean. Our new $^{13}$C module will serve as a useful tool to evaluate the performance of MPI-ESM in paleoclimate and to investigate the past changes in the ocean, for instance within the ongoing research project PalMod (Latif et al., 2016).

## Appendix A: Governing factors for the water-column DI$^{13}$C inventory changes

The water-column DI$^{13}$C inventory difference is primarily a result of the difference in the net air–sea $^{13}$CO$_2$ flux between PI_Popp and PI_Laws. This is demonstrated by the comparison of the contributions of the governing factors for the water-column DI$^{13}$C inventory changes (Table A1), including air–sea gas exchange, loss of POC and CaCO$_3$ to marine sediment, diffusion of the remineralised DIC from sediment into the water column, input of DOC and CO$_3^{2-}$, and the exchange with other marine carbon pools (phytoplankton, CaCO$_3$, etc.). Table A1 also reveals that the current method to determine the $^{13}$C input (see Sect. 2.3.2) only has a small contribution to the change in the water-column DI$^{13}$C inventory.

**Table A1.** Contributions to the rate of the water-column DI$^{13}$C inventory change (in Gmol yr$^{-1}$), averaged in the last 50 years in the corresponding pre-industrial spin-up simulations. Positive values denote contributions to the increase in the water-column DI$^{13}$C inventory. The last column gives the relative contribution to the total rate difference with relative contribution = (PI_Laws-PI_Popp) / total rate difference.

| $^{13}$C fluxes into the water column (Gmol yr$^{-1}$) | PI_Popp | | PI_Laws | | PI_Laws – PI_Popp | | Relative contribution |
|---|---|---|---|---|---|---|---|
| Air–sea gas exchange | 1824.4 | | 1552.3 | | −272.1 | | 1.1 |
| POC loss to sediment | −34902.9 | | −34626.4 | | 276.5 | | |
| CaCO$_3$ loss to sediment | −16672.1 | | −16674.3 | | −2.2 | | |
| DOC input | 13612.7 | sum: 596.1 | 13506.8 | sum: 626.6 | −105.9 | sum: 30.5 | −0.1 |
| CO$_3^{2-}$ input | 16505.2 | | 16506.9 | | 1.7 | | |
| Sediment DIC reflux | 22053.2 | | 21913.6 | | −139.6 | | |
| From other water-column carbon pools | 63.8 | | 64.2 | | −0.4 | | 0.001 |
| Total rate | 2484.7 | | 2242.7 | | −242.0 | | 1 |

## Appendix B: Model–observation comparison of ocean physics

Sea surface temperature (SST) and salinity (SSS) generally show good performance (Fig. B1 and Table B1). The most
5 striking bias is seen for SSS (2–3 psu) in the Arctic Ocean. In the ocean interior, the performance of temperature and salinity is similar to other ocean general circulation models, e.g. Tjiputra et al. (2020) (comparing our Table B1 to their Fig. 2). The model biases shown here (for instance, the sur-
10 face layers are too cold, whereas the water between 500 and 2500 m is too warm and saline; see Fig. B2) are typically seen in MPIOM; see Jungclaus et al. (2013) for a detailed discussion.

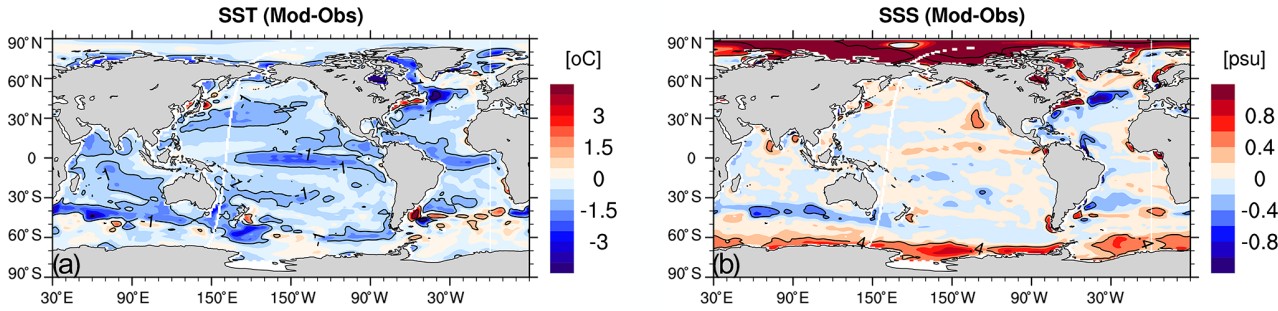

**Figure B1.** Biases in sea surface temperature (SST, panel **a**) and salinity (SSS, panel **b**). Both model and observational data (EN4 version 4.2.0; Good et al., 2013) are averaged for 1960–1999.

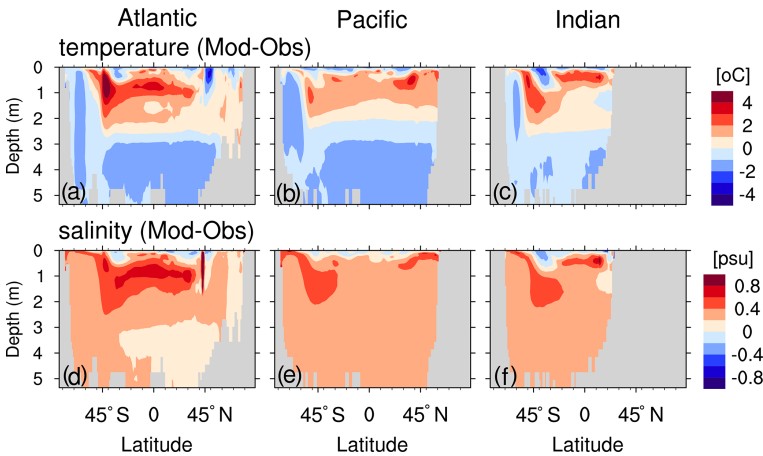

**Figure B2.** Zonal-mean biases of seawater temperature **(a–c)** and salinity **(d–f)** with respect to observations (EN4 version 4.2.0; Good et al., 2013) for the Atlantic Ocean **(a, d)**, Pacific Ocean **(b, e)** and Indian Ocean **(c, f)**. TS14

**Table B1.** Summary of the spatial correlation coefficient $r$ and normalised root mean square error (NRMSE) between model data and observations from EN4 (version 4.2.0; Good et al., 2013).

| Depth (km) | Temperature | | Salinity | |
|---|---|---|---|---|
| | $r$ | NRMSE | $r$ | NRMSE |
| 0 | 0.997 | 0.099 | 0.95 | 0.41 |
| 0.5 | 0.90 | 0.58 | 0.88 | 0.43 |
| 1 | 0.87 | 0.89 | 0.83 | 0.70 |
| 3 | 0.91 | 1.09 | 0.92 | 1.62 |

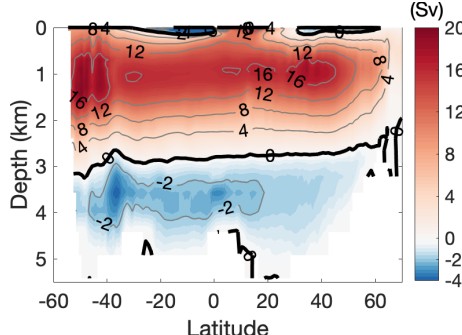

**Figure B3.** Atlantic Meridional Overturning Circulation (AMOC) stream function (Sv).

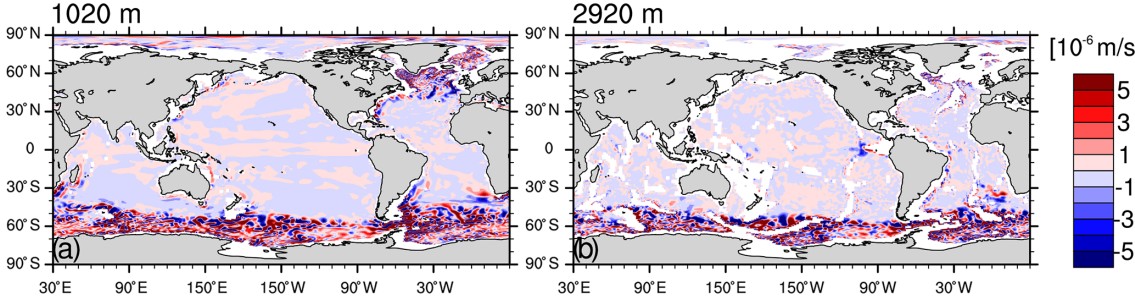

**Figure B4.** 1990–2009 mean vertical velocity (m s$^{-1}$) in the model at 1020 m depth **(a)** and 2920 m depth **(b)**.

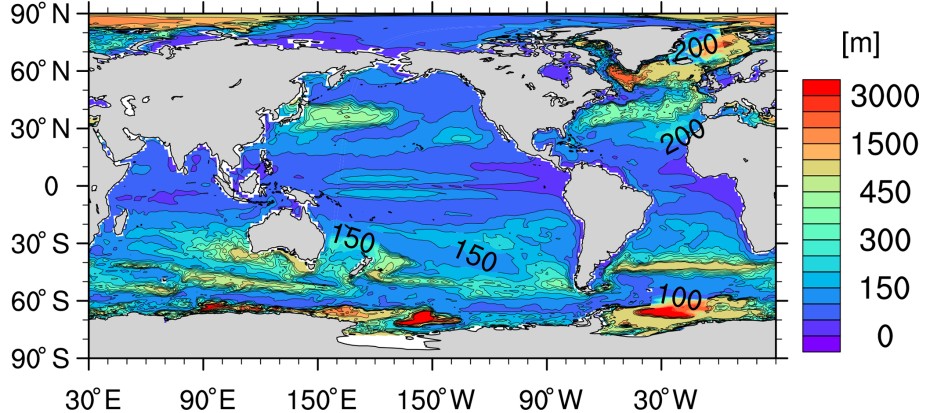

**Figure B5.** The mean of the annual maximum of the monthly mixed layer depth (m) for the period 1970–1999 in the model. The mixed layer depth is defined as the depth at which a $0.03\,\mathrm{kg\,m^{-3}}$ change in potential density with respect to the surface has occurred. Contour intervals are 50 for 0–500 and 500 for 500–3000.

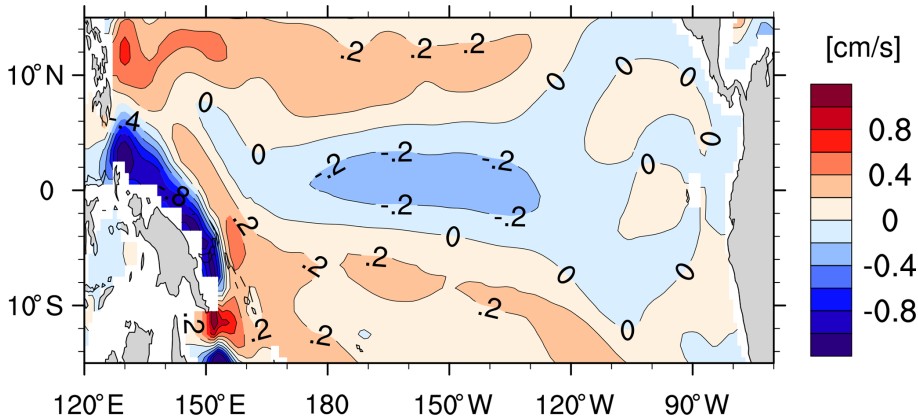

**Figure B6.** The simulated zonal current ($\mathrm{cm\,s^{-1}}$) at 960 m depth in the equatorial Pacific (averaged over January 2003–August 2009). Positive values indicate eastward flow.

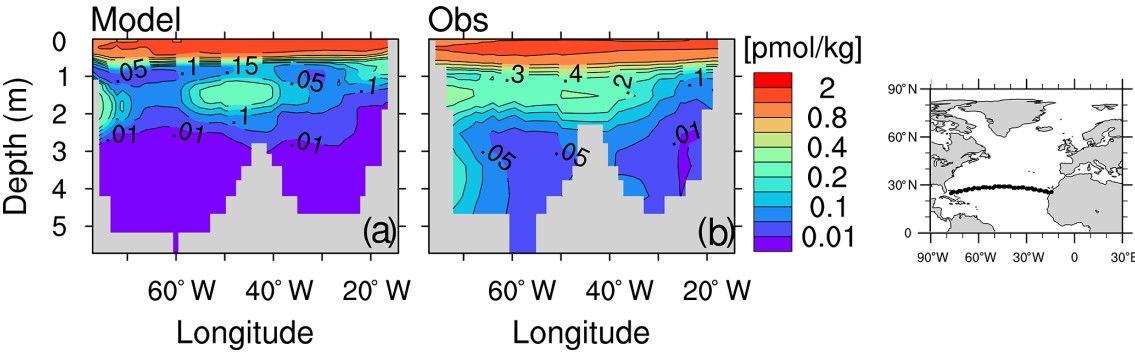

**Figure B7.** CFC-12 concentration ($\mathrm{pmol\,kg^{-1}}$) in February 1998 along the A5 section in the Atlantic Ocean (see right panel) of the model **(a)** and of observations from GLODAPv1 database (panel **b**; Key et al., 2004). Contour intervals are 0.01, 0.05, 0.1, 0.15, 0.2, 0.3, 0.4, 0.6, 0.8, 1.2 and 2 pmol kg$^{-1}$. TS15

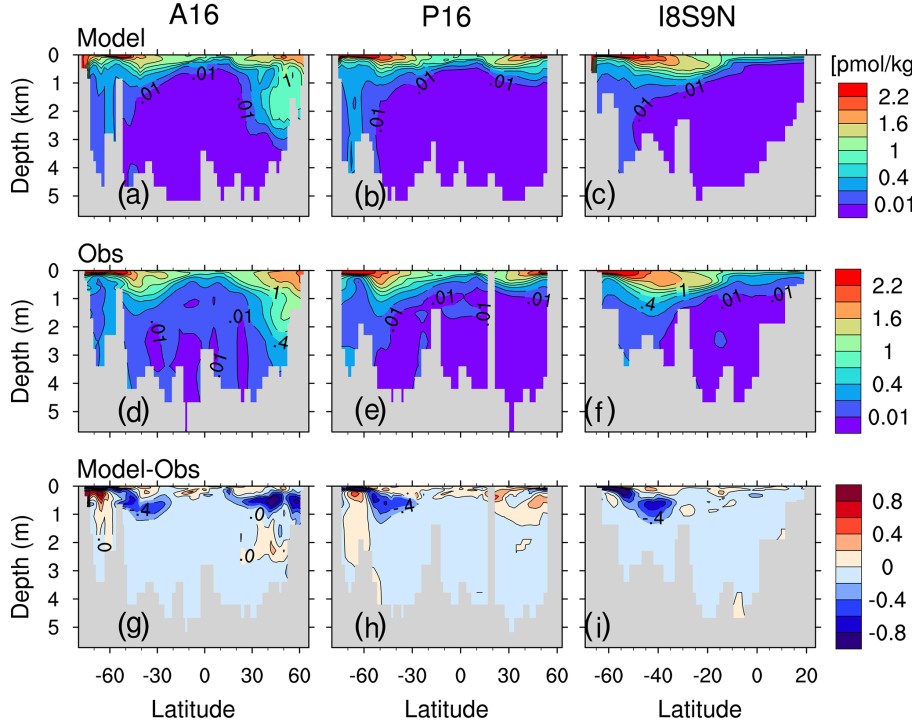

**Figure B8.** (a–c) CFC-12 concentration (pmol kg $^{-1}$) for the section A16 (a), P16 (b) and I8S9N (c). Panels (d)–(f) and (g)–(i) are as panels (a)–(c) but for the observed CFC-12 (GLODAPv1; Key et al., 2004) and for the difference between model and observation, respectively. The isolines in panels (a)–(f) are 0.01, 0.1, 0.4, 0.7, 1.0, 1.3, 1.6, 1.9 and 2.2 pmol kg $^{-1}$. The isoline increment in panels (g)–(i) is 0.2 pmol kg $^{-1}$. TS16

## Appendix C: Model–observation comparison of ocean biogeochemistry

### C1 Net primary production, growth rate, biomass and limiting nutrients

The simulated net primary production, 48.7 Gt yr$^{-1}$ for bulk phytoplankton and 3 Gt yr$^{-1}$ for cyanobacteria, compares well with the satellite-based estimate of $\sim$ 52 Gt yr$^{-1}$ (Westberry et al., 2008; Silsbe et al., 2016).

The simulated growth rate $\mu$ (Fig. C1a and b, only shown for bulk phytoplankton because cyanobacteria has a much lower primary production) is broadly consistent with the large-scale patterns of the satellite-based $\mu$ estimates from Westberry et al. (2008) (Figs. C1c and C1d) and with field observations. In the central equatorial Pacific the simulated $\mu$ well reproduces the observed range (0.55–0.7 d$^{-1}$, Chavez et al., 1996; note the satellite-based estimates overestimate $\mu$ due to excluding iron limitation). In the subtropical gyres, the simulated $\mu$ (annual-mean 0.1–0.25 d$^{-1}$) is at the lower side of both the observations (annual mean 0.3–0.53 d$^{-1}$ in the North Pacific subtropical gyre, Letelier et al., 1996; annual mean 0.13–0.62 d$^{-1}$ in the North Atlantic subtropical gyre, Marañón, 2005) and the satellite-based $\mu$ estimates. In the Pacific sector of the Southern Ocean, the simulated $\mu$ (0.3–0.4 d$^{-1}$) in the austral summer is higher than the observations (about 0.1–0.2 d$^{-1}$; Boyd et al., 2000) and the satellite-based estimates. The simulated phytoplankton biomass is too high in the equatorial Pacific ($>$ 100 mg C m$^{-3}$) and the Southern Ocean ($>$ 50 mg C m$^{-3}$; Fig. C2) compared to the satellite-based estimates ($<$ 30 mg C m$^{-3}$ for both regions; Westberry et al., 2008).

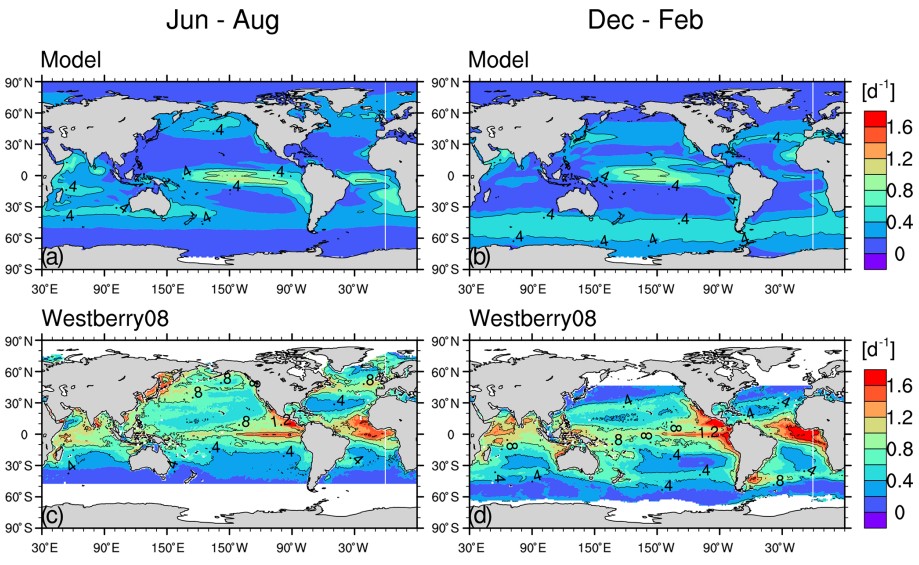

**Figure C1.** The 1999–2004 climatological-mean surface phytoplankton growth rates (d$^{-1}$) of the model (**a, b**, for bulk phytoplankton) and of the satellite-based estimates from Westberry et al. (2008) (**c, d**) for the boreal summer (**a, c**) and winter (**b, d**) TS17. The growth rate is identical between Hist_Popp and Hist_Laws.

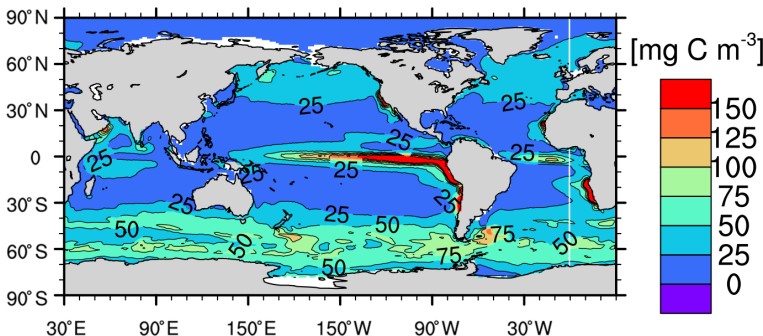

**Figure C2.** The 1999–2004 averaged annual-mean surface phytoplankton biomass (mg C m$^{-3}$) of the model.

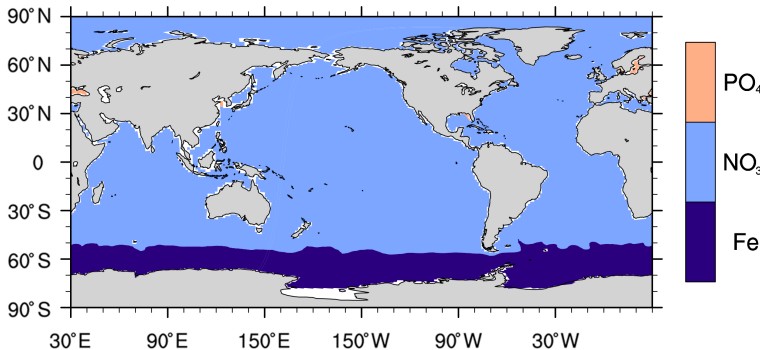

**Figure C3.** Limiting nutrients for primary production in the model.

## C2   Additional model–observation comparison for oceanic biogeochemical variables

The model captures the major features of the observed phosphate, DIC, oxygen and nitrate distribution. The biases of the above four variables are shown in Figs. 9b, 10g–i, C4, C5 and C6. We slightly underestimate the global mean phosphate by 0.2 mmol m$^{-3}$, DIC by 41.3 mmol m$^{-3}$, oxygen by 15 mmol m$^{-3}$ and nitrate by 4.7 mmol m$^{-3}$.

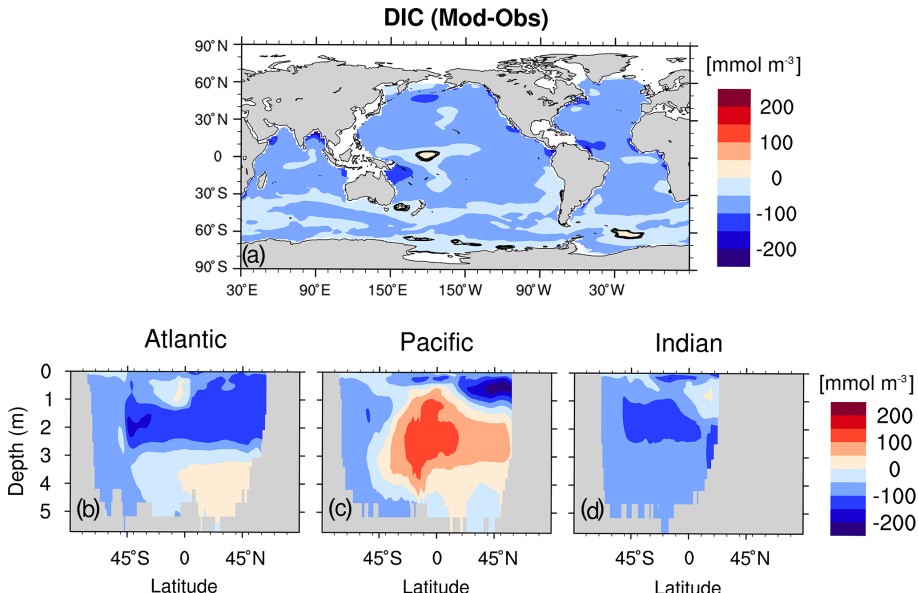

**Figure C4. (a)** DIC biases with respect to observation (GLODAPv1; Key et al., 2004) at the sea surface. **(b–d)** Zonal-mean DIC biases for the Atlantic, Pacific and Indian Ocean, respectively. Model data are averaged for 1990–1999. TS18

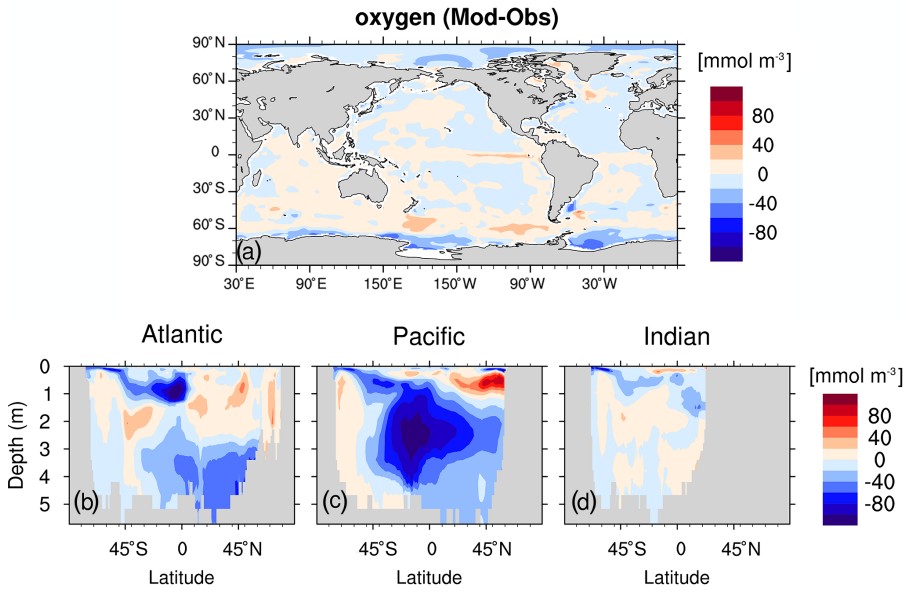

**Figure C5.** As Fig. C4 but for simulated oxygen and observation from WOA13 (Garcia et al., 2013b). TS19

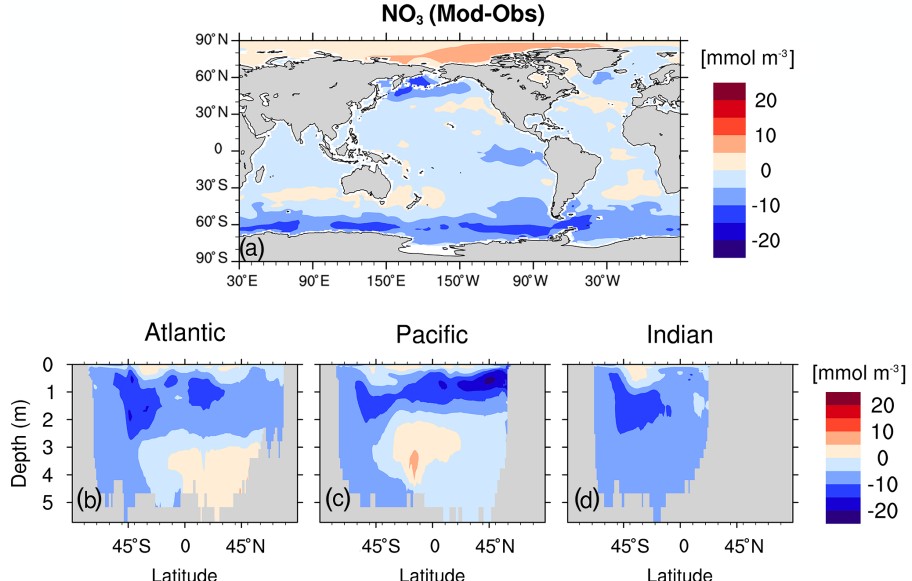

**Figure C6.** As Fig. C4 but for simulated nitrate and observation from WOA13 (Garcia et al., 2013a). TS20

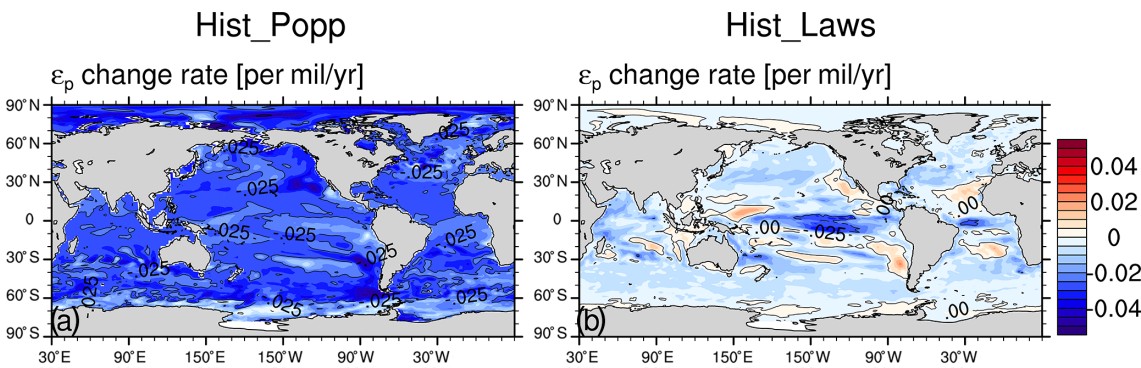

**Figure C7.** The change rate of biological fractionation $\epsilon_p$ from pre-industrial times to the 1990s TS21.

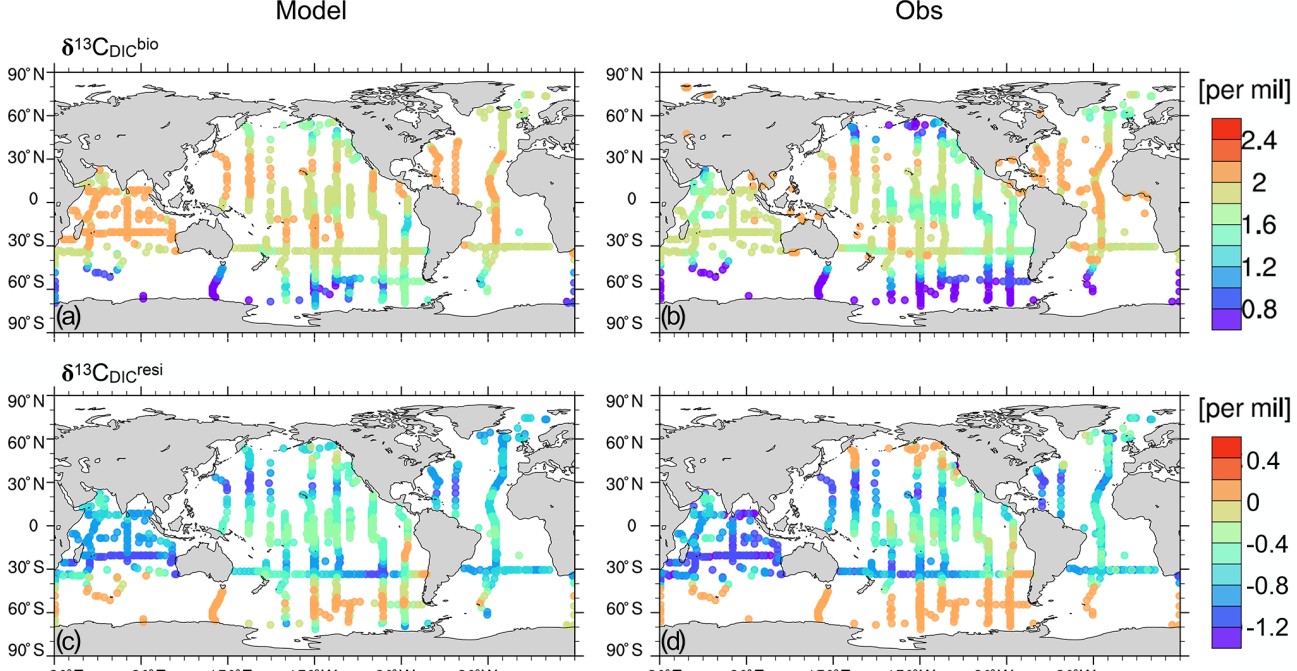

**Figure C8.** The biological component $\delta^{13}C_{\text{DIC}}^{\text{bio}}$ at the ocean surface for the model Hist_Popp (**a**) and observation (**b**). Panels (**c–d**) are as panels (**a–b**) but for the residual component $\delta^{13}C_{\text{DIC}}^{\text{resi}}$.

## Appendix D: The regenerated component of $\delta^{13}$C$_{DIC}$

The regenerated component of $\delta^{13}$C$_{DIC}$, $\delta^{13}$C$^{reg}$, relates to organic matter remineralisation and calcium carbonate dissolution. We neglect the dissolution of CaCO$_3$ following Sonnerup et al. (1999), who argued that this simplification only results in a small offset ($< 2\%$). $\delta^{13}$C$^{reg}$ is calculated as

$$\delta^{13}\text{C}^{reg} = \delta^{13}\text{C}_{DIC} - \delta^{13}\text{C}^{pref}, \tag{D1}$$

with $\delta^{13}$C$^{pref}$ given in Eq. (15). Note that the calculation of $\delta^{13}$C$^{pref}$ in Eq. (15) only applies below the 200 m, which is roughly the euphotic zone depth (Eide et al., 2017a).

The temporal change of the regenerated component $\delta^{13}\text{C}^{reg}_{SE} = \delta^{13}\text{C}^{reg}_{1990s} - \delta^{13}\text{C}^{reg}_{PI}$ (Fig. D1a–c) generally shows a much smaller magnitude than $\delta^{13}\text{C}^{pref}_{SE}$ (Fig. 12d–f). Above 1500 m, the $\delta^{13}\text{C}^{reg}_{SE}$ is mainly caused by the change in remineralisation, as is illustrated by the change in AOU (Fig. D1d–f). Below 1500 m, the $\delta^{13}\text{C}^{reg}_{SE}$ is generally negative because $\delta^{13}$C$_{POC}$ decreases by 2.2‰ from the pre-industrial period to the 1990s, mainly due to the decline of the biological fractionation factor $\epsilon_p$ under increasing surface CO$_2$(aq) (Fig. C7a).

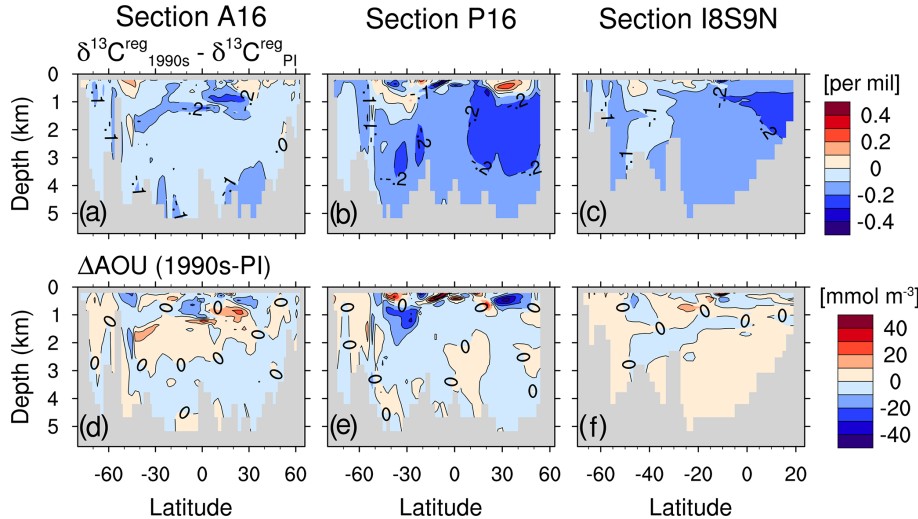

**Figure D1.** The simulated change in the regenerated component $\delta^{13}\text{C}^{reg}_{SE} = \delta^{13}\text{C}^{reg}_{1990s} - \delta^{13}\text{C}^{reg}_{PI}$ for vertical sections A16 in the Atlantic Ocean **(a)**, P16 in the Pacific Ocean **(b)** and I8S9N in the Indian Ocean **(c)**. The locations of the vertical sections are shown in Fig. 12. Panels **(d–f)** are as panels **(a–c)** but for the change in AOU from pre-industrial times to the 1990s.

## Appendix E: Applying the Eide et al. (2017a) approach to the model data

### E1 Description of the Eide et al. (2017a) approach

To derive the global oceanic $^{13}$C Suess effect, Eide et al. (2017a) (hereafter E17) first applied the two-stage back-calculation method developed by Olsen and Ninnemann (2010) to calculate the $^{13}$C Suess effect using data from the World Ocean Circulation Experiment sections. The steps and assumptions of this stage are explained below. Next E17 mapped these $^{13}$C Suess effect estimates onto a $1 \times 1°$ grid with 24 vertical layers and obtained the three-dimension distribution of the $^{13}$C Suess effect in the global ocean. For simplicity, hereafter the above procedure is collectively referred to as E17's approach.

E17 first assume that any oceanic CFC-12 signal before 1940 is negligible and the oceanic $^{13}$C Suess effect at any time $t$ after 1940, $\delta^{13}C_{SE(t-1940)}$, is proportional to CFC-12 partial pressure at time $t$:

$$\delta^{13}C_{SE(t-1940)} \sim a \cdot pCFC\text{-}12_t. \tag{E1}$$

Here the proportionality factor $a$ is time-invariant. $\delta^{13}C_{DIC}$ at any time $t$ after year 1940 is decomposed as

$$\delta^{13}C_t = \delta^{13}C_{SE(t-1940)} + \delta^{13}C^{pref}_{1940} + \delta^{13}C^{reg}_{1940}. \tag{E2}$$

The calculation of $\delta^{13}C^{pref}$ is given in Eq. (15) and $\delta^{13}C^{reg}$ in Eq. (D1). E17 include two additional terms on the right-hand side of the above equation $\Delta\delta^{13}C^{reg}$ and $\Delta\delta^{13}C^{pref}$ (see their Eq. 4), which represent any changes not related to the $^{13}$C Suess effect, e.g. changes in ocean carbon cycle. We do not explicitly write these two terms as they are set to zero by E17.

Decomposing the left-hand side of Eq. (E2) into a preformed component and a regenerated component gives

$$\delta^{13}C^{pref}_t = \delta^{13}C_{SE(t-1940)} + \delta^{13}C^{pref}_{1940}$$
$$- (\delta^{13}C^{reg}_t - \delta^{13}C^{reg}_{1940}). \tag{E3}$$

Following Gruber et al. (1996), E17 assume a steady-state ocean over the period of interest and set $(\delta^{13}C^{reg}_t - \delta^{13}C^{reg}_{1940})$ to zero, and this gives

$$\delta^{13}C^{pref}_t = \delta^{13}C_{SE(t-1940)} + \delta^{13}C^{pref}_{1940}. \tag{E4}$$

Combining Eqs. (E1) and (E4) yields linear relationship between $\delta^{13}C^{pref}_t$ and pCFC-12$_t$:

$$\delta^{13}C^{pref}_t \sim a \cdot pCFC\text{-}12_t + b, \tag{E5}$$

where $b$ contains term $\delta^{13}C^{pref}_{1940}$. Thus, the proportionality factor $a$ can be determined with $\delta^{13}C^{pref}_t$ and pCFC-12$_t$ at time $t$, and $\delta^{13}C_{SE(t-1940)}$ can be obtained with Eq. (E1).

To scale $\delta^{13}C_{SE(t-1940)}$ to $\delta^{13}C_{SE(t-PI)}$ for the full industrial period, the assumption is used that the oceanic $\delta^{13}C_{DIC}$ change scales with the atmospheric $\delta^{13}CO_2$ change, i.e.:

$$\delta^{13}C_{SE(t-PI)} = f_{atm} \cdot \delta^{13}C_{SE(t-1940)} = f_{atm} \cdot a \cdot pCFC\text{-}12_t, \tag{E6}$$

with

$$f_{atm} = \frac{\delta^{13}CO_{2,t} - \delta^{13}CO_{2,PI}}{\delta^{13}CO_{2,t} - \delta^{13}CO_{2,1940}}. \tag{E7}$$

### E2 Calculation of SE$_{pref}$, the oceanic $^{13}$C Suess effect estimate using E17's approach and model data

To achieve a result comparable to E17, we select the model data at the geographic locations for which both CFC-12 and $\delta^{13}C_{DIC}$ measurements are available. The observational data set of E17 has data from one cruise in the South Atlantic (A13.5) in 2010. We do not include this cruise data because the applied ERA20C forcing, and, thus, our simulations ends in 2009. Here we use the observations compiled by Schmittner et al. (2013) because $\delta^{13}C_{DIC}$ in this data set has been quality controlled and is publicly available. Following E17, we use data at the model layers between 200 m and the simulated CFC-12 penetration depth (defined as pCFC-12 = 20 patm (pico-atmosphere) CE8; see the thick grey lines in Fig. 14). We take model data of year $t = 1994$. By performing a linear regression (Eq. E5) for five ventilation regions (the North Atlantic, South Atlantic, North Pacific, South Pacific and Indian Ocean) we obtain the regression parameters, hereafter referred to as $a_{pref}$ and $b_{pref}$. Applying Eq. (E6) to the three-dimension model data of pCFC-12 for $t = 1994$, regression slope $a_{pref}$ and $f_{atm} = 1.5$ (determined with Eq. E7 for year 1994), we obtain the estimate of the global oceanic $^{13}$C Suess effect, SE$_{pref}$, in year 1994 (Eq. 16).

The regressional relationships between $\delta^{13}C^{pref}_{1994}$ and pCFC-12$_{1994}$ and the regression coefficients $a_{pref}$ and $b_{pref}$ are shown in Fig. E2 (the water masses in this figure are defined in Table E1). The coefficient of determination $r^2$, the percentage of the variance in the data explained by the regressional relationship, ranges between 0.33 and 0.66. The strength of these linear relationships is acceptable considering the lowest $r^2 = 0.22$ in E17.

The regression relationships between $\delta^{13}C^{pref}$ and pCFC-12 in our model (Fig. E2) show some quantitative differences compared to those of E17 (see their Fig. 3). These differences originate from model biases in the distribution and properties of water masses. These mismatches do not affect the analysis and conclusions in Sect. 5. Nevertheless, we briefly discuss their causes for better understanding of the model behaviour.

First, the definitions of several water masses in the model are slightly different from those of E17 (comparing our Table E1 with their Table 2).

Second, our simulated $\delta^{13}C^{pref}_t$ in the deep and bottom waters (Antarctic Bottom Water, Circumpolar Deep Water, Pacific Deep Water and Indian Deep Water) in the Southern

Hemisphere (Fig. E2c and e and Fig. E3c) is higher than
that in E17 (see their Fig. 3a, c and e). The possible rea-
sons for this difference are related to mixing and primary
production in the Southern Ocean. Here, the simulated deep
convection, which primarily occurs in the open ocean rather
than the along continental shelf, is too strong in the model.
This can be seen by the large mixed layer depth (Fig. B5) and
by the CFC-12 bias along selected vertical sections (Fig. B8),
which feature persistent positive biases off the Antarctic con-
tinental shelf in the Atlantic, Pacific and Indian sectors of
the Southern Ocean. Furthermore, the Southern Ocean has
a primary production that is too high in the model (about a
factor of 1.5 of the satellite-based net primary production es-
timates from Westberry et al., 2008). The high primary pro-
duction causes higher surface $\delta^{13}C_{DIC}$ than observations (see
the South Pacific Ocean in Fig. 8c). Consequently, the simu-
lated preformed component $\delta^{13}C_t^{pref}$ in the bottom and deep
water masses of the Southern Ocean is higher than observed
values in E17.

Third, the lowest values of $\delta^{13}C_t^{pref}$ ($< 1.4\,‰$) are often
found in the upwelling regions in the model. This is due to
the upward transport of water from the ocean interior that has
lower $\delta^{13}C_{DIC}$ than observations (Fig. 10e and f).

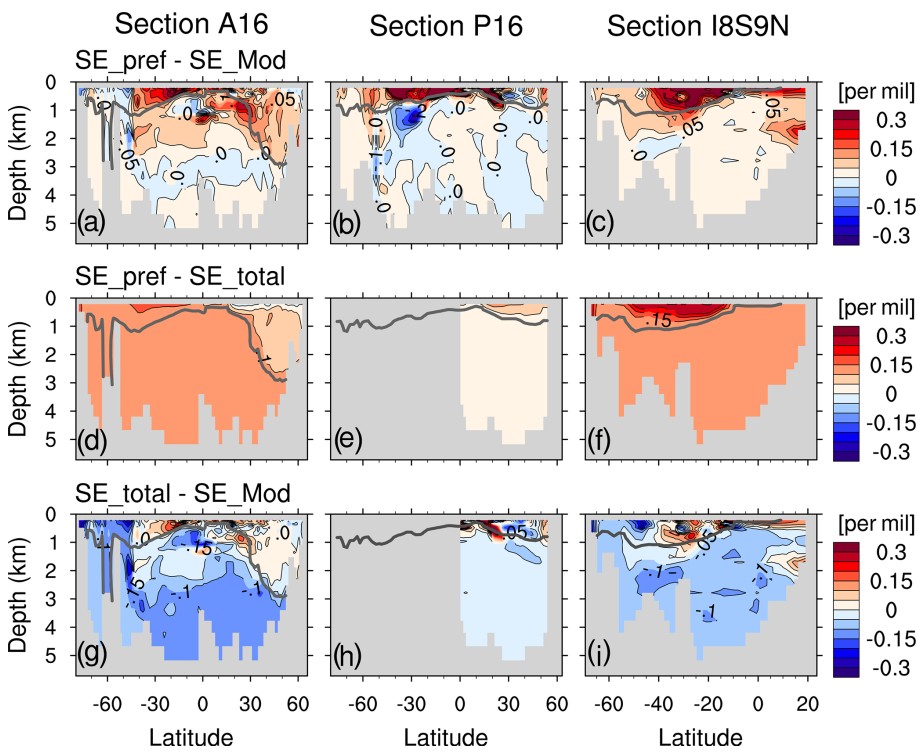

**Figure E1.** The difference (SE$_{pref}$ − SE$_{Mod}$) for the vertical sections A16 in the Atlantic Ocean **(a)**, P16 in the Pacific Ocean **(b)** and I8S9N
in the Indian Ocean **(c)**. Panels **(d–f)** and **(g–i)** are as panels **(a–c)** but for (SE$_{total}$ − SE$_{Mod}$) and (SE$_{pref}$ − SE$_{total}$), respectively. The isoline
increment is $0.05\,‰$. The thick grey line is the pCFC-12$_{1994}$ = 20 patm isoline, below which SE$_{pref}$ is generally very small ($< 0.05\,‰$).

## E3   Linear regression for subregions in the Indian Ocean

We can span regressional relationships for the subtropical gyres of the Indian Ocean and North Pacific Ocean because we use all model levels between 200 m and the pCFC-12 = 20 patm isoline at a given geographical location, and therefore we have more data points than field measurements. In the Indian Ocean, performing linear regression for $\delta^{13}C_{1994}^{pref}$ and pCFC-12$_{1994}$ in the Subtropical Gyre Water and Sub-Antarctic Mode Water yields regression parameters $a_{pref}^{STGW} = -0.65 \times 10^{-3}$, $b_{pref}^{STGW} = 1.98$ and $r^2 = 0.49$. The more negative $a_{pref}^{STGW}$ compared to regression slope $a_{pref} = -0.47 \times 10^{-3}$ obtained for the whole Indian Ocean suggests an underestimation of the $^{13}$C Suess effect. The mean pCFC-12 in the Indian subtropical region at 200 m pCFC-12$_{1994}^{STGW} = 440$ patm. Following Eq. (E6), we can calculate the mean underestimation for the subtropical Indian Ocean as $f_{atm} \cdot (a_{pref} - a_{pref}^{STGW}) \cdot$ pCFC-12$_{1994}^{STGW} = 0.12$ ‰.

## E4   Calculation of SE$_{total}$

To calculate SE$_{total}$ we perform a linear regression for the total oceanic $^{13}$C Suess effect $\delta^{13}C_{SE(1994-1940)}$ and pCFC-12$_{1994}$:

$$\delta^{13}C_{SE(1994-1940)} \sim a_{total} \cdot \text{pCFC-12}_{1994} + b_{total}. \qquad \text{(E8)}$$

Here the model data are subsampled in the same manner as in Sect. E2. Next, applying a correction for the period prior to 1940 (in analogy to Eq. E6) we obtain the expression of SE$_{total}$ in Eq. (19).

The regression relationships in Eq. (E8) and regression coefficients are given in Fig. E3. For the Indian, North Pacific, North Atlantic and South Atlantic Ocean, $r^2$ lies between 0.34 and 0.67, which suggests an acceptable strength of the relationships. In the South Pacific Ocean we find low $r^2 = 0.07$. This low $r^2$ is a result of the high variability in the change in the regenerated component (Fig. 14h) which corrupts the regression. Therefore we omit the South Pacific in the calculation of SE$_{total}$.

**Table E1.** Water masses and their definitions in the model.

| Water mass | Definition in the model |
| --- | --- |
| Indian Ocean ventilated waters | |
| Upwelling regions | north of 10° N in the Arabian Sea; north of 8° N in the Bay of Bengal |
| STGW (Subtropical Gyre Water), SAMW (Sub-Antarctic Mode Water)[a] | $\sigma_\theta \leq 27.0$ |
| AAIW (Antarctic Intermediate Water) | $27.0 < \sigma_\theta \leq 27.45$ [b] |
| IDW (Indian Deep Water), CDW (Circumpolar Deep Water) | $\sigma_\theta > 27.45$ [b] |
| North Pacific ventilated waters | |
| Upwelling regions | east of 160° W, south of 25° N, $\sigma_\theta > 26.4$ |
| STGW | $\sigma_\theta \leq 26.7$ |
| NPIW (North Pacific Intermediate Water) | $\sigma_\theta > 26.7$ |
| South Pacific ventilated waters | |
| Upwelling regions | east of 160° W, north of 15° S, $\sigma_\theta > 26.5$; east of 90° W, north of 40° N, $\sigma_\theta > 26.5$ |
| STGW, SAMW[a] | $\sigma_\theta \leq 27.15$ |
| AAIW | $26.7 < \sigma_\theta \leq 27.7$, salinity $< 35.0$ psu |
| PDW (Pacific Deep Water), CDW | $\sigma_\theta > 27.7$ |
| North Atlantic ventilated waters | |
| STGW | $\sigma_\theta \leq 27.2$, south of 45° N |
| SPMW (Subpolar Mode Water) | $26.95 < \sigma_\theta \leq 27.5$ [b] |
| NSOW (Nordic Seas Overflow Water), NADW (North Atlantic Deep Water), LSW (Labrador Sea Water) | $\sigma_\theta > 27.5$ [b] |
| South Atlantic ventilated waters | |
| STGW | $\sigma_\theta \leq 26.9$ |
| SAMW, AAIW[a] | $26.9 < \sigma_\theta < 27.4$ |
| AABW (Antarctic Bottom Water), CDW | $\sigma_\theta > 27.4$ |

[a] Water masses are combined together rather than separately defined as in Eide et al. (2017a).
[b] A different $\sigma_\theta$ threshold is used here compared to Eide et al. (2017a).

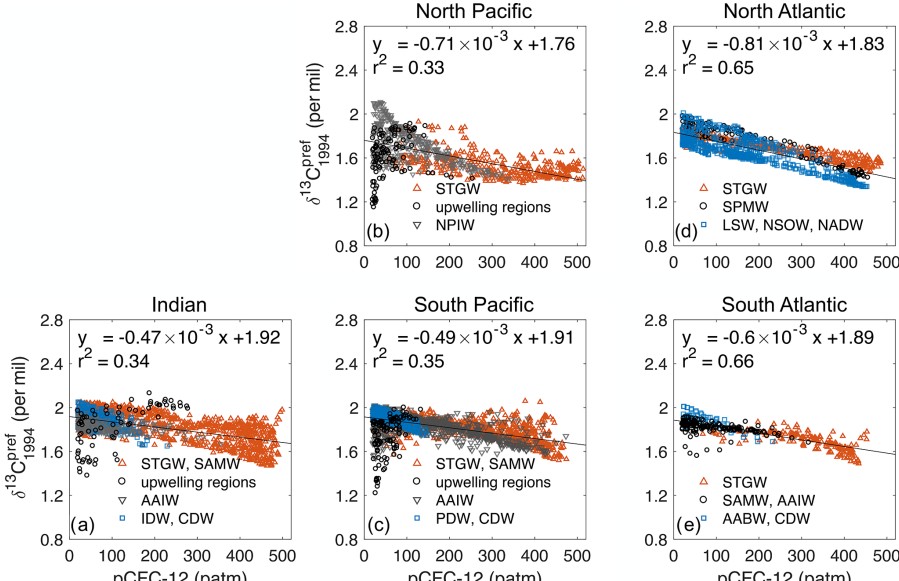

**Figure E2.** Regressional relationships $\delta^{13}\mathrm{C}^{\mathrm{pref}}_{1994} \sim a_{\mathrm{pref}} \cdot \mathrm{pCFC\text{-}12}_{1994} + b_{\mathrm{pref}}$ for the Indian Ocean **(a)**, the North Pacific **(b)**, the South Pacific **(c)**, the North Atlantic **(d)** and the South Atlantic **(e)**. Different colours and symbols indicate different water masses. The full names, as well as the definitions, of the water masses are listed in Table E1. The regression slopes $a_{\mathrm{pref}}$ are used to calculate $\mathrm{SE}_{\mathrm{pref}}$ in Eq. (16). In the Indian Ocean the regression relationship for the Subtropical Gyre Water and Sub-Antarctic Mode Water (red upward triangle in panel **a**) is $y = -0.65 \times 10^{-3}x + 1.98, r^2 = 0.49$. In the North Pacific the regression relationship for the Subtropical Gyre Water (red upward triangle in panel **b**) is $y = -0.44 \times 10^{-3}x + 1.66, r^2 = 0.26$.

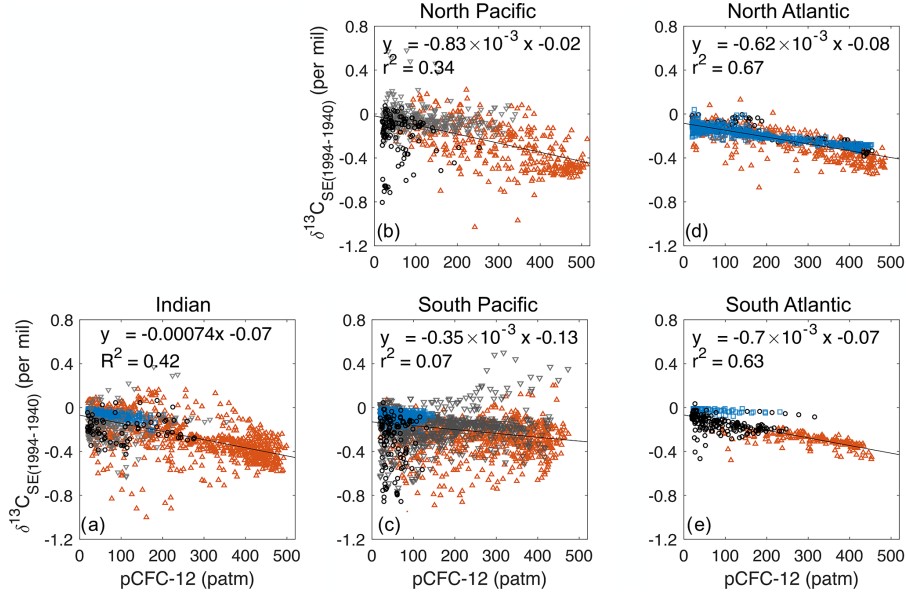

**Figure E3.** As Fig. E2 but for the regression relationships $\delta^{13}\mathrm{C}_{\mathrm{SE(1994-1940)}} \sim a_{\mathrm{total}} \cdot \mathrm{pCFC\text{-}12}_{1994} + b_{\mathrm{total}}$. The regression coefficients $a_{\mathrm{total}}$ and $b_{\mathrm{total}}$ are used to calculate $\mathrm{SE}_{\mathrm{total}}$ following Eq. (19).

*Code and data availability.* Primary data and code for this study are available from the corresponding author upon request. All observational data used in this study are available from public databases or literature, including the Met Office Hadley Centre EN4 observational data set (https://www.metoffice.gov.uk/hadobs/en4/, last access: 4 March 2020, Good et al., 2013), World Ocean Atlas 2013 (https://www.nodc.noaa.gov/OC5/woa13/, last access: 19 February 2015, Garcia et al., 2013a, b TS22), Global CE9 Data Analysis Project version 1 (https://www.ncei.noaa.gov/access/ ocean-carbon-data-system/oceans/glodap/, last access: 18 November 2004, Key et al., 2004), net air–sea $CO_2$ flux (now available at https://www.ncei.noaa.gov/access/ocean-carbon-data-system/ oceans/SPCO2_1982_2011_ETH_SOM_FFN.html, 19 April 2018, Landschützer et al., 2015), ocean primary production and growth rate (http://sites.science.oregonstate.edu/ocean.productivity/, last access: 13 November 2019, Westberry et al., 2008), $\delta^{13}C_{POC}$ (data provided by Andreas Schmittner in September 2019; this data set was originally compiled by Goericke and Fry, 1994), $\delta^{13}C_{DIC}$ (https://andreasschmittner.github.io/publications.html, last access: 8 October 2018, Schmittner et al., 2013), and the oceanic $^{13}C$ Suess effect estimate (https://doi.pangaea.de/10.1594/PANGAEA. 872004, last access: 21 March 2018, Eide et al., 2017c TS23) CE10.

*Author contributions.* BL performed the $^{13}$C model development, conducted the simulations and wrote the manuscript. KDS contributed to the model implementation and in setting up the experiments. All authors of the paper critically discussed the analysis of the results and provided valuable input on the presentation of the paper.

*Competing interests.* The authors declare that they have no conflict of interest.

*Acknowledgements.* This research contributes to the German paleoclimate modelling initiative PalMod (FKZ: 01LP1505A, 01LP1515C). PalMod is funded by the Bundesministerium für Bildung und Forschung (BMBF), and it is part of the Research for Sustainable Development initiative (FONA). Simulations were performed at the German Climate Computing Center (DKRZ). We thank Irene Stemmler for her valuable input, and we thank Friederike Fröb for the internal review of this paper. We also thank the two reviewers Anne Morée and Pearse Buchanan for their constructive comments.

*Financial support.* This research has been supported by the Bundesministerium für Bildung und Forschung (PalMod initiative).

The article processing charges for this open-access publication were covered by the Max Planck Society. TS24

*Review statement.* This paper was edited by Jack Middelburg and reviewed by Anne Morée and Pearse Buchanan.

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

## Remarks from the language copy-editor

## Remarks from the typesetter

TS24   Please note that the information given in the acknowledgements should not differ from the information given in the financial support section. It is important that all funders and grant nos. are listed in this section rather than in the acknowledgements. It is possible that the funding information can be kept in the acknowledgements. However, please check both sections carefully and advise.

TS25   Please confirm reference list entry for the data set.

TS26   Please confirm added information.

TS27   Please note that all links (except DOIs) need a last access dates, because URLs might expire.

TS28   Please confirm added information.

TS29   Please confirm.

TS30   Please confirm added information.

TS31   Please confirm added information.