# Peer review of "Incorporating the stable carbon isotope 13C in the ocean biogeochemical component of the Max Planck Institute Earth System Model"

_Biogeosciences, 2021_

## Referee Comment (RC1)

**Liu et al. (2021) – Incorporating the stable carbon isotope $^{13}$C in the ocean biogeochemical component of the Max Planck Institute Earth System Model.**

Review by Pearse J. Buchanan.

**General comments**

Liu and colleagues have made a useful contribution to ongoing efforts to develop carbon isotope routines within ocean biogeochemical models. They use the biogeochemical model HAMOCC6, which is part of the MPI earth system model, to make two main findings:

1.  They compare two accepted parameterisations for the biological fractionation of DI$^{13}$C by phytoplankton to show that the Popp parameterisation performs much better than the Laws parameterisation. This judgement of skill is made by comparing with observed d13C$_{POC}$ values.
2.  They perform hindcast simulations from 1920-2010 using the ERA20C reanalysis to quantify the Suess effect, and then use the model to explore why the Suess effect might be underestimated by a recent observation-based approach to quantify it (Eide et al., 2017a).

I find the paper to be a valuable contribution and certainly worthy of publication. The authors have done a lot of work and their approach is rigorous. I have two major requests:

a)  The agreement with d13C$_{POC}$ data using Popp parameterisation is compelling as shown by Figure 4. Meanwhile, the consistent underestimation of d13C$_{POC}$ by the Laws parameterisation is very clear, but it remains the only parameterisation (to my knowledge) that includes the effect of growth rate, which is an important effect as shown by yourselves in the Subantarctic. You discuss this in your section 3, noting that for the Laws parameterisation to be improved, we would need to alter the slope and intercept of that relationship (although it may actually not be able to perform as well as the Popp parameterisation if it is in fact limited by the nature of the equation (inverse vs. logarithm)). I wonder whether this study could be made even more valuable by refitting the Laws parameterisation? Given that you have the observations, and at each point you have model $CO_2$(aq) and growth rate, could you optimise the Laws parameterisation by re-solving for the slope and intercept? As a fellow biogeochemical model developer, I would find this paper incredibly valuable if it offered a means to improve the Laws parameterisation, which is commonly used.

b)  I might be slow, but I only understood things in section 4.2 with multiple re-readings of this paper using more brain power than I'd like. I understood everything more or less easily up until the paragraph beginning at line 489, at which point you explain why the method of Eide (SE$_{pref}$) underestimates the Suess effect compared to your method with perfect knowledge of the preformed d13C$_{DIC}$ (SE$_{total}$). In the Indian Ocean, North Pacific and South Atlantic, you show that Eide's method underestimates your model method. You explain that there are two reasons for this:
    1.  That preformed d13C$_{DIC}$ at 1940 (the intercept) is constant in the regression equation, but in reality this is spatially variable and decreases with depth, such that

use of a constant value over a subregion like the Indian Ocean will bias near-surface values as underestimated and deeper values as overestimated (?).

2. That the neglect of the suess effect in waters without CFC-12 prior to 1940 is going to underestimate the effect.

Point 2 I get easily, and also appears to be the main reason for difference between Eide's method and the "best" possible estimate using linear regression with the modelled CFC-12. Point 1 if I'm honest I still don't understand well, if at all. Part of this confusion comes from the fact that your earlier explanation of error between $SE_{total}$ and $SE_{mod}$ must take into account the error around the linear regression, which includes **both** coefficients. I would really appreciate a clearer explanation, not only here as a response to myself, but also I think if I am not getting this easily, it would be worth re-visiting section 4.2 and editing how this information is presented so that it is more easily digested by others. I think section 4.2 needs to be made more concise and more clear.

**Specific comments**

- Line 92: So the input of nutrients at the ocean surface happens where exactly? Everywhere or just where rivers are?
- Paragraph beginning at line 164: This needs simplifying. I would detail the spin-ups first, then talk about the reanalysis runs from 1850-2010.
- Line 275: I would expect the Laws parameterisation to have a higher global mean $d13C_{DIC}$ because of burial, not lower. Yes more $^{13}C$-deplete material is remineralised in the interior, but over time more $^{13}C$-depleted material would have been buried in the sediments. Meanwhile, at the surface the $d13C_{DIC}$ is higher in Laws, which according to your model's balancing of budgets would add more $^{13}C$ to the surface than was buried. Over time, this would lead to an increase in $d13C_{DIC}$ in the parameterisation that produced more $^{13}C$-deplete organic matter (i.e. Laws). A positive excursion of deep ocean $d13C_{DIC}$ in the paleoceanographic record is actually explained by an increase in the biological flux of material to the sediments. This suggests to me that you may not have run these simulations to steady state. The good news about this is that it doesn't affect your conclusions of which parameterisation is better at calculating $d13C_{POC}$. Popp is clearly the better parameterisation and should continue to be even if your simulations were run to steady state.
- Paragraph beginning line 315: But too strong upwelling and too shallow remineralisation is not an explanation for too high biological fractionation in the Southern Ocean. Macronutrients are already unlimiting to primary production here, so I would say that iron concentrations are too high, much like they are in the North Pacific.
- Paragraph beginning 352: Please see Figure 2 in Buchanan et al., (2019) in Geoscientific Model Development for other model-data measures of agreement with $d13C_{DIC}$.
- Equation 23: Surely the intercept from $SE_{pref}$ is important here? Why is it excluded? Some error between $SE_{pref}$ and $SE_{total}$ must be due to differences in the intercepts?
- Line 562: The underestimation of global mean $d13C_{DIC}$ by Laws parameterisation may in fact change if you preindustrial spin-ups were run for longer. Might be worth noting that here.

**Technical corrections:**

- Can the authors double check the coefficients in equations 5 and 7 please? The kinetic fractionation should by -0.88 per mil, and the second coefficient in equation 7 should be -1.07 according to Orr et al., (2017). Also, even though I'm sure it doesn't make much difference to the results, I find it strange that they didn't just include the full equations as detailed by Orr et al., (2017) when this has absolutely miniscule effect on computation time. In other words, there doesn't seem to be any real reason to not include the full equation that other modelling groups have implemented.
- Line 132: "preferentially utilised over".
- Line 335: "negative" should be "positive".
- Line 364: "generally" should be "general"
- Line 421: should "above" be "below"? deeper than?
- Line 440: you mean global mean surface ocean-atmosphere right? Not global mean ocean-atmosphere, which would include all depths.
- Line 508: "CFC-12" should be "CFC-12-free"
- Line 574: "Model" should be "Mode"

Thank you for considering my input to your research,

Pearse

---

## Referee Comment (RC2)

**General Comments**

This manuscript by Bo Liu, Katharina Six and Tatiana Ilyina provides the biogeochemical ocean modelling community with valuable information for both model development and analysis. The implementation of the 13C tracers in an ESM is a tremendous and tedious effort, which I want to acknowledge. First of all, Liu et al. provide a detailed analysis of the differences in the fractionation parameterization used for photosynthesis – showing that for their model the parameterization of Popp et al. 1998 leads to better results especially for d13C_POC. Secondly, Liu et al. have explored the Suess effect in their historical model run by comparing to the observational-based Suess effect estimate of Eide et al., (2017). Using their model, they have been able to pin down the causes of the Eide et al. 2017 underestimation of the Suess effect. Most of the manuscript is well-written and thorough, making it generally easy to read and navigate as a reader. I recommend the article for publication, although I first see several points that could be improved as listed in the remainder of this document. I wish to start with a few general comments:

1. In the Introduction, a stronger argument could be made for why you decided to compare the different parameterizations, and why specifically these two. Some earlier studies state that model δ13C distributions are not very sensitive to the chosen parameterization (e.g., Schmittner 2013), especially not in the surface ocean (e.g. Jahn et al., 2015). It should be discussed in Sect. 3 why your model shows something else. An interesting study on biological fractionation is e.g. that of Young et al. (2013) – such results should be discussed in light of your own.

2. As you in Sect.3 on model results mostly discuss d13C, I think it is important to add a section under Sect. 3 where you introduce the reader to the model performance for the other tracers: Plus, even though this study is on the biogeochemistry, I think it is important for the reader to get an introduction to the physical ocean model performance and specifics as well, and how it performs compared to obs (fe AMOC / Drake / SST / SSS / T / S / sea ice) - which probably has been described in a separate study but would be good to repeat here. I mainly stress this because d13C is governed by both circulation and biological processes, so any results obtained with your model setup also depend on the simulated circulation. Please summarize the biogeochemical performance shortly as well (i.e. how does e.g. PO4. O2, DIC and NO3 distributions compare to obs).

3. Section 4.2 is difficult to follow. Help the reader by clarifying why you are re-calculating the E17 approach beyond that you have more data available as you use a model. Extra subsections, a thorough shortening and more focus on the results would help. I have to note that I am not an expert on the Suess effect or the E17 approach, but I feel I should be able to follow it based on my experience with d13C modelling. It seems quite some text is used to describe the figures: one could instead refer to the figures and only highlight the most important features of the figures.

Best wishes,

Anne Morée

**Specific Comments**

**Abstract**

p1, l2-3: 'Direct comparison between paleo oceanic δ13C records and model results facilitates assessing simulated distributions and properties of water masses in the past.'

This is true, but your study (mostly) focuses on the use of d13C in understanding the contemporary ocean, as you are able to use observations of POC-d13C and explore the Suess effect. This first sentences of your abstract sounds as if the goal of implementing d13C in HAMOCC was motivated by paleoceanographic questions only. Please rephrase.

P1, l11: 'because the latter results in a too strong preference for 12C'

I think here the reader could get confused (how can d13C_DIC be OK for both parametrizations but d13C_POC be better for Popp?). Maybe add something like '.. during C fixation, resulting in too low d13C_POC.'

p1, l15: It is not entirely clear from this sentience where the 'that' refers to: Has your model ample spatial and temporal data coverage? I think you want to stress here that you can repeat the Eide et al (2017) procedure with the advantage that you model has higher temporal and spatial resolution. Please clarify/rephrase.

P1, l14-20: This part about the Eide et al. approach makes up for almost one third of your abstract and does not connect so well to the first part. Why did you focus on the Eide et al approach, and how did you apply the findings from the first part of the study to the Eide et al part?

It would also be good to finish the abstract with an outlook or overarching conclusion/summary.

**Sect. 1 Introduction**

p2, l36: Here one could add additional studies such as referring to HAMOCC2s by Heinze and Maier-Reimer (1999).

p2 l43-45: Note that the biogeochemical model in NorESM is also called HAMOCC - one could confuse the reader here by making general statements suggesting there is only one 'HAMOCC'. See Tjiputra et al., 2020 for the description of the implementation of d13C in NorESM-OC.

p3, l55: After this sentence I would expect a paragraph on both d13C_DIC (which you provide) and d13C_POC (which is missing). Let the reader know already here what data you've used of d13C_POC like you do for DIC.

**Sect 2. Model description**

p 4, l106: It is not only small, but also very uncertain (e.g., Zeebe and Wolf-Gladrow 2001). one could also mention here that this is commonly omitted in modeling studies. To name a few:

Schmittner, A. et al. Biology and air-sea gas exchange controls on the distribution of carbon isotope ratios (δ13C) in the ocean. Biogeosciences 10, 5793-5816, doi:10.5194/bg-10-5793-2013 (2013).

Lynch-Stieglitz, J., Stocker, T. F., Broecker, W. S. & Fairbanks, R. G. The influence of air-sea exchange on the isotopic composition of oceanic carbon: Observations and modeling. Global Biogeochemical Cycles 9, 653-665, doi:10.1029/95GB02574 (1995)

Tjiputra, J. F., Schwinger, J., Bentsen, M., Morée, A. L., Gao, S., Bethke, I., Heinze, C., Goris, N., Gupta, A., He, Y. C., Olivié, D., Seland, Ø., and Schulz, M.: Ocean biogeochemistry in the Norwegian Earth System Model version 2 (NorESM2), Geosci. Model Dev., 13, 2393-2431, 10.5194/gmd-13-2393-2020, 2020.

p4, l108-109: I suggest to move this sentence down. This would make it clearer that you first discuss total C exchange and then go into the isotope exchange.

P5, l124: You actually deviate here from the OMIP protocol of Orr et al 2017, who recommend taking 0.88 permil

p5, l126: Similarly here, you deviate from the OMIP protocol / the original formula who use 0.0144 and 0.107.

p5, l130: Why did you decide to simplify the equations here, when it computationally is not a large burden to include the whole equation? Also, I would argue again that you are not following the CMIP protocol here if you decide to simplify the air-sea gas exchange equations.

P5, l132: Use 'is preferred over' instead of 'is preferentially utilised than' or rephrase in another way.

P7, l177-178: so do I understand it correctly that you base your 13C model field on 12C (i.e. total C) of the model and the PO4 of the model, after the initial spinup without the 13C? Please clarify. The initialization process of an isotope model is important for spinup duration, so detailed information on this can be valuable to the readers of your work.

P8, l182-184: This input rate is the input to compensate for the loss to the sediments, right? Is it equally distributed over the surface ocean?

P8, l186-187: the inventory adjusts to be consistent with the simulated processes – what does that mean? Is the result agreeing with observational fields? This relates maybe also to my general comment that I miss an overview/summary of the non-13C performance of the model regarding both circulation and biogeochemistry.

P8, l195: Is the sediment 13C also already equilibrated after the 2500y spinup?

**Sect 3. Model results and observations in the late 20th century**

p8, l199: One could mention here that you are not using the Eide et al. estimate of pre-industrial d13C because that is based on her estimate of the Suess effect, which you have re-evaluated. That said, you could (like Eide et al. have done) share your Suess effect estimate as a gridded dataset. On p21, l419-420 you also explain that the E17 dataset and the Schmittner dataset are not so different, which you could mention earlier.

P8, l200-202: Why not regrid the obs data to the model grid, instead of regridding to 1x1 and then doing the same for the model? Also you mean the model-obs comparison for d13C_DIC here, because for POC in Fig. 4 for example you are comparing model and obs without regridding – did you then take the nearest model value?

P9, l207: Why do you start with POC here instead of DIC? I expect more readers will be familiar with the d13C_DIC distributions. Also, this Section is about *simulated* isotopic signature, maybe add this to the title?

P9, l221-224: I think you should not only refer to Appendix B when it comes to what the model simulates, but also refer to Appendix B when it comes to how that compares to observational estimates. You could summarize the most important results of Appendix B here. Also, Fig B1 is for the Popp results? Note that for clarity if you refer to model results (like also in e.g. Fig D3) it is good to say which model run you mean.

P10, l242: I understand you want to make an evaluation of the model performance here of CO2_aq around 45S in Fig 4g, but why choose this particular dataset - aren't there more recent data with better coverage available?

P10, l245: Hist_Laws captures more than Hist_Popp, but not all and more importantly offset by a few permil. If this offset is really very constant, wouldn't it be possible to adjust the Laws et al parameterization. One could evaluate the offset needed and see if after that offset Popp is still superior?

P11, Fig. 4: Are there uncertainty estimates available for the obs data, or would it be possible to at least shade an estimate of the uncertainty?

P12, l281: this seems a bit repetitive, wouldn't a too steep vertical gradient always lead too too low deep d13C_DIC if surface d13C_DIC is reasonable? This could be rephrased.

P13, l284-285: this is a very important transition point for the reader, where you decide to mainly focus on Popp from now on. This could be mentioned earlier (or even in the abstract), or denoted by a new section here. For example, make a 3.2.1 and a 3.2.2 section.

P13, l286: I got slightly lost here. Maybe explain the reader how this d13C_DIC comparison is different from the one in Fig. 7. It could help to make Fig 8a,b,c one figure, and present the d13C_bio and d13C_resi and the net air-sea CO2 flux separately. Regarding d13C_bio and d13C_resi I think it helps with comparison to earlier studies if you show the absolute values and not only the model-obs difference.

P13, l294: Why do you use Δ_photo and not ε_p like before here?

P13, l295: Do you use the same R_C:P for the components calculations of both obs and model?

P13, l296: I think it would be appropriate here to remind the reader that ε_p actually varies, referring e.g. to Fig. 3b,d. Also, what model MO DIC, PO4 and d13C were used? Is the R_C:P of 122 the one used in the model for consistency?

P15, l302-304: Their MO values should be included here as done for the obs.

P17, l312-314: This is some information on the physical model performance that I think should be introduced earlier and possibly in an own subsection under Sect. 2.

p17, l324-326: One could quantify the effect on d13C_DIC by analysing the bio and resi components for the PI run instead of the or in addition to the hist runs.

P17, l329: Here you continue to Fig. 9- the reader could use a bit more guidance here: what are you going to present and discuss here in this section/paragraph? Why do you go away from showing d13Cbio and d13Cresi?

**Sect. 4 Oceanic 13C Suess effect**

p18, l 367-369: Clarify here why this is important for your discussion of the Suess effect. You could also add a comment here that even though the model does well simulating the total anthropogenic C uptake, locally air-sea exchange fluxes deviate from obs (Fig. 8f).

P19, l385: This section is quite long and heavy, and I have to admit I did not follow all of it. I would suggest to cut it up in different subsections, which can discuss the different aspects which you have investigated. Start early in Sect. 4.2 with why it is relevant to re-evaluate the approach by E17 (e.g., is it often used?). Large parts of the text also feel like a methods section – could more of this section be moved to a supplement/appendix C, such that more focus on the results can be given here?

P20, l393: You have only discussed total global anthropogenic C uptake, and you have compared to E17 at depth (Sect. 4.1) - now you are going to explore the

E17 underestimation after concluding that your model produces similar results to E17? This is somewhat of a confusing step.

p26, l518: How does this compare to observational estimates (e.g. Young et al., 2013)?

**Sect. 5 Summary and conclusions**

p27, l565: One should add a short paragraph here summarizing your d13C_bio and d13C_resi component analysis for the hist_Popp run.

P28, l582: the Popp et al., 1989 parameterization has a satisfactory performance for the PI and historical times. I would agree this encourages reliability for paleoclimatic simulations, but I think some more critical remarks are in place (which should come before Sect. 5). E.g., in the past $\varepsilon\_p$ was possibly different due to different ecosystem structures or other influences, Redfield ratios could have changed (Ödalen et al., 2020).

**Technical Corrections**

p6, l146: Zeeb_e_

P17, l335: _positive_ biases between 1000 and 3000m

p19, l387: _the_ 13C Suess effect

p20, l391: _at_ 200 m _depth_

p27, l561: _again_ yields slightly better agreement

p28, l574: _Mode_ Water and explain_s_

p26, l528: Fig. D_Z_h?

**References of the review**

Heinze, C. and Maier-Reimer, E.: The Hamburg Oceanic CarbonCycle Circulation Model Version "HAMOCC2s" for long time integrations,Tech. rep., Max Planck Institute for Meteorology, Hamburg,Germany, Series: Technical Reports, no. 20, ISSN 0940-9327, 1999.

Jahn, A. _et al._ Carbon isotopes in the ocean model of the Community Earth System Model (CESM1). _Geoscientific Model Development_ **8**, 2419-2434, doi:10.5194/gmd-8-2419-2015 (2015).

Orr, J. C., Najjar, R. G., Aumont, O., Bopp, L., Bullister, J. L., Danabasoglu, G., Doney, S. C., Dunne, J. P., Dutay, J.-C., Graven, H., Griffies, S. M., John, J. G., Joos, F., Levin, I., Lindsay, K., Matear, R. J., McKinley, G. A., Mouchet, A., Oschlies, A., Romanou, A., Schlitzer, R., Tagliabue, A., Tanhua, T., and Yool, A.: Biogeochemical protocols and diagnostics for the CMIP6 Ocean Model Intercomparison Project (OMIP), Geoscientific Model Development, 10, 2169–2199, https://doi.org/10.5194/gmd-10-2169-2017, 2017.

Schmittner, A., Gruber, N., Mix, A. C., Key, R. M., Tagliabue, A., and Westberry, T. K.: Biology and air-sea gas exchange controls on the distribution of carbon isotope ratios (δ 13 C) in the ocean, Biogeosciences, 10, 5793–5816, https://doi.org/10.5194/bg-10-5793-2013, 2013.

Young, J. N., Bruggeman, J., Rickaby, R. E. M., Erez, J., and Conte, M. (2013), Evidence for changes in carbon isotopic fractionation by phytoplankton between 1960 and 2010, *Global Biogeochem. Cycles*, 27, 505– 515, doi:10.1002/gbc.20045.

Ödalen, M., Nycander, J., Ridgwell, A., Oliver, K. I. C., Peterson, C. D., and Nilsson, J.: Variable C/P composition of organic production and its effect on ocean carbon storage in glacial-like model simulations, Biogeosciences, 17, 2219–2244, https://doi.org/10.5194/bg-17-2219-2020, 2020.

---

## Author Comment (AC1)

**Response to the comments of Referee 1**

We are thankful for the constructive remarks of Pearse J. Buchanan. In the following we reply to the comments point by point.

**Two major requests**

**Comment 1** The agreement with d13C$_{\text{POC}}$ data using Popp parameterisation is compelling as shown by Figure 4. Meanwhile, the consistent underestimation of d13C$_{\text{POC}}$ by the Laws parameterisation is very clear, but it remains the only parameterisation (to my knowledge) that includes the effect of growth rate, which is an important effect as shown by yourselves in the Subantarctic. You discuss this in your section 3, noting that for the Laws parameterisation to be improved, we would need to alter the slope and intercept of that relationship (although it may actually not be able to perform as well as the Popp parameterisation if it is in fact limited by the nature of the equation (inverse vs. logarithm)). I wonder whether this study could be made even more valuable by refitting the Laws parameterisation? Given that you have the observations, and at each point you have model CO2(aq) and growth rate, could you optimise the Laws parameterisation by re-solving for the slope and intercept? As a fellow biogeochemical model developer, I would find this paper incredibly valuable if it offered a means to improve the Laws parameterisation, which is commonly used.

**Response**: Thank you for the inspiring question. As the optimisation of the Laws parameterisation is beyond the scope of this study, we use a back-of-the-envelope calculation to refit the slope and intercept of $\epsilon_{\text{p}}^{\text{Laws}}$ (Eq. 7).

Because $\alpha_{\text{Phy}\leftarrow\text{DIC}}$, $\alpha_{\text{aq}\leftarrow\text{g}}$ and $\alpha_{\text{DIC}\leftarrow\text{g}}$ are close to unity, $\epsilon_{\text{Phy}\leftarrow\text{DIC}} \approx \delta^{13}\text{C}_{\text{Phy}} - \delta^{13}\text{C}_{\text{DIC}}$ and $\epsilon_{\text{Phy}\leftarrow\text{DIC}} \approx \epsilon_{\text{p}} + \epsilon_{\text{aq}\leftarrow\text{DIC}} \approx \epsilon_{\text{p}} + \epsilon_{\text{aq}\leftarrow\text{g}} - \epsilon_{\text{DIC}\leftarrow\text{g}}$. As $\delta^{13}\text{C}_{\text{Phy}} \approx \delta^{13}\text{C}_{\text{POC}}$, $\delta^{13}\text{C}_{\text{POC}}$ can be approximated as

$$\delta^{13}\text{C}_{\text{POC}} \approx \epsilon_{\text{p}} + \epsilon_{\text{aq}\leftarrow\text{g}} - \epsilon_{\text{DIC}\leftarrow\text{g}} + \delta^{13}\text{C}_{\text{DIC}}. \tag{R1}$$

As sea water temperature (determining $\epsilon_{\text{DIC}\leftarrow\text{g}}$), CO2(aq) and growth rate are independent of the choice of $\epsilon_{\text{p}}$, and the surface d13DIC is only marginally affected by the choice of $\epsilon_{\text{p}}$ (see Section 3.2 and Figs. 5a and 5b), we can approximate $\epsilon_{\text{p}}^{\text{Laws}}$ using existing monthly output of the above model variables. The surface d13POC obtained with Eq. (R1) (r=0.74, NRMSE=2.8, Figs. R1a-R1c) is indeed very close to the simulated d13POC (r=0.71, NRMSR=2.5, Figs. 4b, 4d and 4f).

$$\epsilon_{\text{p}}^{\text{Laws}} = 68.3 \, \frac{\mu}{\text{CO}_2(\text{aq})} - 24.7, \tag{7}$$

$$\epsilon_{\text{p}}^{\text{Laws\_V1}} = 68.3 \, \frac{\mu}{\text{CO}_2(\text{aq})} - 20, \tag{R2}$$

$$\epsilon_p^{\text{Laws\_V2}} = 68.3\, \frac{\mu}{\text{CO}_2(\text{aq})} - 16.7, \tag{R3}$$

$$\epsilon_p^{\text{Laws\_V3}} = 100\, \frac{\mu}{\text{CO}_2(\text{aq})} - 20. \tag{R4}$$

As the intercept increases (comparing Eq. 7, R2 and R3), d13POC generally increases (Figs. R1a-R1i) and accordingly NRMSE decreases from 2.8 for Eq. (7) to 1.4 for Eq. (R2) and 0.75 for Eq. (R3), while the spatial relation coefficient remains unchanged.

[Figure]

Figure R1: As Figure 4b, 4d and 4f in the manuscript, but for d13POC estimated for $\epsilon_p^{\text{Laws}}$ (Eq. 7) using Eq. (R1) (a-c). (d-f), (g-i) and (j-l): As (a-c), but for $\epsilon_p^{\text{Laws\_V1}}$ (Eq. R2), $\epsilon_p^{\text{Laws\_V2}}$ (Eq. R3) and $\epsilon_p^{\text{Laws\_V3}}$ (Eq. R4), respectively.

As the slope increases (comparing Eq. R2 and R4), the d13POC difference between low and high latitudes increases (Figs. R1d-R1f, R1j-R1l) and NRMSE slightly decrease from 1.4 to 1.3. However, d13POC in the low latitude of the Atlantic shows too high variability compared to the observation. Accordingly, the spatial correlation coefficient decreases from 0.74 to 0.66.

In summary, a systematic refitting of the Laws parameterisation could be an interesting step to improve the model performance and might be considered in our future work.

**Comment 2** I might be slow, but I only understood things in section 4.2 with multiple re-readings of this paper using more brain power than I'd like. I understood everything more or less easily up until the paragraph beginning at line 489, at which point you explain why the method of Eide (SE_{pref}) underestimates the Suess effect compared to your method with perfect knowledge of the preformed d13C_{DIC} (SE_{total}). In the Indian Ocean, North Pacific and South Atlantic, you show that Eide's method underestimates your model method. You explain that there are two reasons for this:

1. That preformed d13C$_{DIC}$ at 1940 (the intercept) is constant in the regression equation, but in reality this is spatially variable and decreases with depth, such that use of a constant value over a subregion like the Indian Ocean will bias near-surface values as underestimated and deeper values as overestimated (?).

2. That the neglect of the suess effect in waters without CFC-12 prior to 1940 is going to underestimate the effect.

Point 2 I get easily, and also appears to be the main reason for difference between Eide's method and the 'best' possible estimate using linear regression with the modelled CFC-12. Point 1 if I'm honest I still don't understand well, if at all. Part of this confusion comes from the fact that your earlier explanation of error between SE$_{total}$ and SE$_{mod}$ must take into account the error around the linear regression, which includes both coefficients. I would really appreciate a clearer explanation, not only here as a response to myself, but also I think if I am not getting this easily, it would be worth re-visiting section 4.2 and editing how this information is presented so that it is more easily digested by others. I think section 4.2 needs to be made more concise and more clear.

**Response**: Both 13C Suess effect $\delta^{13}$C$_{SE(t\text{-}1940)}$ and pCFC-12 in the ocean result from the invasion of atmospheric signal. Thus their spatial distribution resembles each other: both show larger absolute values at the surface than in the interior ocean. $\delta^{13}$C$_{1940}^{pref}$ also has a specific vertical structure in our model: it is generally more positive in the upper ocean than the deep ocean. According to the equation

$$\delta^{13}\text{C}_t^{pref} = \delta^{13}\text{C}_{SE(t\text{-}1940)} + \delta^{13}\text{C}_{1940}^{pref},$$

and $\delta^{13}$C$_{SE(t\text{-}1940)} < 0$, $\delta^{13}$C$_t^{pref}$ shows a smaller vertical gradient than $\delta^{13}$C$_{SE(t\text{-}1940)}$. Thus, a linear regression for $\delta^{13}$C$_t^{pref}$ and pCFC-12 results in a less negative slope $a_{pref}$ than a slope obtained with a spatially-uniform $\delta^{13}$C$_{1940}^{pref}$.

In the revised manuscript the above explanation is incorporated to clarify Point 1.

To improve the readability of this section, we restructure it and divide it into subsections. We also move the detailed description of the E17 approach and calculation procedures to the Appendix to make this section more focused on the results and discussion.

**Specific comments**

**Comment 3**  Line 92: So the input of nutrients at the ocean surface happens where exactly? Everywhere or just where rivers are?

**Response**: The input of nutrients are added uniformly at the ocean surface. This is now specified in the revised manuscript.

**Comment 4**  Paragraph beginning at line 164: This needs simplifying. I would detail the spin-ups first, then talk about the reanalysis runs from 1850-2010.

**Response**:  Agree. In the revised manuscript this paragraph only focus on the spin-up runs.

**Comment 5**  Line 275: I would expect the Laws parameterisation to have a higher global mean d13C$_{\text{DIC}}$ because of burial, not lower. Yes more 13C-deplete material is remineralised in the interior, but over time more 13C-depleted material would have been buried in the sediments. Meanwhile, at the surface the d13C$_{\text{DIC}}$ is higher in Laws, which according to your model's balancing of budgets would add more 13C to the surface than was buried. Over time, this would lead to an increase in d13C$_{\text{DIC}}$ in the parameterisation that produced more 13C-deplete organic matter (i.e. Laws). A positive excursion of deep ocean d13C$_{\text{DIC}}$ in the paleoceanographic record is actually explained by an increase in the biological flux of material to the sediments. This suggests to me that you may not have run these simulations to steady state. The good news about this is that it doesn't affect your conclusions of which parameterisation is better at calculating d13C$_{\text{POC}}$. Popp is clearly the better parameterisation and should continue to be even if your simulations were run to steady state.

**Response**:

Our pre-industrial spin-up simulations have been run to equilibrium regarding d13DIC, according to the OMIP protocol: Equilibrium states are reached with 98% of the ocean volume having a $\delta^{13}$C$_{\text{DIC}}$ drift of less than 0.001‰ year$^{-1}$.

The difference in DI$^{13}$C water-column inventory (and therefore difference in global mean d13DIC) between Popp and Laws is determined by the differences in air-sea gas exchange, input of DO$^{13}$C and $^{13}$CO$_3^{2-}$, loss of PO$^{13}$C and $^{13}$CO$_3^{2-}$ to sediment and the sediment DI$^{13}$C reflux, see Table A1. Indeed more 13C is added to the surface than buried, which leads to slight increase of 13C inventory over time. This increase is larger in Laws than Popp by 30.5 Gmol C/yr. However, Laws parameterisation leads to a slightly higher mean surface d13DIC. This difference in surface d13DIC causes Laws to have a smaller air-sea $^{13}$CO$_2$ flux into the ocean than Popp, with a difference of -272.1 Gmol C /yr. Thus the lower DI$^{13}$C inventory in PI_Laws than PI_Popp primarily results from the difference in air-sea gas exchange.

We correct an error in Table A1: the numbers 626.6 and 596.1 should be swapped.

**Comment 6**  Paragraph beginning line 315: But too strong upwelling and too shallow remineralisation is not an explanation for too high biological fractionation in the Southern Ocean. Macronutrients are already unlimiting to primary production here, so I would say that iron concentrations are too high, much like they are in the North Pacific.

**Response**:  We agree with Referee #1 that too high iron concentrations are likely the main cause of the too high primary production in the Southern Ocean. Iron concentration in the Southern Ocean

is 0.2-0.4 nmol/L in our model, compared to observations (generally <0.25 nmol/L according to eGEOTRACES, https://www.egeotraces.org). Iron limitation in our simulations occurs in a smaller area (south of 50°S) compared to that suggested by observations (south of 40°S, Moore et al., 2013). However, stronger upwelling and shallower remineralisation in the model are also causes for high iron concentration at the surface.

We add these discussions in this paragraph of the revised manuscript and a figure for the simulated iron limitation in the Appendix.

**Comment 7**  Paragraph beginning 352: Please see Figure 2 in Buchanan et al., (2019) in Geoscientific Model Development for other model-data measures of agreement with d13C$_{\text{DIC}}$.

**Response**:  Thank you, we add the suggested reference in the revised manuscript.

**Comment 8**  Equation 23: Surely the intercept from SE$_{\text{pref}}$ is important here? Why is it excluded? Some error between SE$_{\text{pref}}$ and SE$_{\text{total}}$ must be due to differences in the intercepts?

**Response**:  By definition SE$_{\text{pref}}$ has no intercept because in the assumption of the E17 approach 13C Suess effect is proportional to pCFC-12:

$$\text{SE}_{\text{pref}} := \delta^{13}\text{C}_{\text{SE}(t-\text{PI})} = f_{\text{atm}} \cdot \delta^{13}\text{C}_{\text{SE}(t-1940)} = f_{\text{atm}} \cdot a \cdot \text{pCFC-12}_t.$$

**Comment 9**  Line 562: The underestimation of global mean d13C$_{\text{DIC}}$ by Laws parameterisation may in fact change if you preindustrial spin-ups were run for longer. Might be worth noting that here.

**Response**:   Our spin-up simulations are long enough for d13DIC to reach equilibrium, see the reply to Comment 5. Thus, the underestimation of global mean d13C$_{\text{DIC}}$ by Laws parameterisation will not change if the spin-ups are run for longer.

**Technical corrections**

**Comment 10**  Can the authors double check the coefficients in equations 5 and 7 please? The kinetic fractionation should by -0.88 per mil, and the second coefficient in equation 7 should be -1.07 according to Orr et al., (2017). Also, even though I'm sure it doesn't make much difference to the results, I find it strange that they didn't just include the full equations as detailed by Orr et al., (2017) when this has absolutely miniscule effect on computation time. In other words, there doesn't seem to be any real reason to not include the full equation that other modelling groups have implemented.

**Response**: Thank you for catching this error. Indeed equations 5 and 7 do not follow the OMIP protocol (Orr et al., 2017). We actually adopted these equations, as well as the simplification of Eq. (7) from Schmittner et al. (2013). We ran a short simulation and proved the small differences in Eqs. 5 and 7 lead to negligible changes in model results. Thus we didn't rerun and expensive simulations in this study.

In the revised manuscript we remove the statement about following the OMIP protocol. And we make a note for the differences in Eqs. 5 and 7 between this study and OMIP protocol.

**Comment 11**   Line 132: "preferentially utilised over".

**Response**:  Thanks, this is modified in the revised manuscript.

**Comment 12**   Line 335: "negative" should be "positive".

**Response**:  This is corrected in the revised manuscript.

**Comment 13**   Line 364: "generally" should be "general"

**Response**:  Thanks, modified.

**Comment 14**   Line 421: should "above" be "below"? deeper than?

**Response**:  No, here we do mean above the pCFC=20 patm isoline. The domain referred to here is illustrated in Figure 13 (area above the thick grey line). In the revised manuscript we add a reference to Figure 13 to avoid confusion.

**Comment 15**   Line 440: you mean global mean surface ocean-atmosphere right? Not global mean ocean-atmosphere, which would include all depths.

**Response**:  Yes, "global-mean surface ocean-atmosphere" is used in the revised manuscript.

**Comment 16**   Line 508: "CFC-12" should be "CFC-12-free"

**Response**:  Corrected.

**Comment 17**   Line 574: "Model" should be "Mode"

**Response**:  Corrected.

**References**

C. M. Moore, M. M. Mills, K. R. Arrigo, I. Berman-Frank, L. Bopp, P. W. Boyd, E. D. Galbraith, R. J. Geider, C. Guieu, S. L. Jaccard, T. D. Jickells, J. La Roche, T. M. Lenton, N. M. Mahowald, E. Marañón, I. Marinov, J. K. Moore, T. Nakatsuka, A. Oschlies, M. A. Saito, T. F. Thingstad, A. Tsuda, and O. Ulloa. Processes and patterns of oceanic nutrient limitation. *Nature Geoscience*, 6:701–710, 2013. doi: 10.1038/ngeo1765.

J. C. Orr, R. G. Najjar, O. Aumont, L. Bopp, J. L. Bullister, G. Danabasoglu, S. C. Doney, J. P. Dunne, J.-C. Dutay, H. Graven, S. M. Griffies, J. G. John, F. Joos, I. Levin, K. Lindsay, R. J. Matear, G. A. McKinley, A. Mouchet, A. Oschlies, A. Romanou, R. Schlitzer, A. Tagliabue, T. Tanhua, and A. Yool. Biogeochemical protocols and diagnostics for the cmip6 ocean model intercomparison project (omip). *Geoscientific Model Development*, 10(6):2169–2199, 2017. doi: 10.5194/gmd-10-2169-2017.

A. Schmittner, N. Gruber, A. C. Mix, R. M. Key, A. Tagliabue, and T. K. Westberry. Biology and air-sea gas exchange controls on the distribution of carbon isotope ratios ($\delta^{13}$C) in the ocean. *Biogeosciences*, 10(9):5793–5816, 2013. doi: 10.5194/bg-10-5793-2013.

---

## Author Comment (AC2)

**Response to the comments of Referee 2**

We are thankful for the constructive remarks of Anne Morée. In the following we reply to the comments point by point.

**General comments**

**Comment 1** In the Introduction, a stronger argument could be made for why you decided to compare the different parameterizations, and why specifically these two. Some earlier studies state that model $\delta$13C distributions are not very sensitive to the chosen parameterization (e.g., Schmittner 2013), especially not in the surface ocean (e.g. Jahn et al., 2015). It should be discussed in Sect. 3 why your model shows something else. An interesting study on biological fractionation is e.g. that of Young et al. (2013) – such results should be discussed in light of your own.

**Response**: We compare different parameterisations to choose one that is more suitable for our model. We choose the parameterisations of Popp and Laws because 1) they are of different complexities and 2) their input variables are explicitly computed in our model. Furthermore, there is, to our knowledge, only a recent study by Dentith et al. (2020) that systematically addressed the impact of Popp and Laws parameterisations on both d13POC and d13DIC distributions. The above text is incorporated in the revised Introduction.

We don't agree with the Referee #2 that our model shows contrasting conclusions from Schmittner et al. (2013) and Jahn et al. (2015). Jahn et al. (2015) show similar surface d13DIC when using different parameterisations as seen in our simulations (comparing our Fig. 7 to their Fig. 5). The similar surface d13DIC for different parameterisation is an expected result because the same atmospheric CO2 and 13CO2 are used. In the ocean interior, Jahn et al. (2015) show lower d13DIC for stronger biological fractionation, which is again consistent with this study.

Thank you for bringing the study of Young et al. (2013) to our attention. Our simulated rate of change in $\epsilon_p$ for 1960-2009 has a global mean value of $-0.026‰$ $\mathrm{yr}^{-1}$ in Hist_Popp, which it is close to Young et al. (2013)'s estimate of $-0.022‰$ $\mathrm{yr}^{-1}$. Hist_Laws shows changes in $\epsilon_p$ with a global-mean rate of $-0.005‰$ $\mathrm{yr}^{-1}$ because Laws parameterisation is less sensitive to the change of CO2(aq). Spatially Hist_Popp simulates relatively small $\epsilon_p$ change rates in easter tropical Pacific and south of 60°S, in aggreement with Young et al. (2013). The discussion on the change rate of $\epsilon_p$ is added in Section 3.1 (Isotopic signature of particular organic carbon in the surface ocean) in the revised manuscript. The spatial distributions of $\epsilon_p$ change rate are added in the Appendix.

**Comment 2** As you in Sect.3 on model results mostly discuss d13C, I think it is important to add a section under Sect. 3 where you introduce the reader to the model performance for the other tracers: Plus, even though this study is on the biogeochemistry, I think it is important for the reader

to get an introduction to the physical ocean model performance and specifics as well, and how it performs compared to obs (fe AMOC / Drake / SST / SSS / T / S / sea ice) - which probably has been described in a separate study but would be good to repeat here. I mainly stress this because d13C is governed by both circulation and biological processes, so any results obtained with your model setup also depend on the simulated circulation. Please summarize the biogeochemical performance shortly as well (i.e. how does e.g. PO4. O2, DIC and NO3 distributions compare to obs).

**Response**: We fully agree that d13C is strongly affected by both circulation and biological processes in the model. When discussing the performance of 13C tracers, we have already included the performance of several physical and biogeochemical variables. Examples can be found for the upwelling and mixed layer depth, AMOC geometry and ventilation in the North Atlantic, the Equatorial Intermediate Current System and Equatorial Deep Jets, PO4 and AOU.

In the revised manuscript, we further provide the volume transport across the Drake Passage and AMOC stream function. The distributions of sea water temperature and salinity, O2, DIC and NO3 are shown for the surface and for the zonal mean in the Atlantic, Pacific and Indian Ocean.

**Comment 3** Section 4.2 is difficult to follow. Help the reader by clarifying why you are re-calculating the E17 approach beyond that you have more data available as you use a model. Extra subsections, a thorough shortening and more focus on the results would help. I have to note that I am not an expert on the Suess effect or the E17 approach, but I feel I should be able to follow it based on my experience with d13C modelling. It seems quite some text is used to describe the figures: one could instead refer to the figures and only highlight the most important features of the figures.

**Response**: We test the E17 approach because our model simulation provides an opportunity to gain more insights into the source of the E17 approach's uncertainty because of its satisfactory performance in simulating the oceanic $\delta^{13}$C in the late 20th century, the oceanic anthropogenic $CO_2$ sink, as well as the invasion of CFC-12 into the ocean.

To improve the readability of this section, we restructure it and divide it into subsections. We also move the detailed description of the E17 approach and calculations procedures to the Appendix to make this section more focused on the results and discussion.

**Specific comments**

*Abstract*

**Comment 4** p1, l2-3: 'Direct comparison between paleo oceanic $\delta13$C records and model results facilitates assessing simulated distributions and properties of water masses in the past.'

This is true, but your study (mostly) focuses on the use of d13C in understanding the contemporary ocean, as you are able to use observations of POC-d13C and explore the Suess effect. This first sentences of your abstract sounds as if the goal of implementing d13C in HAMOCC was motivated by paleoceanographic questions only. Please rephrase.

**Response**: The above-mentioned sentence is changed to "The stable carbon isotopic composition $\delta^{13}$C is an important variable to study ocean carbon cycle across different time scales."

**Comment 5** P1, l11: 'because the latter results in a too strong preference for 12C'
I think here the reader could get confused (how can d13C_DIC be OK for both parametrizations but d13C_POC be better for Popp?). Maybe add something like '.. during C fixation, resulting in too low d13C_POC.'

**Response**: The suggested phrase is added.

**Comment 6** p1, l15: It is not entirely clear from this sentience where the 'that' refers to: Has your model ample spatial and temporal data coverage? I think you want to stress here that you can repeat the Eide et al (2017) procedure with the advantage that you model has higher temporal and spatial resolution. Please clarify/rephrase.

**Response**: We delete this half sentence starting with 'that' because it is confusing. When applying the E17 approach we also used spatially-sparse model data that were sub-sampled at the geographic locations of E17. When evaluating the underestimation we compare the estimated Suess effect to the "true" simulated Suess effect.

**Comment 7** P1, l14-20: This part about the Eide et al. approach makes up for almost one third of your abstract and does not connect so well to the first part. Why did you focus on the Eide et al approach, and how did you apply the findings from the first part of the study to the Eide et al part?

**Response**: We test Eide's approach because "The satisfactory model performance using $\epsilon_{\mathrm{p}}^{\mathrm{Popp}}$, regarding the present-day oceanic $\delta^{13}$C distribution and the anthropogenic CO$_2$ uptake, allows us to further investigate the potential uncertainties of Eide et al. (2017a)'s approach for estimating the oceanic $^{13}$C Suess effect." This sentence is incorporated in the revised text.

**Comment 8** It would also be good to finish the abstract with an outlook or overarching conclusion/summary.

**Response**: Thank you for the suggestion. We add a summary statement: "The new $^{13}$C module in the ocean biogeochemical component of MPI-ESM shows satisfying performance. It is a useful tool to study the ocean carbon sink under the anthropogenic influences and it will be applied to investigating variations of ocean carbon cycle in the past. "

*Sect. 1 Introduction*

**Comment 9**   p2, l36: Here one could add additional studies such as referring to HAMOCC2s by Heinze and Maier-Reimer (1999).

**Response**:   The suggested reference is added.

**Comment 10**   p2 l43-45: Note that the biogeochemical model in NorESM is also called HAMOCC – one could confuse the reader here by making general statements suggesting there is only one 'HAMOCC'. See Tjiputra et al., 2020 for the description of the implementation of d13C in NorESM-OC.

**Response**:   We now specify the model version HAMOCC3 here.

**Comment 11**   p3, l55: After this sentence I would expect a paragraph on both d13C_DIC (which you provide) and d13C_POC (which is missing). Let the reader know already here what data you've used of d13C_POC like you do for DIC.

**Response**:   We remove the specification of $\delta^{13}C_{POC}$ and $\delta^{13}C_{DIC}$ here because in fact we not only evaluate these two variables but also the simulated ocean physical and other ocean biogeochemical variables. In the next paragraph we focus on the oceanic 13C Suess effect, i.e. the $\delta^{13}C_{DIC}$ decrease due to anthropogenic CO2 emission.

*Sect 2. Model description*

**Comment 12**    p 4, l106: It is not only small, but also very uncertain (e.g., Zeebe and Wolf-Gladrow 2001). one could also mention here that this is commonly omitted in modeling studies. To name a few:

Schmittner, A. et al. Biology and air-sea gas exchange controls on the distribution of carbon isotope ratios ($\delta^{13}C$) in the ocean. Biogeosciences 10, 5793-5816, doi:10.5194/bg-10-5793-2013 (2013).

Lynch-Stieglitz, J., Stocker, T. F., Broecker, W. S. & Fairbanks, R. G. The influence of air-sea exchange on the isotopic composition of oceanic carbon: Observations and modeling. Global Biogeochemical Cycles 9, 653-665, doi:10.1029/95GB02574 (1995)

Tjiputra, J. F., Schwinger, J., Bentsen, M., Morée, A. L., Gao, S., Bethke, I., Heinze, C., Goris, N., Gupta, A., He, Y. C., Olivié, D., Seland, Ø., and Schulz, M.: Ocean biogeochemistry in the Norwegian Earth System Model version 2 (NorESM2), Geosci. Model Dev., 13, 2393-2431, 10.5194/gmd-13-2393-2020, 2020.

**Response**: Thank you, we add the suggested argument and references in the revised manuscript.

**Comment 13**   p4, l108-109: I suggest to move this sentence down. This would make it clearer that you first discuss total C exchange and then go into the isotope exchange.

**Response**: The suggested change is implemented.

**Comment 14**   P5, l124: You actually deviate here from the OMIP protocol of Orr et al 2017, who recommend taking 0.88 permil

**Comment 15**   p5, l126: Similarly here, you deviate from the OMIP protocol / the original formula who use 0.0144 and 0.107.

**Comment 16**   p5, l130: Why did you decide to simplify the equations here, when it computationally is not a large burden to include the whole equation? Also, I would argue again that you are not following the CMIP protocol here if you decide to simplify the air-sea gas exchange equations.

**Response**: This is the reply to Comments 14-16.

Thank you for catching these errors. Indeed equations 5 and 7 deviate from the OMIP protocol (Orr et al., 2017). We actually adopted these equations, as well as the simplification of Eq. (7) from Schmittner et al. (2013). We ran a short simulation and proved the small differences in Eqs. 5 and 7 lead to negligible changes in model results. Thus we didn't rerun and expensive simulations in this study.

In the revised manuscript we remove the statement about following the OMIP protocol. And we make a note to the differences in Eqs. 5 and 7 between this study and OMIP protocol.

**Comment 17**   P5, l132: Use 'is preferred over' instead of 'is preferentially utilised than' or rephrase in another way.

**Response**: We rephrase it as "is preferentially utilised over".

**Comment 18**   P7, l177-178: so do I understand it correctly that you base your 13C model field on 12C (i.e. total C) of the model and the PO4 of the model, after the initial spinup without the 13C? Please clarify. The initialization process of an isotope model is important for spinup duration, so detailed information on this can be valuable to the readers of your work.

**Response**: Yes, here DIC (total C) and PO4 are from the quasi-equilibrium state of the spin-up run without 13C. This is now clarified in the revised manuscript.

**Comment 19**   P8, l182-184: This input rate is the input to compensate for the loss to the sediments, right? Is it equally distributed over the surface ocean?

**Response**: Yes, it is uniformly distributed over the surface ocean. This information is added.

**Comment 20**  P8, l186-187: the inventory adjusts to be consistent with the simulated processes – what does that mean? Is the result agreeing with observational fields? This relates maybe also to my general comment that I miss an overview/summary of the non-13C performance of the model regarding both circulation and biogeochemistry.

**Response**: This sentence is inaccurate and redundant, so it is removed in the revised manuscript.

A summary of the performance of ocean physical and non-13C biogeochemical variables is added in the Appendix.

**Comment 21**  P8, l195: Is the sediment 13C also already equilibrated after the 2500y spinup?

**Response**: The sediment 13C is not in equilibrium. We note this in the revised manuscript.

*Sect. 3 Model results and observations in the late 20th century*

**Comment 22**  p8, l199: One could mention here that you are not using the Eide et al. estimate of pre-industrial d13C because that is based on her estimate of the Suess effect, which you have re-evaluated. That said, you could (like Eide et al. have done) share your Suess effect estimate as a gridded dataset. On p21, l419-420 you also explain that the E17 dataset and the Schmittner dataset are not so different, which you could mention earlier.

**Response**: Thank you for the suggestion. We add a paragraph to note the reasons for not using the Eide et al. estimate of pre-industrial d13C.

Our Suess effect estimate will be archived in the Max Planck Society Publication Repository as primary data and available to the public.

Please note the "E17 dataset" mentioned in the above comment is the field measurements of d13C, DIC, PO4, etc., which Eide et al. (2017a) used to estimate 13C Suess effect, not the gridded data of Eide et al. (2017b). The E17 dataset is not mentioned in Section 3 because it does not provide additional useful information.

**Comment 23**  P8, l200-202: Why not regrid the obs data to the model grid, instead of regridding to 1x1 and then doing the same for the model? Also you mean the model-obs comparison for d13C_DIC here, because for POC in Fig. 4 for example you are comparing model and obs without regridding – did you then take the nearest model value?

**Response**:  By re-gridding the observational data to a $1°x1°$ grid rather than to model grid (nominal resolution $1.5°$) we keep slightly more detailed spatial features. The different means of re-gridding observations only has a marginal impact on model evaluation (compare for instance

the gridded d13DIC observation in our Figs. 7 and 9 to Jahn et al. (2015)'s non-gridded d13DIC observation in Figs. 5 and 7).

For Figure 4 we also re-gridded d13POC observations. In the revised text we specify: "For the model-observation comparison, we first grid the observed $\delta^{13}C_{POC}$ and $\delta^{13}C_{DIC}$ horizontally onto a $1°x1°$ grid ..."

**Comment 24** P9, l207: Why do you start with POC here instead of DIC? I expect more readers will be familiar with the d13C_DIC distributions. Also, this Section is about simulated isotopic signature, maybe add this to the title?

**Response**: The vertical gradient of d13DIC mainly results from the biological fractionation. A number of discussions on d13DIC depend on the results and discussion about d13POC. Thus, it seems to us more logical to begin with the discussion on d13POC.

This section includes both simulation results and comparison to observations. Thus, we keep the original section title.

**Comment 25** P9, l221-224: I think you should not only refer to Appendix B when it comes to what the model simulates, but also refer to Appendix B when it comes to how that compares to observational estimates. You could summarize the most important results of Appendix B here. Also, Fig B1 is for the Popp results? Note that for clarity if you refer to model results (like also in e.g. Fig D3) it is good to say which model run you mean.

**Response**: In the revised manuscript, Appendix B is now referred to at the beginning of Section 3 to note the performance of the model.

Phytoplankton growth rate in Fig B1 is identical for Popp and Laws because it is for the total carbon and is not affected by 13C. This is now noted in the revised Appendix.

**Comment 26** P10, l242: I understand you want to make an evaluation of the model performance here of CO2_aq around 45S in Fig 4g, but why choose this particular dataset - aren't there more recent data with better coverage available?

**Response**: Thank you for the question. We chose this particular dataset because it provides contemporaneous measurements of both d13POC and CO2(aq). This is clarified in the revised manuscript.

**Comment 27** P10, l245: Hist_Laws captures more than Hist_Popp, but not all and more importantly offset by a few permil. If this offset is really very constant, wouldn't it be possible to adjust the Laws et al parameterization. One could evaluate the offset needed and see if after that offset Popp is still superior?

**Response**: As the optimisation of the Laws parameterisation is beyond the scope of this study, we use a back-of-the-envelope calculation to refit the slope and intercept of $\epsilon_p^{\text{Laws}}$ (Eq. 7).

Because $\alpha_{\text{Phy}\leftarrow\text{DIC}}$, $\alpha_{\text{aq}\leftarrow\text{g}}$ and $\alpha_{\text{DIC}\leftarrow\text{g}}$ are close to unity, $\epsilon_{\text{Phy}\leftarrow\text{DIC}} \approx \delta^{13}\text{C}_{\text{Phy}} - \delta^{13}\text{C}_{\text{DIC}}$ and $\epsilon_{\text{Phy}\leftarrow\text{DIC}} \approx \epsilon_p + \epsilon_{\text{aq}\leftarrow\text{DIC}} \approx \epsilon_p + \epsilon_{\text{aq}\leftarrow\text{g}} - \epsilon_{\text{DIC}\leftarrow\text{g}}$. As $\delta^{13}\text{C}_{\text{Phy}} \approx \delta^{13}\text{C}_{\text{POC}}$, $\delta^{13}\text{C}_{\text{POC}}$ can be approximated as

$$\delta^{13}\text{C}_{\text{POC}} \approx \epsilon_p + \epsilon_{\text{aq}\leftarrow\text{g}} - \epsilon_{\text{DIC}\leftarrow\text{g}} + \delta^{13}\text{C}_{\text{DIC}}. \tag{R1}$$

As sea water temperature (determining $\epsilon_{\text{DIC}\leftarrow\text{g}}$), CO2(aq) and growth rate are independent of the choice of $\epsilon_p$, and the surface d13DIC is only marginally affected by the choice of $\epsilon_p$ (see Section 3.2 and Figs. 5a and 5b), we can approximate $\epsilon_p^{\text{Laws}}$ using existing monthly output of the above model variables. The surface d13POC obtained with Eq. (R1) (r=0.74, NRMSE=2.8, Figs. R1a-R1c) is indeed very close to the simulated d13POC (r=0.71, NRMSR=2.5, Figs. 4b, 4d and 4f).

$$\epsilon_p^{\text{Laws}} = 68.3\,\frac{\mu}{\text{CO}_2(\text{aq})} - 24.7, \tag{7}$$

$$\epsilon_p^{\text{Laws\_V1}} = 68.3\,\frac{\mu}{\text{CO}_2(\text{aq})} - 20, \tag{R2}$$

$$\epsilon_p^{\text{Laws\_V2}} = 68.3\,\frac{\mu}{\text{CO}_2(\text{aq})} - 16.7, \tag{R3}$$

$$\epsilon_p^{\text{Laws\_V3}} = 100\,\frac{\mu}{\text{CO}_2(\text{aq})} - 20. \tag{R4}$$

As the intercept increases (comparing Eq. 7, R2 and R3), d13POC generally increases (Figs. R1a-R1i) and accordingly NRMSE decreases from 2.8 for Eq. (7) to 1.4 for Eq. (R2) and 0.75 for Eq. (R3), while the spatial relation coefficient remains unchanged.

As the slope increases (comparing Eq. R2 and R4), the d13POC difference between low and high latitudes increases (Figs. R1d-R1f, R1j-R1l) and NRMSE slightly decrease from 1.4 to 1.3. However, d13POC in the low latitude of the Atlantic shows too high variability compared to the observation. Accordingly, the spatial correlation coefficient decreases from 0.74 to 0.66.

In summary, a systematic refitting of the Laws parameterisation could be an interesting step to improve the model performance and might be considered in our future work.

**Comment 28** P11, Fig. 4: Are there uncertainty estimates available for the obs data, or would it be possible to at least shade an estimate of the uncertainty?

**Response**: The uncertainties for observations are added.

**Comment 29** P12, l281: this seems a bit repetitive, wouldn't a too steep vertical gradient always lead too too low deep d13C_DIC if surface d13C_DIC is reasonable? This could be rephrased.

[Figure]

Figure R1: As Figure 4b, 4d and 4f in the manuscript, but for d13POC estimated for $\epsilon_{\mathrm{p}}^{\mathrm{Laws}}$ (Eq. 7) using Eq. (R1) (a-c). (d-f), (g-i) and (j-l): As (a-c), but for $\epsilon_{\mathrm{p}}^{\mathrm{Laws\_V1}}$ (Eq. R2), $\epsilon_{\mathrm{p}}^{\mathrm{Laws\_V2}}$ (Eq. R3) and $\epsilon_{\mathrm{p}}^{\mathrm{Laws\_V3}}$ (Eq. R4), respectively.

**Response**: This sentenced is rephrased: " Hist_Laws generally shows too strong vertical gradients of $\delta^{13}\mathrm{C}_{\mathrm{DIC}}$ and therefore too low $\delta^{13}\mathrm{C}_{\mathrm{DIC}}$ values in the ocean interior".

**Comment 30**   P13, l284-285: this is a very important transition point for the reader, where you decide to mainly focus on Popp from now on. This could be mentioned earlier (or even in the abstract), or denoted by a new section here. For example, make a 3.2.1 and a 3.2.2 section.

**Response**: This transition does not affect the statements in the earlier text because before this transition the performance of both Hist_Popp and Hist_Laws is discussed.

We now divide this section into subsections: Section 3.2.1 on the comparison between Hist_Popp and Hist_Laws and to observed d13DIC, Section 3.2.2 on the source of surface d13DIC biases in Hist_Poppand Section 3.2.3 on the source of d13DIC biases in the interior ocean of Hist_Popp.

**Comment 31**   P13, l286: I got slightly lost here. Maybe explain the reader how this d13C_DIC comparison is different from the one in Fig. 7. It could help to make Fig 8a,b,c one figure, and present the d13C_bio and d13C_resi and the net air-sea CO2 flux separately. Regarding d13C_bio and d13C_resi I think it helps with comparison to earlier studies if you show the absolute values and not only the model-obs difference.

**Response**: We rephrase "interior-ocean $\delta^{13}\mathrm{C}_{\mathrm{DIC}}$" to "zonal-mean $\delta^{13}\mathrm{C}_{\mathrm{DIC}}$ in the Atlantic, Pacific and Indian Ocean" to make a distinction from Fig. 7.

Figure 8 is divided into two as suggested. The absolute values of d13C_bio and d13C_resi are added.

**Comment 32**  P13, l294: Why do you use $\Delta\_photo$ and not $\epsilon\_p$ like before here?

**Response**:  Because they are different variables: $\Delta\_photo$ equals $(\alpha_{\text{Phy}\leftarrow\text{DIC}} - 1) \times 10^3$ (see Eq. 8), whereas $\epsilon_p = (\alpha_{\text{Phy}\leftarrow\text{aq}} - 1) \times 10^3$.

**Comment 33**  P13, l295: Do you use the same R_C:P for the components calculations of both obs and model?

**Response**:  Yes, the same R_C:P is used. We specify this in the revised manuscript.

**Comment 34**  P13, l296: I think it would be appropriate here to remind the reader that $\epsilon\_p$ actually varies, referring e.g. to Fig. 3b,d. Also, what model MO DIC, PO4 and d13C were used? Is the R_C:P of 122 the one used in the model for consistency?

**Response**:  Sentences are added regarding the spatial variation of $\Delta\_photo$, the mean values of DIC, PO4 and d13DIC.

   Yes, R_C:P of 122 is used in the model for consistency, we specify this in the revised manuscript.

**Comment 35**  P15, l302-304: Their MO values should be included here as done for the obs.

**Response**:  The MO values for the model are added.

**Comment 36**  P17, l312-314: This is some information on the physical model performance that I think should be introduced earlier and possibly in an own subsection under Sect. 2.

**Response**:  Additional figures on physical model performance are added in the revised Appendix.

**Comment 37**  p17, l324-326: One could quantify the effect on d13C_DIC by analysing the bio and resi components for the PI run instead of the or in addition to the hist runs.

**Response**:  The purpose of decomposing d13C_DIC into d13_bio and d13_resi components is to better understand the sources of model-observation differences. There is no observational information to evaluate d13_bio and d13_resi components for the pre-industrial periods. Thus we focus on the discussion and model-data comparison for the late 20th century.

**Comment 38**  P17, l329: Here you continue to Fig. 9- the reader could use a bit more guidance here: what are you going to present and discuss here in this section/paragraph? Why do you go away from showing d13Cbio and d13Cresi?

**Response**:  This transition is now marked by a new subsection on the source of $\delta^{13}\text{C}_{\text{DIC}}$ biases in the interior ocean of Hist_Popp.

*Sect. 4 Oceanic 13C Suess effect*

**Comment 39** p18, l 367-369: Clarify here why this is important for your discussion of the Suess effect. You could also add a comment here that even though the model does well simulating the total anthropogenic C uptake, locally air-sea exchange fluxes deviate from obs (Fig. 8f).

**Response**: Our motivation to discuss Suess effect is stated in the previous paragraph: "The oceanic $^{13}$C Suess effect, could serve as benchmark for ocean models to evaluate the uptake and re-distribution of the anthropogenic $CO_2$ emissions in the ocean".

A statement on the local biases in air-sea flux is added.

**Comment 40** P19, l385: This section is quite long and heavy, and I have to admit I did not follow all of it. I would suggest to cut it up in different subsections, which can discuss the different aspects which you have investigated. Start early in Sect. 4.2 with why it is relevant to re-evaluate the approach by E17 (e.g., is it often used?). Large parts of the text also feel like a methods section ? could more of this section be moved to a supplement/appendix C, such that more focus on the results can be given here?

**Response**: See our reply to Comment 3.

**Comment 41** P20, l393: You have only discussed total global anthropogenic C uptake, and you have compared to E17 at depth (Sect. 4.1) - now you are going to explore the E17 underestimation after concluding that your model produces similar results to E17? This is somewhat of a confusing step.

**Response**: This comparison is for the qualitative behaviour of 13C Suess effect, whereas E17's underestimation only has some quantitative impact. It is expected that our model produces similar pattern as E17. This is because E17's Suess effect estimate is proportional to pCFC-12. The oceanic distributions of CFC-12 and 13C Suess effect are both dominated by the uptake of the atmospheric signal and the subsequent transport in the ocean, and these processes are well simulated in our model.

The above discussion is incorporated in the revised manuscript.

**Comment 42** p26, l518: How does this compare to observational estimates (e.g. Young et al., 2013)?

**Response**: See the last paragraph of our reply to Comment 1.

**Comment 43**   p27, l565: One should add a short paragraph here summarizing your d13C_bio and d13C_resi component analysis for the hist_Popp run.

**Response**:   We add sentences on findings based on this decomposition: "Our model slightly overestimates surface $\delta^{13}C_{DIC}$. By decomposing $\delta^{13}C_{DIC}$ into a biological component and a residual component, we find the overestimation in the high latitude ocean is dominated by biases in the biological component. ".

**Comment 44**   P28, l582: the Popp et al., 1989 parameterization has a satisfactory performance for the PI and historical times. I would agree this encourages reliability for paleoclimatic simulations, but I think some more critical remarks are in place (which should come before Sect. 5). E.g., in the past $\epsilon$_p was possibly different due to different ecosystem structures or other influences, Redfield ratios could have changed (Ödalen et al., 2020).

**Response**:   We add a text to note the potential limitations of $\epsilon$_p in long-term simulations for the past climate.

Past changes of C/P ratios discussed in Ödalen et al. (2020) affect the production of 13C-depleted organic matter (according to Eq. 8) and its export to the deep ocean. Changes of C/P ratios do not directly affect parameterisation of the biological fractionation, thus no discussion is added on C/P ratios.

**Technical corrections**

**Comment 45**   p6, l146: Zeebe

**Comment 46**   P17, l335: positive biases between 1000 and 3000m

**Comment 47**   p19, l387: the 13C Suess effect

**Comment 48**   p20, l391: at 200 m depth

**Comment 49**   p27, l561: again yields slightly better agreement

**Comment 50**   p28, l574: Mode Water and explains

**Comment 51**   p26, l528: Fig. D7h?

**Response**:   The technical corrections in Comments 45 - 51 are implemented in the revised manuscript.

**References**

J. E. Dentith, R. F. Ivanovic, L. J. Gregoire, J. C. Tindall, and L. F. Robinson. Simulating stable carbon isotopes in the ocean component of the famous general circulation model with moses1 (xoavi). *Geoscientific Model Development*, 13(8):3529–3552, 2020. doi: 10.5194/gmd-13-3529-2020.

M. Eide, A. Olsen, U. S. Ninnemann, and T. Eldevik. A global estimate of the full oceanic $^{13}$c suess effect since the preindustrial. *Global Biogeochemical Cycles*, 31(3):492–514, 2017a. doi: 10.1002/2016GB005472.

M. Eide, A. Olsen, U. S. Ninnemann, and T. Johannessen. A global ocean climatology of preindustrial and modern ocean $\delta^{13}$c. *Global Biogeochemical Cycles*, 31(3):515–534, 2017b. doi: 10.1002/2016GB005473.

A. Jahn, K. Lindsay, X. Giraud, N. Gruber, B. L. Otto-Bliesner, Z. Liu, and E. C. Brady. Carbon isotopes in the ocean model of the community earth system model (cesm1). *Geoscientific Model Development*, 8(8):2419–2434, 2015. doi: 10.5194/gmd-8-2419-2015.

M. Ödalen, J. Nycander, A. Ridgwell, K. I. C. Oliver, C. D. Peterson, and J. Nilsson. Variable C/P composition of organic production and its effect on ocean carbon storage in glacial-like model simulations. *Biogeosciences*, 17(8):2219–2244, 2020. doi: 10.5194/bg-17-2219-2020.

J. C. Orr, R. G. Najjar, O. Aumont, L. Bopp, J. L. Bullister, G. Danabasoglu, S. C. Doney, J. P. Dunne, J.-C. Dutay, H. Graven, S. M. Griffies, J. G. John, F. Joos, I. Levin, K. Lindsay, R. J. Matear, G. A. McKinley, A. Mouchet, A. Oschlies, A. Romanou, R. Schlitzer, A. Tagliabue, T. Tanhua, and A. Yool. Biogeochemical protocols and diagnostics for the cmip6 ocean model intercomparison project (omip). *Geoscientific Model Development*, 10(6):2169–2199, 2017. doi: 10.5194/gmd-10-2169-2017.

A. Schmittner, N. Gruber, A. C. Mix, R. M. Key, A. Tagliabue, and T. K. Westberry. Biology and air-sea gas exchange controls on the distribution of carbon isotope ratios ($\delta^{13}$C) in the ocean. *Biogeosciences*, 10(9):5793–5816, 2013. doi: 10.5194/bg-10-5793-2013.

J. N. Young, J. Bruggeman, R. E. M. Rickaby, J. Erez, and M. Conte. Evidence for changes in carbon isotopic fractionation by phytoplankton between 1960 and 2010. *Global Biogeochemical Cycles*, 27(2):505–515, 2013. doi: 10.1002/gbc.20045.